# Formation and degradation of strongly reducing cyanoarene-based radical anions towards efficient radical anion-mediated photoredox catalysis

Yonghwan Kwon[1,2], Jungwook Lee[1], Yeonjin Noh[1,2], Doyon Kim[2], Yungyeong Lee[1], Changhoon Yu[1], Juan Carlos Roldao[3,4], Siyang Feng[3], Johannes Gierschner ®[3] ✉, Reinhold Wannemacher ®[3] ✉ & Min Sang Kwon ®[1] ✉

Cyanoarene-based photocatalysts (PCs) have attracted significant interest owing to their superior catalytic performance for radical anion mediated photoredox catalysis. However, the factors affecting the formation and degradation of cyanoarene-based PC radical anion (PC•−) are still insufficiently understood. Herein, we therefore investigate the formation and degradation of cyanoarene-based PC•− under widely-used photoredox-mediated reaction conditions. By screening various cyanoarene-based PCs, we elucidate strategies to efficiently generate PC•− with adequate excited-state reduction potentials ($E_{red}^*$) via supra-efficient generation of long-lived triplet excited states (T$_1$). To thoroughly investigate the behavior of PC•− in actual photoredox-mediated reactions, a reductive dehalogenation is carried out as a model reaction and identified the dominant photodegradation pathways of the PC•−. Dehalogenation and photodegradation of PC•− are coexistent depending on the rate of electron transfer (ET) to the substrate and the photodegradation strongly depends on the electronic and steric properties of the PCs. Based on the understanding of both the formation and photodegradation of PC•−, we demonstrate that the efficient generation of highly reducing PC•− allows for the highly efficient photoredox catalyzed dehalogenation of aryl/alkyl halides at a PC loading as low as 0.001 mol% with a high oxygen tolerance. The present work provides new insights into the reactions of cyanoarene-based PC•− in photoredox-mediated reactions.

Over the last decade, visible light-driven photoredox catalysis that utilizes the energy of photons has risen to prominence in organic synthesis owing to its mild conditions, high tolerance to various functional groups, and unique operating mechanism[1–6]. Upon photo-excitation, a photoredox catalyst (PC) can participate in single-electron transfer (SET) events with substrates, consequently generating reactive radical intermediates from a variety of bench-stable substrates. This formerly inaccessible reaction strategy has enabled significant developments in radical chemistry for organic[7–13] and polymer synthesis[14–20].

To further enhance the efficiency and expand the reaction scope of visible light-driven photoredox catalysis, it is essential to maximize

**Fig. 1 | Schematic illustration of the current work.** Reaction scheme of the formation and photodegradation of cyanoarene-based photocatalyst radical anion (PC$^{\bullet-}$). Here, ISC, $T_1$, PET, and Sub denote intersystem crossing, triplet excited state, photoinduced electron transfer, and substituent, respectively.

the reducing power and concentration of active PC species that activate the substrate of interest through an electron transfer (ET) process. This can mostly be facilitated in a reductive quenching cycle wherein a one-electron-reduced PC (i.e., PC$^{\bullet-}$) commonly acts as an active PC species because PC$^{\bullet-}$ usually exhibits a far longer lifetime than the optically excited PC species (i.e., $^{1,3}$PC$^*$) acting as an active PC intermediate in an oxidative quenching cycle[21]. Moreover, PC$^{\bullet-}$ is regarded as a core intermediate for the recently proposed multiphoton excitation catalysis mechanism based on consecutive photoinduced electron transfer (ConPET)[22–24] and electrophotocatalysis[25–27], therefore, merits special attention.

The concentration of PC$^{\bullet-}$ in the photostationary state and the ground state reduction potential of the PC ($E_{red}^0$(PC)) both play critical roles in photoredox-mediated catalytic reactions employing PC$^{\bullet-}$ as an active species[21]. A high concentration of PC$^{\bullet-}$ implies a high collision frequency with the substrate under illumination by visible light, which facilitates ET events. In addition, a more negative $E_{red}^0$ indicates an increase in the driving force for ET, thus accelerating ET processes. To achieve a highly negative $E_{red}^0$ of a PC, a high energy lowest unoccupied molecular orbital (LUMO) is required[15]. Furthermore, to ensure visible light absorption by such a PC, the energy of the highest occupied molecular orbital (HOMO) should scale with that of the LUMO; however, the accompanying decrease in $E_{red}^*$ is detrimental to the photoinduced electron transfer (PET) between a PC and a reductant. In other words, it is very difficult to target PCs combining the following properties: (i) good visible light absorption, (ii) adequate initial PET with a sacrificial reductant, and iii) a highly negative reduction potential. Thus, the generation of PC$^{\bullet-}$ is normally targeted in PCs with a less negative reduction potential (e.g., perylene diimide[22,28], Acr-Mes$^+$BF$_4^-$[8,29], Rh6G[30], and Ru(bpy)$_3$Cl$_2$[7,31–33]) or under special reaction conditions[23,34,35] (Supplementary Table 1 in the Supplementary Information (SI)).

As exemplified by the recent reports by the groups of Zhang[36], Zeitler[37], Kwon[15], and others[11,38–44], cyanoarenes have emerged as attractive organic PCs. Such PCs exhibit excellent catalytic performances for a variety of visible light-driven organic reactions[38–40] and polymerizations[45–49]. Among them, 4DP-IPN and its analogs have attracted considerable interest owing to their superior catalytic performance for radical anion-mediated photoredox catalysis. For example, Wickens et al. reported the successful photocatalyzed reductive cleavage of strong C(sp$^2$)−N and C(sp$^2$)−O bonds by the electrochemically generated 4DP-IPN$^{\bullet-}$[41]. More recently, the groups of Wickens[42] and Wu[43] used 4DP-IPN analogs as PCs to perform the phosphonylation, borylation, and hydroarylation of highly inactivated aryl chlorides. Through careful characterization of the radical anion of 4DP-IPN, they proposed that its high reducing power ($E_{red}^0$ = −1.66 V vs SCE, all redox potentials in the current work are against saturated calomel electrode (SCE) unless otherwise noted) is the crucial factor.

Meanwhile, recent studies on cyanoarene-based PC$^{\bullet-}$ have revealed that a photodegradation of PCs is involved in photoredox catalysis[50–53], which might induce unwanted catalytic activities of the photodegraded adducts. However, despite these research efforts, it is still unclear which factors affect the formation and degradation of the radical anion of cyanoarene-based PCs. This lack of understanding can lead to inefficiencies in radical anion-mediated photoredox reactions such as inappropriate choice of PC, excessive PC loading, and inadequate selection of the excitation source. However, no studies have focused on the in-depth investigation of the cyanoarene-based PC$^{\bullet-}$.

Herein, we investigate the formation and degradation of the cyanoarene-based PC$^{\bullet-}$ under widely-used photoredox-mediated reaction conditions (Fig. 1). Through the investigation of various cyanoarene-based PCs with different redox potentials and abilities to generate triplet excited states ($T_1$), we found that organic PCs exhibiting both the ultra-efficient generation of long-lived $T_1$ and adequately positive excited state reduction potentials ($E_{red}^*$(PC)) enable these PCs to efficiently form strongly reducing PC$^{\bullet-}$ under mild visible light illumination. During the screening of these cyanoarenes, we also found that the different photodegradation behaviors of PCs depend on their electronic and steric properties. We also identified a strong correlation between the photodegradation reaction of cyanoarenes and the abilities of PCs to be one-electron reduced. To further investigate the photodegradation behavior of PC$^{\bullet-}$ in actual photoredox catalysis, we carried out the reductive dehalogenation of aryl halides as a model reaction. From in situ monitoring of the reaction, we revealed that the dehalogenation and photodegradation of PCs are co-dependent on the rate of the ET process. Furthermore, we demonstrated the highly efficient dehalogenation of aryl/alkyl halides at a very small loading of 4DP-IPN with a high oxygen tolerance; it thus outperformed other conventional PCs that were used as controls.

## Results
### PET event between 4DP-IPN and sacrificial reducing agents
We first investigated the initial PET from amine-based sacrificial donors to 4DP-IPN; tertiary amines were used because they are the most commonly used sacrificial reducing agents. Given the ground state oxidation potentials of the prepared amines ($E_{ox}^0$(DIPEA) = 0.68 V, $E_{ox}^0$(TBA) = 0.88 V, and $E_{ox}^0$(TEA) = 0.96 V, where DIPEA, TBA, and TEA represent diisopropylethylamine, triethylamine, and tributylamine, respectively)[54,55], and the excited state reduction potential of the PC ($E_{red}^*$(4DP-IPN) = 0.63 V), the PET is expected to be very slow owing to its unfavorable thermodynamics. To monitor the PET, we conducted photoluminescence (PL) decay quenching experiments using time-correlated single-photon counting (TCSPC). The UV-Vis absorption and PL emission spectra of 4DP-IPN are shown in Supplementary Fig. 3. Under degassed conditions without a sacrificial electron donor, the decay lifetimes of the prompt and delayed components of 4DP-IPN

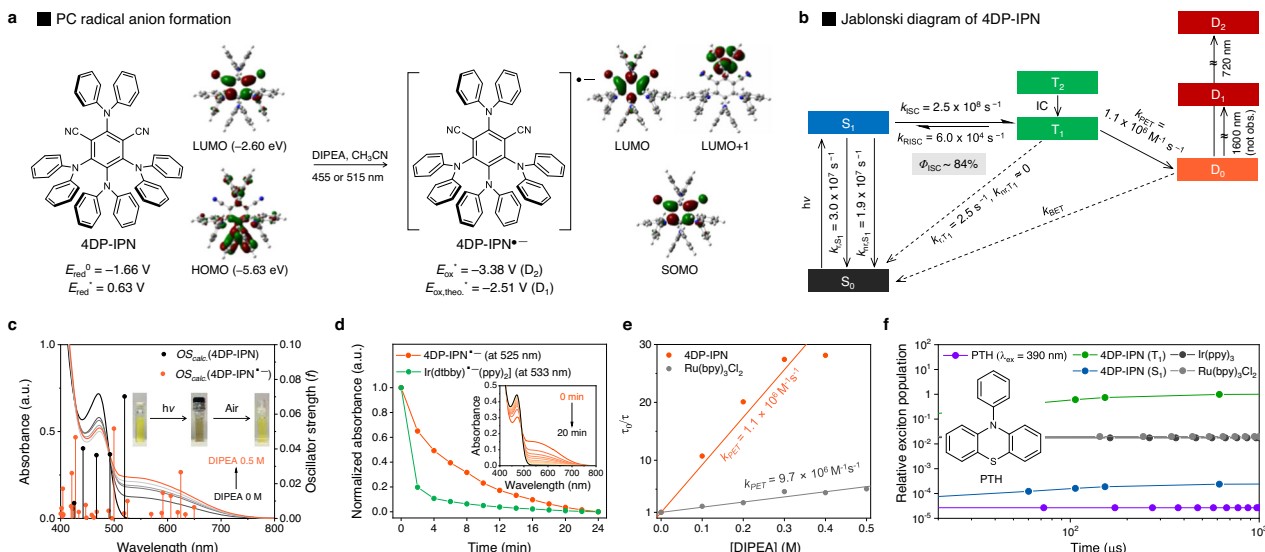

**Fig. 2 | Characterization of 4DP-IPN and 4DP-IPN·⁻. a** Reaction scheme of the formation of 4DP-IPN·⁻. The calculated frontier molecular orbitals (MO) topologies of 4DP-IPN and 4DP-IPN·⁻ are shown. Their excited state redox potentials ($E_{red}^*$(PC) and $E_{ox}^*$(PC·⁻)) were estimated from $E_{red}^* = E_{0-0} + E_{red}^0$ and $E_{ox}^*$(PC·⁻) $= -E_{0-0}$(PC·⁻) $+ E_{red}^0$(PC); $E_{0-0}$(PC) and $E_{0-0}$(PC·⁻) were evaluated by the onset of gated photo-luminescence (PL) emission spectrum in CH₃CN at 65 K and the onset of UV-Vis absorption spectrum at room temperature (RT), respectively. **b** Jablonski diagram of 4DP-IPN. The rate constants of all photophysical processes were evaluated in the current work; here, IC, (R)ISC, $D_0$, and $S_1/T_1/D_n$ denote internal conversion, (reverse)intersystem crossing, doublet ground state and singlet/triplet/doublet excited state, respectively. **c** UV-Vis absorption spectra of 4DP-IPN (black line) and 4DP-IPN·⁻ (orange line) in CH₃CN; here, a.u. denotes arbitrary units. UV-Vis absorption spectra of 4DP-IPN·⁻ were taken right after illumination by two 3 W

515 nm LEDs for 1 min at RT. TD-DFT results (oscillator strengths) are shown as stick spectra. **d** Time-dependent changes of the UV-Vis absorbance of 4DP-IPN·⁻ at 525 nm and Ir(dtbby)·⁻(ppy)₂PF₆ at 533 nm (compare Supplementary Fig. 9). PC·⁻ was generated under the illumination of two 3 W 515 nm LEDs for 3 min (for 4DP-IPN) or two 3 W 455 nm LEDs for 1 min (for Ir(dtbby)(ppy)₂PF₆) at RT. Changes in the UV-Vis absorption spectrum of freshly generated 4DP-IPN·⁻ were recorded every 2 min under dark conditions (inset). **e** Stern–Volmer plots for the PL decays quenching of 4DP-IPN and Ru(bpy)₃Cl₂ in CH₃CN by DIPEA at RT. **f** Results of kinetics simulation of the relative excited state population of selected PCs (5.0 × 10⁻³ M) over time under continuous 455 nm (or 390 nm for 10-phenylphenothiazine (PTH)) irradiation (see the SI for the full details of the kinetics simulation).

were measured to be 3.3 ns and 104 µs, respectively (Supplementary Fig. 5). Adding tertiary amines significantly shortened the decay life-times of the delayed components (Supplementary Fig. 6), whereas those of the prompt components changed negligibly (Supplementary Fig. 7), implying that $T_1$ is mainly responsible for the PET events. In fact, the measured rate for PET is larger than the reverse intersystem crossing (RISC) rate and smaller than the ISC rate, further supporting this argument (Fig. 2b). Among the sacrificial donors, DIPEA exhibited the strongest quenching effect (Fig. 2e and Supplementary Fig. 6). The PET rate constants ($k_{PET}$) increased with an increasing thermodynamic driving force ($-\Delta G_{PET}$), consistent with Marcus normal region behavior (Supplementary Fig. 6)[56]. As a control, we performed a PL quenching experiment with DIPEA on Ru(bpy)₃Cl₂, which is widely used as a PC for reductive quenching photocatalytic cycles[31–33]. In the presence of the same amount of DIPEA, $k_{PET}$ for Ru(bpy)₃Cl₂ ($9.7 \times 10^6$ M⁻¹ s⁻¹) was higher than that for 4DP-IPN ($1.1 \times 10^6$ M⁻¹ s⁻¹), consistent with the trend in the excited state reduction potentials of the PCs ($E_{red}^*$(Ru(bpy)₃Cl₂) = 0.77 V)[57].

Interestingly, for the same amount of DIPEA, the PL decay quenching in 4DP-IPN changed to a greater extent than that in Ru(bpy)₃Cl₂. This indicates that the rate of PET ($v_{PET}$) from DIPEA to the PC is faster in 4DP-IPN than that in Ru(bpy)₃Cl₂, although the ther-modynamic driving force for PET is less favorable in 4DP-IPN. We attribute this finding to the highly efficient long-lived $T_1$ generation by 4DP-IPN. In fact, the excited state population of PC is a crucial factor for generating the PC·⁻ because the molar rate of PET, $v_{PET}$ (in M s⁻¹), is described by

$$v_{PET} = k_{PET}[PC^*][Q] \qquad (1)$$

where $[PC^*]$ is the concentration of the PC in the excited state and $[Q]$ is the concentration of the quencher (i.e., DIPEA). We then modeled the time-dependent excited state concentrations of selected PCs after turning on irradiation; this model was based on the rate law to estimate the PET ability of 4DP-IPN and to compare it to those of other well-known PCs (Fig. 2f; see also Supplementary Fig. 13 in the SI for details of the simulation and the rate equations employed). The rate constants used in the kinetics simulations were obtained from either our experiments or the literature. In the photostationary state, the concentration of 4DP-IPN molecules in $T_1$ was approximately 10² times higher than those of both Ru(bpy)₃Cl₂ and Ir(ppy)₃ and was approximately 10⁵ times higher than that of 10-phenylphenothiazine (PTH), which has commonly been used as a highly reducing PC ($E_{ox}^*$(PTH) = −2.10 V)[58]. This indeed suggests that 4DP-IPN is super-ior for PET with sacrificial agents, despite its unfavorable thermodynamics.

**Generation of PC·⁻ of cyanoarene-based PCs**

The formation of the PC·⁻ of 4DP-IPN can be directly monitored by UV-Vis absorption spectroscopy. As the PC·⁻ is known to be sensitive to oxygen and moisture[8,59], all samples were prepared inside a glove box. A degassed solution of 4DP-IPN was first irradiated with a 515 nm light emitting diode (LED) for 1 min in the presence of an excess amount of DIPEA; in this context, it should be noted that exogenous tertiary amines have been reported to be able to suppress back electron transfer (BET) events[60]. The UV-Vis absorption spectra of the 4DP-IPN solution were recorded before and immediately after irradiation (Fig. 2c). Spectral changes were observed, in agreement with a color change that could be observed by the naked eye (Fig. 2c, inset). A decrease in the absorption peak at 470 nm implied the depletion of

4DP-IPN, while a new broad absorption band appeared at 500–750 nm, indicating the generation of 4DP-IPN$^{\cdot-}$; perfect isosbestic points (at 428 and 493 nm) also appeared for the reaction. Once the resulting PC$^{\cdot-}$ was exposed to air, close to 100% of the 4DP-IPN was rapidly regenerated, demonstrating the nearly perfect reversibility of this transformation (Supplementary Fig. 8a). The stability of the PC$^{\cdot-}$ was also measured by recording changes in the UV-Vis absorption spectrum of freshly generated PC$^{\cdot-}$ every 2 min under dark conditions (orange line in Fig. 2d). A gradual decay was observed in the dark over more than 20 min, which is approximately two times longer than that of Ir(dtbby)$^{\cdot-}$(ppy)$_2$PF$_6$ reported by König et al. under similar conditions (green line in Fig. 2d)[23]. This superior stability of 4DP-IPN$^{\cdot-}$ is likely due to retarded BET in the triplet contact radical ion pair involving the 4DP-IPN$^{\cdot-}$, which is related to the lack of heavy atoms and, thus, a decreased spin–orbit coupling.

The formation of the PC$^{\cdot-}$ was further confirmed by (time-dependent) density functional theory, (TD-)DFT, calculations (using the B3LYP functional, 6–311++G* basis set, and the polarizable continuum model, PCM, with acetonitrile as the solvent); such calculations provide insights into the underlying electronic situation. In the neutral form of 4DP-IPN (black line in Fig. 2c), the low-energy absorption band consisted of a number of transitions with partial charge transfer (CT) character and a relatively low oscillator strength $f$ (Supplementary Fig. 14); for instance, S$_1$ was well described by a transition from the HOMO to the LUMO (HOMO→LUMO), as depicted in Fig. 2a. Furthermore, the TD-DFT calculations correctly reproduced the appearance of a red-shifted absorption band for the PC$^{\cdot-}$ (orange line in Fig. 2c), which was shown to consist of a multitude of CT and locally excited (LE) transitions with a small $f$; here, the D$_2$ state corresponded to the excitation from the singly occupied molecular orbital (SOMO) to the LUMO+1, which was of LE character but had a small differential overlap, thus generating a small $f$ (Fig. 2a and Supplementary Fig. 14). Note, however, that the theoretical evidence clearly indicates the appearance of a near-infrared (NIR) band for the PC$^{\cdot-}$, which corresponded to the D$_1$ state (SOMO→LUMO; Fig. 2b and Supplementary Fig. 14), independent of the chosen DFT functional. At the B3LYP level of theory, it appeared at 1454 nm. We thus recorded the absorption spectrum in the NIR range to 1500 nm but did not find evidence for such absorption in this range (Supplementary Fig. 10). However, an absorption band might possibly have appeared at a wavelength >1500 nm and was therefore not detected. Nonetheless, note that the energy difference between the bands at 1450 and 1600 nm, for example, is only ~0.1 eV, which is within the error of the DFT calculations.

Next, we investigated the formation of PC$^{\cdot-}$ for a set of cyanoarene-based PCs with different redox potentials and abilities to generate T$_1$; nine additional PCs were synthesized with different donor moieties (Fig. 3 and Supplementary Fig. 11). In most cases, the low-energy absorption bands appeared to red-shift relative to those of the PC; even 4-p,p-DCDP-IPN shows broad absorption band in the NIR region (Fig. 3j). These results indicate the broad applicability of our strategy. Furthermore, to verify the relevance of T$_1$ for PC$^{\cdot-}$ formation, two 4DP-IPN analogs (4-p-MCDP-IPN and 4-o,p-DCDP-IPN) with better $E_{red}^*$ (i.e., a lower HOMO) but a negligibly small concentration of T$_1$, were prepared (Fig. 3, Supplementary Fig. 5, and Supplementary Table 4); this finding might be due to fast RISC mediated by vibronic coupling[61], and further in-depth investigations are currently underway. Interestingly, PC$^{\cdot-}$ was not noticeably generated for such PCs in our experimental conditions, clearly confirming that the long-lived T$_1$ generation of a PC is crucial for the formation of PC$^{\cdot-}$. Nevertheless, in most strongly twisted donor–acceptor structures, $^3$PC$^*$ is efficiently generated; furthermore, by changing the donor and/or acceptor moieties, the redox potentials are delicately controlled over a broad range, enabling the use of radical ions with tailored redox potentials for a variety of highly efficient conventional photoredox catalysis, multiphoton excitation catalysis, and photoelectrocatalysis[13,15].

## Photodegradation behavior of 4DP-IPN

The PC$^{\cdot-}$ formation was determined to be sensitive to the wavelength and intensity of irradiation. Spectral changes similar to those in the solutions of 4DP-IPN and DIPEA under 515 nm irradiation were observed when irradiated with a 455 nm LED for 1 min; however, the spectrum did not fully recover even after exposure to air, implying that the PC is likely to be degraded by 455 nm irradiation (Supplementary Fig. 8). In fact, continuous 455 nm LED irradiation over 5 min resulted in the photodegradation of 4DP-IPN (Fig. 4a and Supplementary Fig. 8c). These results suggest that the PC$^{\cdot-}$ can be efficiently generated under 455 nm LED irradiation, although followed by molecular degradation or unwanted chemical reactions[50–53].

We carefully monitored the degradation of 4DP-IPN in the presence of the sacrificial reductant (i.e., DIPEA) alone under both 455 and 515 nm LED irradiation, and the degradation under 455 nm illumination was much faster than that under 515 nm illumination. This can possibly be ascribed to differences in the absorption efficiencies ($\varepsilon = 9.0 \times 10^3$ M$^{-1}$ cm$^{-1}$ at 455 nm and $\varepsilon = 4.5 \times 10^2$ M$^{-1}$ cm$^{-1}$ at 515 nm). As illustrated in Fig. 4a, 4DP-IPN decomposed into two green luminescent compounds. After scaling up the reaction, we successfully isolated the photodegraded adducts. Intensive structural characterization via 1D/2D NMR analyses and mass spectroscopy clearly confirmed that one of the two CN groups of 4DP-IPN was substituted with a methyl group or hydrogen atom to yield 4DP-Me-BN (major product) and 4DP-H-BN (minor product), respectively (Supplementary Figs. 20, 21).

Control experiments were performed in combination with DFT calculations to investigate the mechanistic pathway of the 4DP-IPN degradation under the given conditions (Supplementary Figs. 18 and 19). Interestingly, no degradation could be identified by thin-layer chromatography (TLC) in the absence of DIPEA (Supplementary Fig. 18a), suggesting that DIPEA plays an important and specific role in the photodegradation process of 4DP-IPN. We thus assumed that long-lived 4DP-IPN$^{\cdot-}$ first formed in the presence of DIPEA, followed by the methyl and hydrogen substitution reaction of the PC$^{\cdot-}$. Here, the methyl and hydrogen seem to have been provided through the $\beta$-scission of the one-electron-oxidized adduct of DIPEA (DIPEA$^{\cdot+}$). In fact, the C−C (and C−H) bonds located in the β-position of DIPEA$^{\cdot+}$ are well known to be substantially weaker than those of neutral DIPEA; hence, $\beta$-scission normally occurs to generate the radical species[42,62–66], which was also well reproduced by our DFT calculations (Fig. 4b). Moreover, the possibility that the methyl and hydrogen originated from the solvent (CH$_3$CN) could be ruled out because in deuterated acetonitrile (CD$_3$CN), the CH$_3$ substitution reaction was still observed instead of the CD$_3$ substitution (Fig. 4a).

We further investigated the photodegradation of other cyanoarene-based PCs. As illustrated in Fig. 5, very interesting patterns emerged, which relied on the electronic properties (i.e., propensity for PC$^{\cdot-}$ formation) and structural features (i.e., steric environments nearby the CN group) of PCs. As in 4DP-IPN, methyl substitution at one of the two CN groups was observed for 3DP-Cz-IPN and 3DP-DCDP-IPN, both of which have similar electronic and structural properties compared with 4DP-IPN (Fig. 5a). However, in 4Cz-IPN and 4tCz-IPN, ethyl substitution occurred with a small amount of methyl substitution, suggesting that the steric environment near the CN group is a crucial factor in the substitution reaction (Fig. 5b). The ethyl group likely originates from the C−N bond cleavage of DIPEA$^{\cdot+}$ assisted by a 1,2-methyl shift. Indeed, the use of diisopropylmethylamine instead of DIPEA generated a CH$_3$-substituted adduct as a major product, which clearly supports our hypothesis.

Figure 5c shows the results of photodegradation experiments for the PCs in which PC$^{\cdot-}$ was not properly formed. No photodegradation was observed for 3DP-DMDP-IPN or 4-o,p-DCDP-IPN, whereas complex

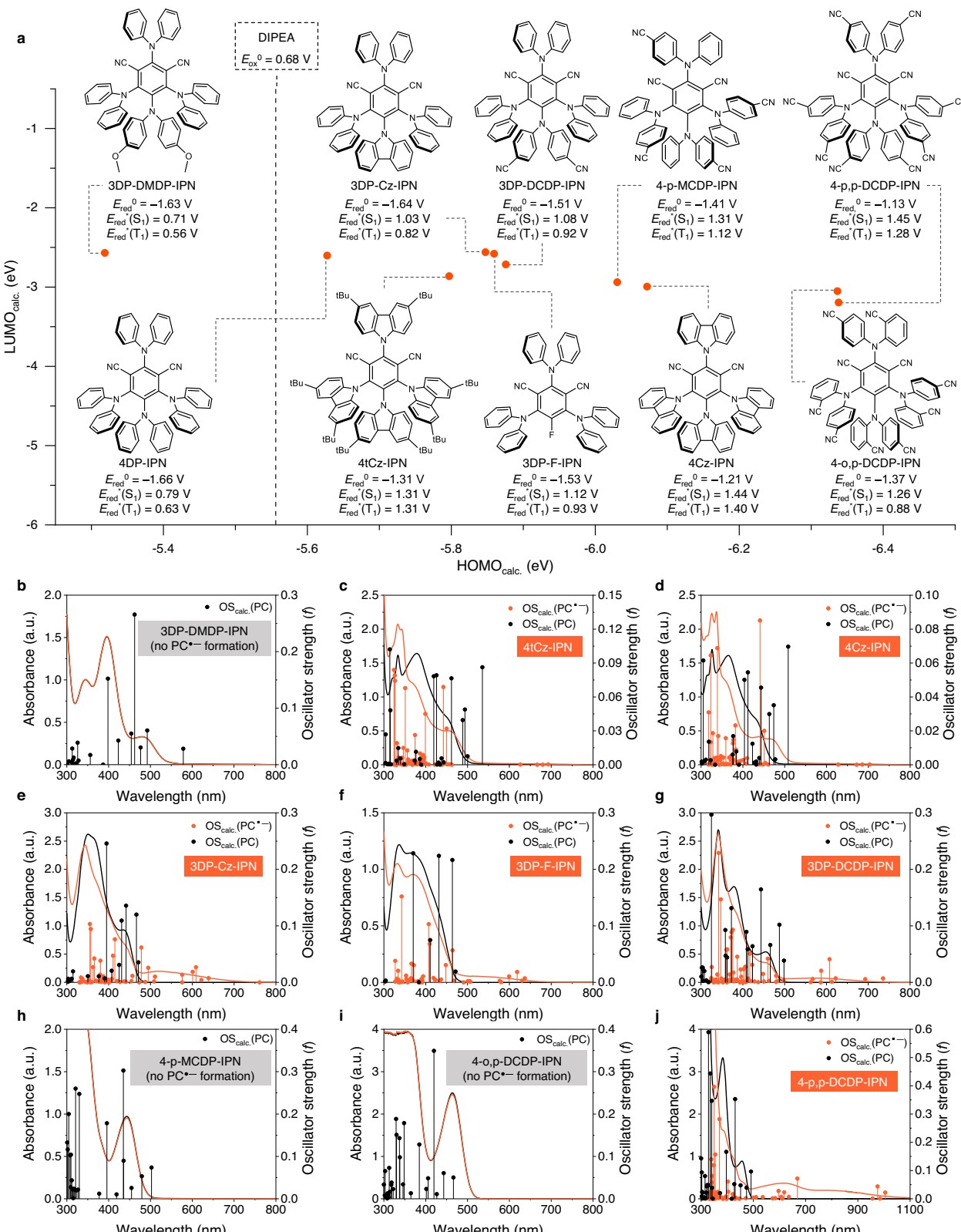

**Fig. 3 | Formation of PC$^{•-}$ of various cyanoarene-based PCs. a** Chemical structures of selected cyanoarene-based PCs and their calculated HOMO and LUMO energies. UV-Vis absorption spectra of selected PC (black line) and PC$^{•-}$ (orange line). It should be noted that the PCs prepared here contain six completely new compounds (3DP-DMDP-IPN, 3DP-Cz-IPN, 3DP-DCDP-IPN, 4-p-MCDP-IPN, 4-o,p-DCDP-IPN, and 4-p,p-DCDP-IPN). All ground state reduction potentials of PCs ($E_{red}^0$(PC)) were measured in the current work and their excited state reduction potentials ($E_{red}^*$(PC)) were estimated from $E_{red}^* = E_{0-0} + E_{red}^0$; $E_{0-0}$(S$_1$) and $E_{0-0}$(T$_1$) were evaluated by the onset of PL emission and gated PL emission, respectively, in CH$_3$CN at 65 K (except for 4tCz-IPN in DMF). UV-Vis absorption spectra were taken from the degassed solutions of PCs (1.0 × 10$^{-4}$ M) and DIPEA (0.5 M) in CH$_3$CN right after illumination of two 3 W 455 nm LEDs for 1 min at RT, **b** 3DP-DMDP-IPN, **c** 4tCz-IPN, **d** 4Cz-IPN, **e** 3DP-Cz-IPN, **f** 3DP-F-IPN, **g** 3DP-DCDP-IPN, **h** 4-p-MCDP-IPN, **i** 4-o,p-DCDP-IPN, and **j** 4-p,p-DCDP-IPN. All solutions were prepared in a glove box and fully degassed. TD-DFT calculation results (oscillator strengths) are shown as stick spectra.

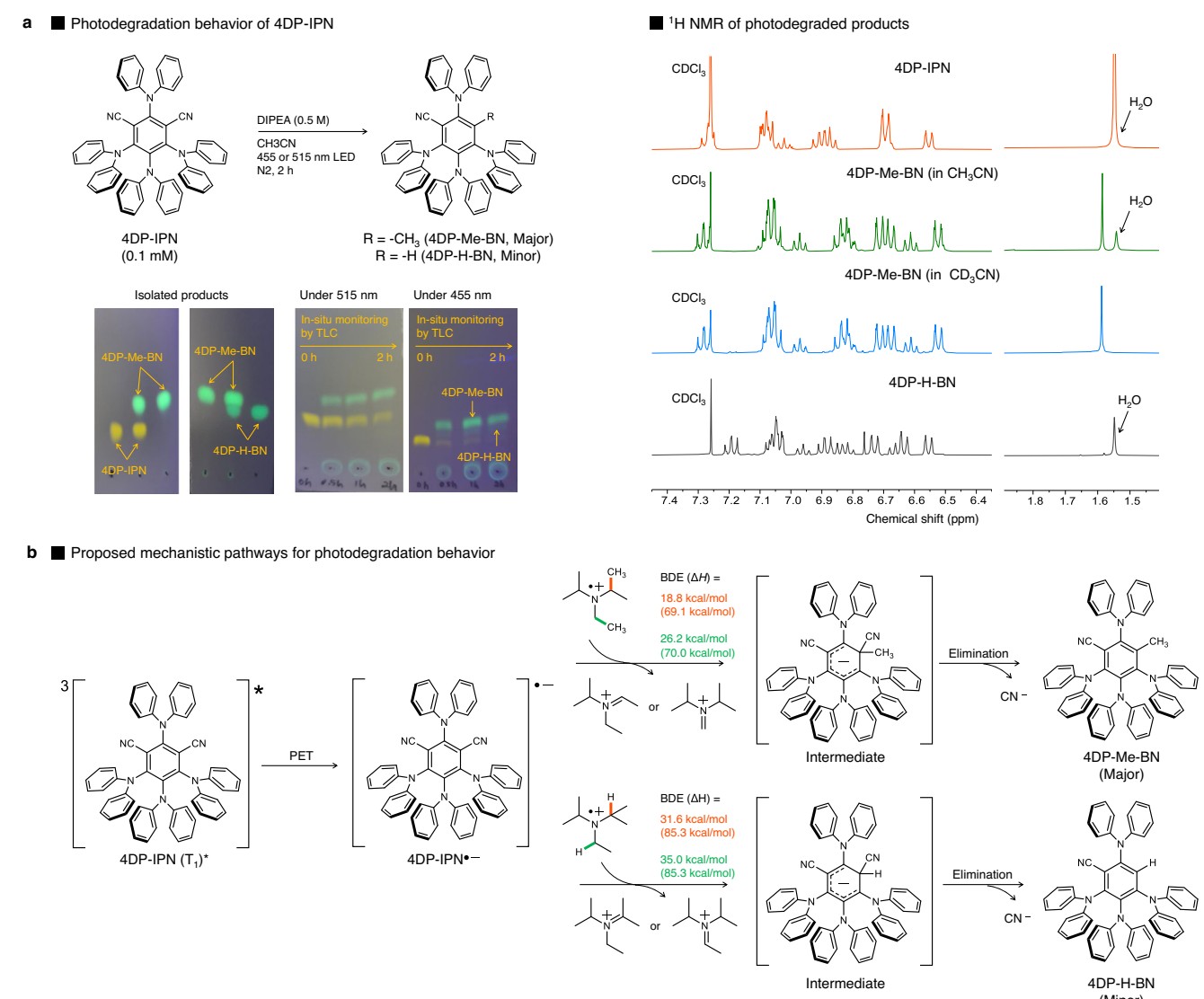

**Fig. 4 | Photodegradation behavior of 4DP-IPN. a** Reactions were performed with 4DP-IPN ($1.0 \times 10^{-4}$ M) and DIPEA (0.5 M) in $CH_3CN$ under the illumination of two 3 W 515 nm LEDs or two 3 W 455 nm LEDs at RT. PC degradations were monitored in situ by TLC with eluent conditions ($CH_2Cl_2$:hexanes, 7:3 v/v). The photodegraded products were isolated by column chromatography, and [1]H NMR spectra confirmed that a methyl (and hydrogen) substitution reaction occurred at the CN position of 4DP-IPN to yield 4DP-Me-BN (and 4DP-H-BN). **b** Proposed mechanistic pathway for the photodegradation behavior of 4DP-IPN in the presence of DIPEA and DFT cal-culations for the bond dissociation energies ($\Delta H$) in DIPEA$^{\cdot+}$; the values in par-enthesis correspond to the calculated bond dissociation energies in DIPEA.

degradation mixtures were formed in 4-p-MCDP-IPN. These results imply that in such PCs, no well-defined degradation pathway through the PC$^{\cdot-}$ intermediate exists, and thus, the photodegradation behavior is determined by the intrinsic photostability of the PCs. Finally, we examined the photodegradation behavior of 3DP-F-IPN and 4-p,p-DCDP-IPN, which effectively generated PC$^{\cdot-}$ and contain additional labile groups such as C−F bonds or other types of C−CN bonds. Complex reaction mixtures formed for both PCs, which was probably due to the degradation of these labile groups (Fig. 5d). Further in-depth investigations are currently underway to fully understand the photodegradation behaviors of cyanoarene-based PCs.

**Dehalogenation of activated aryl/alkyl halides**
Based on the experimental results discussed above, we assumed that the supra-efficient PC$^{\cdot-}$ generation of 4DP-IPN and its highly negative $E_{red}^0$ of −1.66 V would enable the very efficient dehalogenation of aryl/alkyl halides. To that end, we first examined the feasibility of reducing 4-bromobenzonitrile ($E_{red}^0$ = −1.83 V), which was chosen as a substrate owing to its moderate reactivity, thereby allowing us to compare the

catalytic performance of 4DP-IPN with those of well-known PCs. The reaction conditions were optimized by irradiating a mixture of 4-bromobenzonitrile, 4DP-IPN (0.005 mol%), and tertiary amines in degassed acetonitrile with a 455 nm LED light at room temperature. Although no conversion was achieved in the presence of five equiva-lents of TEA, the reduction product, benzonitrile, was obtained in 100% yield after 8 h using ten equivalents of DIPEA (Supplementary Table 5). This result is consistent with the PL quenching experiments, clearly indicating that the efficient generation of PC$^{\cdot-}$ is a key factor in proceeding with the dehalogenation reaction. Notably, 455 nm LED irradiation gave better results (i.e., faster reaction kinetics) than 515 nm LED irradiation, which seems to contradict the UV-Vis absorption results concerning the photodegradation of the PC$^{\cdot-}$. This incon-sistency was most likely due to the fact that the ET between the PC$^{\cdot-}$ and the substrate occurred much faster than the photodegradation of the PC$^{\cdot-}$. To confirm our hypothesis, we investigated the photo-degradation behavior of 4DP-IPN in actual reactions to dehalogenate various aryl halides (Fig. 6a) in which ten equivalents of DIPEA were used as a sacrificial agent. Aliquots of each reaction mixture were taken

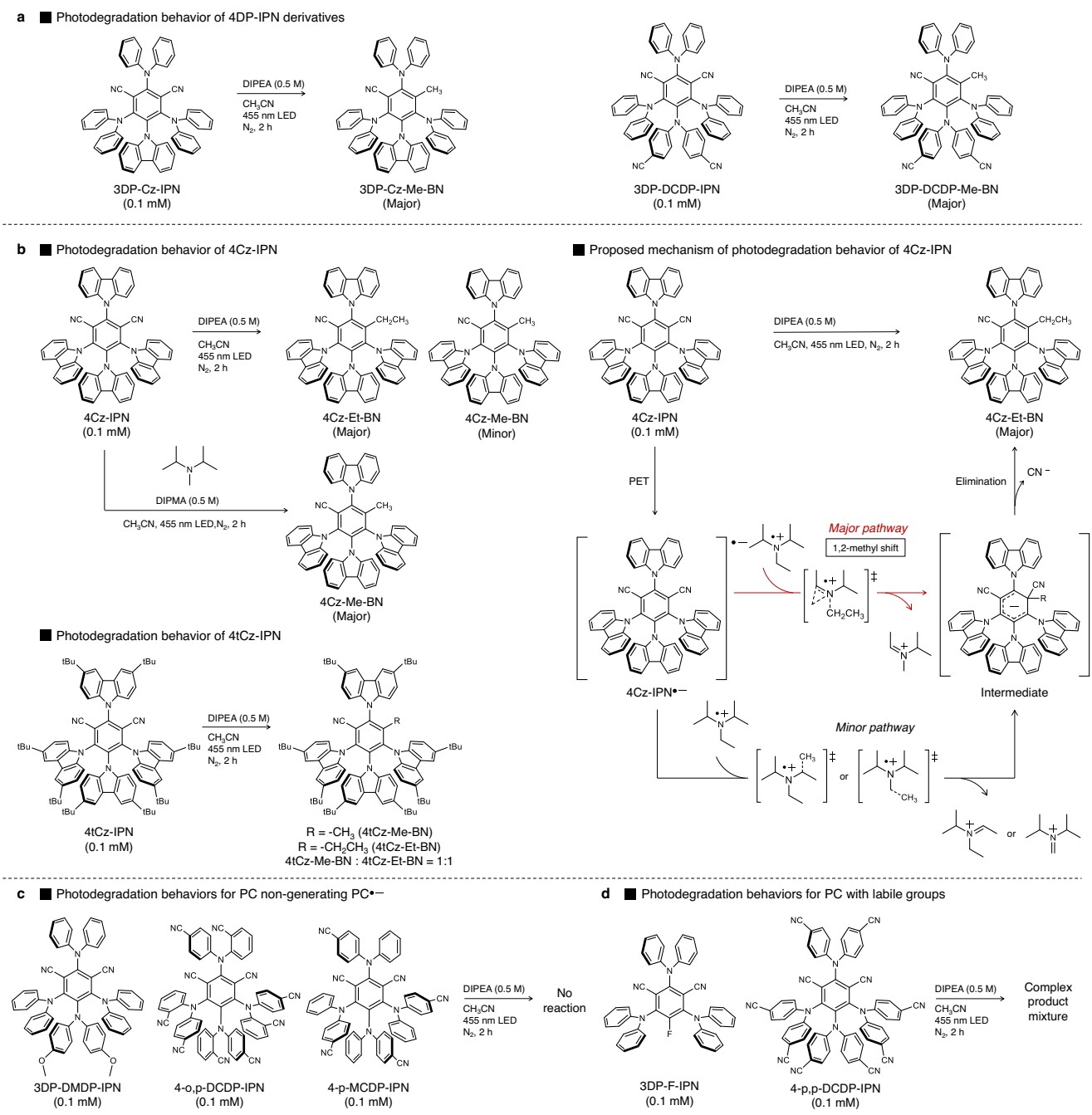

**Fig. 5 | Photodegradation behaviors of various cyanoarene-based PCs. a** Photo-degradation behavior of 3DP-Cz-IPN and 3DP-DCDP-IPN. **b** Photodegradation behaviors of 4Cz-IPN and 4tCz-IPN and the proposed mechanism of photo-degradation in the presence of DIPEA and DIPMA as a reducing agent. **c** Photodegradation behavior of PCs non-generating PC[•−]. **d** Photodegradation behaviors of PCs with labile groups. For the characterizations of the isolated products, see Supplementary Figs. 20–26 in the SI.

at a given time to monitor the progress of the reactions as well as the PC degradation. As only a small amount of PC (here, 1 mol%) was used, the degradation of the PC was monitored by TLC, whereas the reaction was tracked by [1]H NMR.

As shown in Fig. 6a: Reaction A, an 100% yield was obtained for 4-bromobenzonitrile in 1.5 h, while only a trace amount of the photo-degradation product, 4DP-Me-BN, appeared; this implies that 4DP-IPN is an active PC for the dehalogenation of 4-bromobenzonitrile. More interestingly, the degradation of the PC was significantly retarded during the dehalogenation reaction compared to that in the presence of DIPEA alone (Fig. 6a: TLC). This was presumably due to the fact that the PC degradation competes with the dehalogenation reaction,

as described in Fig. 6c. In other words, in the presence of 4-bromo-benzonitrile, the ET from 4DP-IPN[•−] to 4-bromobenzonitrile seemed to be significantly faster than (i) the formation of 4DP-IPN[•−] and (ii) the substitution reaction to form 4DP-Me-BN, which results in a substantial delay in the PC degradation. In fact, faster PC degradation occurred in the dehalogenation reactions of more challenging substrates (i.e., 4-chlorobenzonitrile, 4-iodoanisole, and 4-bromoanisole in Fig. 6a: Reaction B, C, and D, respectively), further supporting our hypothesis.

In a further step, we examined the catalytic activity of the PC degradation adduct, 4DP-Me-BN (Fig. 6b), using 1 mol% of 4DP-Me-BN to proceed with the dehalogenation reactions of 4-bromobenzonitrile and 4-bromoanisole; note that the reactions proceeded under the

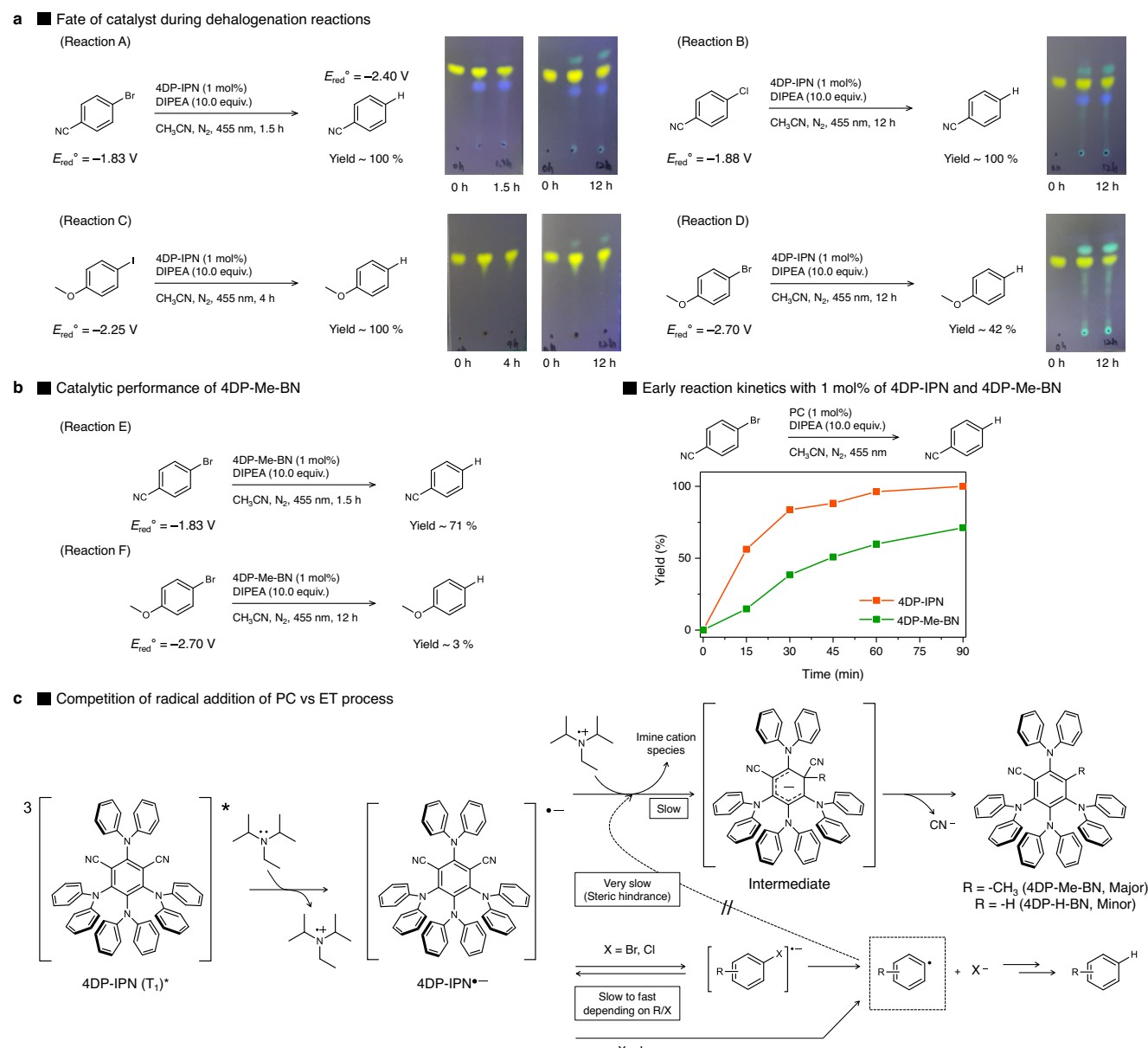

**Fig. 6 | Photodegradation of 4DP-IPN in actual dehalogenation reactions. a** Fate of catalyst during dehalogenation reactions. Reactions were performed with substrates (0.1 M), DIPEA (10.0 equiv.), and 4DP-IPN (1 mol%, $1.0 \times 10^{-3}$ M) in CH$_3$CN (1 mL) under the illumination of two 3 W 455 nm LEDs for several hours at RT. Yields were determined by $^1$H NMR using 1,3,5-trimethoxybenzene as an internal standard. PC degradation was monitored by in situ TLC with eluent conditions (EA:hexanes, 1:4 v/v). All redox potential values were obtained from the literature where the

potential values were measured against the standard calomel electrode (SCE)[67,72,73]. **b** Catalytic performance of 4DP-Me-BN. Reactions were performed under the same conditions as in **a**. Early reaction kinetics of the dehalogenation reactions with 4DP-IPN (orange line) and 4DP-Me-BN (green line) were monitored on a 2 mL scale; the yields were determined by $^1$H NMR using 1,3,5-trimethoxybenzene as an internal standard. **c** Proposed mechanistic pathways for the photodegradation of 4DP-IPN in the presence of aryl halides.

same conditions. With 4DP-Me-BN (4DP-IPN) as a PC, 71% (100%) and 3% (42%) yields were obtained for 4-bromobenzonitrile and 4-bromoanisole, respectively, indicating the lower catalytic performance of 4DP-Me-BN. Reaction kinetics studies of 4-bromobenzonitrile further confirmed the lower catalytic activity of 4DP-Me-BN than that of 4DP-IPN (Fig. 6b, right). More interestingly, when the continuous irradiation was prolonged to 12 h after the completion of the dehalogenation reaction of 4-bromobenzonitrile, only a small amount of the PC degradation adduct appeared, suggesting that the PC degradation was still retarded in the absence of 4-bromobenzonitrile (Fig. 6a: Reaction A, TLC). This might have been due to the fact that the reaction product, benzonitrile ($E_{red}° = −2.40$ V)[67], can act as an electron acceptor instead of 4-bromobenzonitrile, competing with the PC degradation process.

Finally, the molecular origin of the lower catalytic activity of 4DP-Me-BN was studied. According to DFT calculations, the substitution of CN by CH$_3$ did not significantly change the overall three-dimensional structure with respect to that of 4DP-IPN, thus still providing a strongly twisted donor–acceptor conformation (Fig. 7a). Nevertheless, the electronic structure substantially changed in 4DP-Me-BN. The substitution of the strongly electron-withdrawing CN group with the weakly electron-donating CH$_3$ in the central acceptor moiety increased both the HOMO and LUMO energies. However, the increase of 0.47 eV in the LUMO energy was considerably larger than that of the HOMO energy (0.11 eV) because the acceptor (where the LUMO is located; see Figs. 2a and 7a) was directly affected by CH$_3$ substitution, while the HOMO (mainly located on the donor) was only indirectly affected by

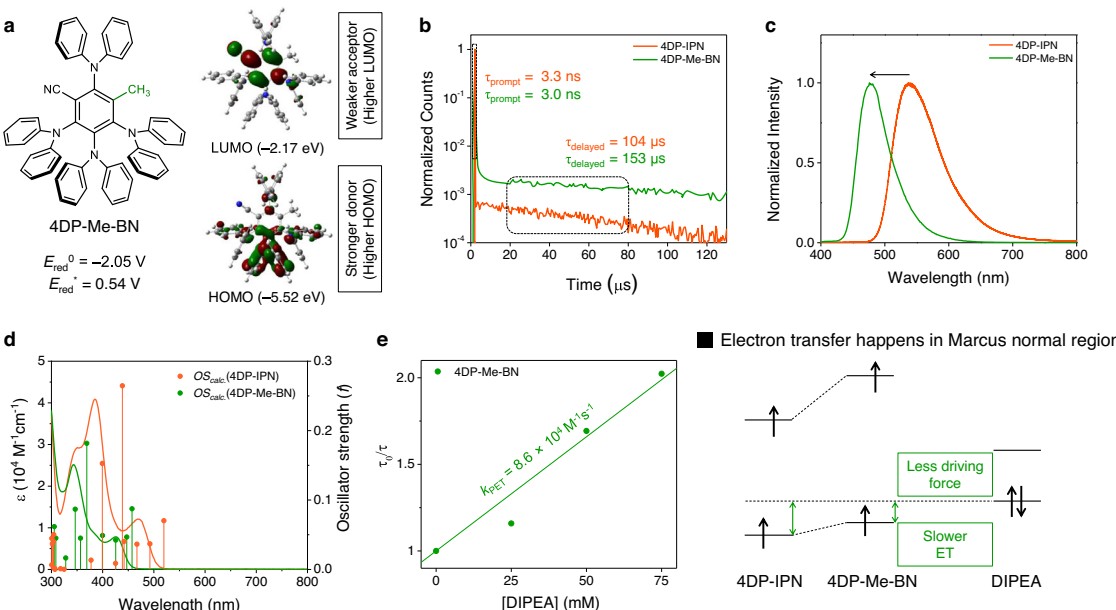

**Fig. 7 | Characterization of 4DP-Me-BN. a** Calculated energies and topologies of the frontier MO of 4DP-Me-BN. **b** PL decay of 4DP-IPN ($1.0 \times 10^{-5}$ M; orange line) and 4DP-Me-BN ($1.0 \times 10^{-5}$ M; green line) in $CH_3CN$ at RT. **c** Steady-state PL emission spectra of 4DP-IPN ($1.0 \times 10^{-5}$ M; orange line) and 4DP-Me-BN ($1.0 \times 10^{-5}$ M; green line) in $CH_3CN$ at RT. **d** UV-Vis absorption spectra of 4DP-IPN ($1.0 \times 10^{-5}$ M; orange line) and 4DP-Me-BN ($1.0 \times 10^{-5}$ M; green line) in $CH_3CN$ at RT. TD-DFT results (oscillator strengths) are shown as stick spectra. **e** Stern–Volmer plots for the PL quenching of 4DP-Me-BN ($1.0 \times 10^{-5}$ M) in $CH_3CN$ by DIPEA at RT. Stern–Volmer plots were obtained from PL decays of 4DP-Me-BN, excitation at $\lambda_{ex} = 377$ nm, and detection at $\lambda_{det} = 470$ nm.

the inductive effect caused by the decrease in the electron-withdrawing power of the acceptor unit.

Photophysical and electrochemical measurements of PCs species are well in accordance with their DFT calculations. Strongly delayed fluorescence with a long decay lifetime of 153 µs was seen in 4DP-Me-BN (Fig. 7b), indicating an intact thermally activated delayed fluorescence cycle not too different from that in 4DP-IPN. The UV-Vis absorption and PL emission spectra of 4DP-Me-BN showed a distinct blue shift with respect to those of 4DP-IPN (Fig. 7c, d), confirming the steeper increase in the LUMO energy upon $CH_3$ substitution than that of the HOMO. Finally, the lower excited state reduction potential ($E_{red}^* = 0.54$ V) and ground state reduction potential ($E_{red}^0 = -2.05$ V) of 4DP-Me-BN compared to those of 4DP-IPN (Supplementary Fig. 3) were also well supported by the DFT results, which revealed that 4DP-Me-BN exhibited higher HOMO and LUMO energies.

The fact that the catalytic performance of 4DP-Me-BN was worse than that of 4DP-IPN might be surprising, considering that 4DP-Me-BN has a (i) highly negative reduction potential, (ii) decent long-lived $T_1$ generation, and (iii) greatly improved stability. However, we concluded that the catalytic activity might have been impaired by i) the very low light absorption of 4DP-Me-BN ($\varepsilon = 1.1 \times 10^3$ M$^{-1}$ cm$^{-1}$ at 455 nm), as observed in the UV-Vis absorption spectrum (Fig. 7d), and (ii) a decrease in the PET rate between 4DP-Me-BN and DIPEA owing to the lowered excited state reduction potential of 4DP-Me-BN. In fact, the $k_{PET}$ of 4DP-Me-BN in the presence of DIPEA measured by a PL quenching experiment was $8.6 \times 10^4$ M$^{-1}$ s$^{-1}$, approximately one order of magnitude lower than that of 4DP-IPN, which was consistent with Marcus normal region behavior described above (Fig. 7e), thus supporting our hypothesis.

Using the optimized conditions, we screened different PCs. As shown in Table 1, 4DP-IPN outperformed the other PCs used for comparison. Ru(bpy)$_3$Cl$_2$ and Ru(phen)$_3$(PF$_6$)$_2$, which are commonly used as PCs for reductive cycles, gave very low yields of 13 and 0%, respectively, even at high catalyst loadings of 5 mol%; this might have been caused by the less negative reduction potential of the catalysts ($E_{red}^0$(Ru(bpy)$_3^{2+}$) = −1.33 V and $E_{red}^0$(Ru(phen)$_3^{2+}$) = −1.36 V) compared

to that of 4DP-IPN (−1.66 V)[57]. Rh6G and PDI employed as PCs for ConPET showed yields of 2 and 4%, respectively, at 5 mol% PC loadings. Although the one-electron-reduced forms of those PCs are known to have highly negative excited state oxidation potentials ($E_{ox}^*$(PDI$^{•-}$) = −1.87 V and $E_{ox}^*$(Rh6G$^{•-}$) = −2.40 V)[28,30], the excited state populations of Rh6G$^{•-}$ and PDI$^{•-}$ were low because of their short excited state lifetimes; this might have led to inefficient ET with 4-bromobenzonitrile and hence very low yields. On the other hand, Ir(ppy)$_3$, known as a PC for oxidative cycles[9], showed decent catalytic activity, but to give comparable yields, it required 100 times higher PC loading (0.5 mol%) compared to that of 4DP-IPN. The higher required catalyst loading might have been due to the relatively shorter excited state lifetime of Ir(ppy)$_3$ compared with that of 4DP-IPN$^{•-}$, despite the highly negative excited state oxidation potential of Ir(ppy)$_3$ ($E_{ox}^*$(Ir(ppy)$_3$) = −1.73 V). Finally, the reaction scope was explored in the presence of 4DP-IPN (0.05–0.005 mol%) for a range of differently substituted aryl/alkyl halides with ground state reduction potentials of less than −2.2 V, which gave corresponding reduction products in nearly quantitative yields (Fig. 9, Activated aryl/alkyl halides).

## Oxygen tolerance

At a higher 4DP-IPN concentration of 0.05 mol%, the reaction of 4-bromobenzonitrile gave a quantitative yield without degassing (Fig. 8b), which enabled a facile gram-scaled reaction under ambient conditions without any preceding degassing (Fig. 9). To understand the origin of this high oxygen tolerance, the kinetics of the reaction with and without degassing were monitored. An inhibition period clearly appeared under non-degassed conditions but not under fully degassed conditions (Fig. 8c). These results imply that dissolved oxygen in the mixture solutions was fully consumed before starting the reductive dehalogenation reaction. According to the previous reports[68–70], oxygen could be consumed through the generation of reactive oxygen species (i.e., singlet oxygen and superoxide anion) and their subsequent reaction with DIPEA and/or its cationic intermediates. The kinetics of the photoredox catalytic reactions of 4-bromobenzonitrile with and without degassing process were then

**Table 1 | Results of reductive dehalogenation of 4-bromobenzonitrile with a variety of PCs**

| Entry | PC | Loading (mol%) | Yield (%) |
|---|---|---|---|
| 1 | – | – | 0 |
| 2 | Ru(bpy)$_3$Cl$_2$ | 5 | 13 |
| 3 | Ru(phen)$_3$(PF$_6$)$_2$ | 5 | 0 |
| 4 | Rh6G | 5 | 2 |
| 5[a] | PDI | 5 | 0 |
| 6[a,b] | | 5 | 4 |
| 7 | 4Cz-IPN | 5 | 75 |
| 8[a] | Ir(ppy)$_3$ | 5 | 100 |
| 9 | | 0.5 | 100 |
| 10 | | 0.05 | 26 |
| 11 | | 0.005 | 0 |
| 12 | 4DP-IPN | 5 | 100 |
| 13 | | 0.5 | 100 |
| 14 | | 0.05 | 100 |
| 15 | | 0.005 | 100 |
| †16[c] | | 0.005 | 0 |
| †17[d] | | 0.005 | 0 |

Reactions were performed with 4-bromobenzonitrile (0.1 M), PC (5–0.005 mol%), and DIPEA (10.0 equiv.) in CH$_3$CN (1 mL) under the illumination of two 3 W 455 nm LEDs for 8 h at RT. All solutions were prepared inside a glove box and degassed by bubbling with Ar. Yields were determined by GC-FID.
†1H NMR using 1,3,5-trimethoxybenzene as an internal standard.
[a]Reaction performed in DMF due to solubility.
[b–d]Control experiments were performed [b]with TEA instead of DIPEA, [c]in the absence of DIPEA, or [d]in the dark conditions.

simulated using the mechanistic model described in Fig. 8a (see Supplementary Fig. 15 and Supplementary Table 7 in the SI for details of the kinetics simulation); all parameters including the rate constants used in the kinetics simulations were obtained from the experiments or from the literature. As shown in Fig. 8c (green line with open squares), the experimental trends were quite well reproduced by the kinetics simulations, supporting our hypothesis. Although these trends were well simulated, a substantial discrepancy appeared between the simulated and experimental kinetics, which was likely due to the fact that ET was overestimated, and BET was not considered in the model. In fact, the nonlinearity observed in the experimental kinetics was adequately reproduced in the simulations by considering the adequate BET for a slower ET (Fig. 8c; green line with open circles). However, significant discrepancies were still observed for dehalogenation reactions in the presence of oxygen (Fig. 8c, right). This is probably due to the depletion of the tertiary amines by reactive oxygen species, which is not considered in model[69,70]. indeed, in air, the oxygen tolerance was determined depending on the amount of DIPEA (Supplementary Table 8).

The experimental results thus far suggest that the dehalogenation of activated aryl/alkyl halides at very low PC loadings and the high oxygen tolerance arise from the highly efficient generation of both the long-lived T$_1$ of 4DP-IPN and the PC$^{•-}$ with a highly negative $E_{red}^0$(4DP-IPN) of −1.66 V. In particular, high PC$^{•-}$ concentrations apparently led to the highly efficient dehalogenation of the halides by overcoming the potential barrier of -0.5 V without involving the ConPET process. In fact, the addition of 4-bromobenzonitrile into the solution of 4DP-IPN$^{•-}$ induced the perfect regeneration of 4DP-IPN, clearly indicating that direct ET occurs between 4DP-IPN$^{•-}$ and 4-bromobenzonitrile, and

thus, that ConPET was not involved in this case (Supplementary Fig. 12).

### Dehalogenation of inactivated aryl/alkyl halides

Finally, the dehalogenation of inactivated aryl/alkyl halides was explored. The $E_{red}^0$ values of the substrates were ranged from −2.2 to −3.0 V, which were expected to be difficult to overcome by conventional photoredox catalysis. However, despite the very high potential energy barrier, moderate to high yields were mostly attained at higher PC loadings (5–0.5 mol%). We thus presumed that ConPET was involved in the dehalogenation of inactivated halides. In fact, the excited state oxidation potential of 4DP-IPN$^{•-}$ was estimated to be −3.38 V according to Eq. (2), which corresponds to the D$_2$ state; relying on Koopmans theorem, this corresponds to an energy of −1.30 eV. According to the (TD-)DFT calculations, the D$_1$ state could also contribute to ConPET. However, considering the energies and corresponding reducing powers (i.e., $E_{ox}^*$(4DP-IPN$^{•-}$ at D$_1$) = −2.51 V), the D$_2$ state was likely to be responsible for ConPET. In fact, the excited state lifetime of D$_2$ was expected to be sufficiently long, as the energy gap between D$_2$ and D$_1$ (0.87 eV from $E_{theo}$(D$_1$−D$_2$) = $E_{0-0,exp}$(PC$^{•-}$) − $E_{theo}$(D$_0$−D$_1$), Supplementary Fig. 14) was similar to that between D$_1$ and D$_0$ (0.85 eV, Supplementary Fig. 14); this should effectively slow down the internal conversion from D$_2$ and thus enhance the D$_2$ lifetime, not considering here any barriers to internal conversion via conical intersections, which could additionally stabilize the D$_2$ state.

$$E_{ox}^*(PC^{•-}) = -E_{0-0}(PC^{•-}) + E_{red}^0(PC) \qquad (2)$$

Thus, the excited state oxidation potential of 4DP-IPN$^{•-}$ was high enough to reduce the inactivated aryl/alkyl halides. Despite similar reduction potentials, aryl chloride gave a lower yield than aryl bromide, even with higher PC loadings (i.e., chlorobenzene vs 4-bromoanisole, see Fig. 9). This might be attributed to the slow bond dissociation of the aryl chloride anion owing to the strong C–Cl bond, facilitating BET from the aryl chloride anion to PC or DIPEA derivatives.

The participation of ConPET in the case of 4-bromoanisole was corroborated by the following semi-quantitative quenching experiments (Supplementary Fig. 12). In fact, the addition of an excess amount of 4-bromobenzonitrile ($E_{red}^0$ = −1.83 V) to the preformed solution of 4DP-IPN$^{•-}$ distinctly changed the color from dark green to yellow in a few seconds, which is indicative of the regeneration of 4DP-IPN. In contrast, the solution of 4DP-IPN$^{•-}$ showed no evident color change in the presence of excess 4-bromoanisole ($E_{red}^0$ = −2.70 V). This suggests that fast ET between 4DP-IPN$^{•-}$ and 4-bromoanisole might be ruled out and that a multiphoton excitation process (i.e., ConPET) might be involved. However, given the low to moderate yields of inactivated aryl chlorides, it seems that the ConPET process was not very efficient here (Fig. 9). This was presumably due to the short excited state lifetime of 4DP-IPN$^{•-}$, which was supported by the fact that steady-state PL emission was not observed for the solution of 4DP-IPN$^{•-}$ (Supplementary Fig. 10). Of course, other possible pathways, such as a halogen atom transfer (XAT) by α-aminoalkyl radicals recently proposed by Leonori et al.[71], cannot be excluded. Further investigation of the excited state dynamics of PC$^{•-}$ and related ConPET processes is underway.

### Discussion

We investigated the formation and degradation of one-electron reduced cyanoarene-based PCs under widely-used photoredox-mediated reaction conditions and found that the cyanoarenes exhibiting both ultra-efficient generation of long-lived T$_1$ and adequately positive $E_{red}^*$ enabled these PCs to efficiently form PC$^{•-}$ under mildly visible light illumination. We further investigated (i) the photodegradation of various cyanoarene-based PCs with

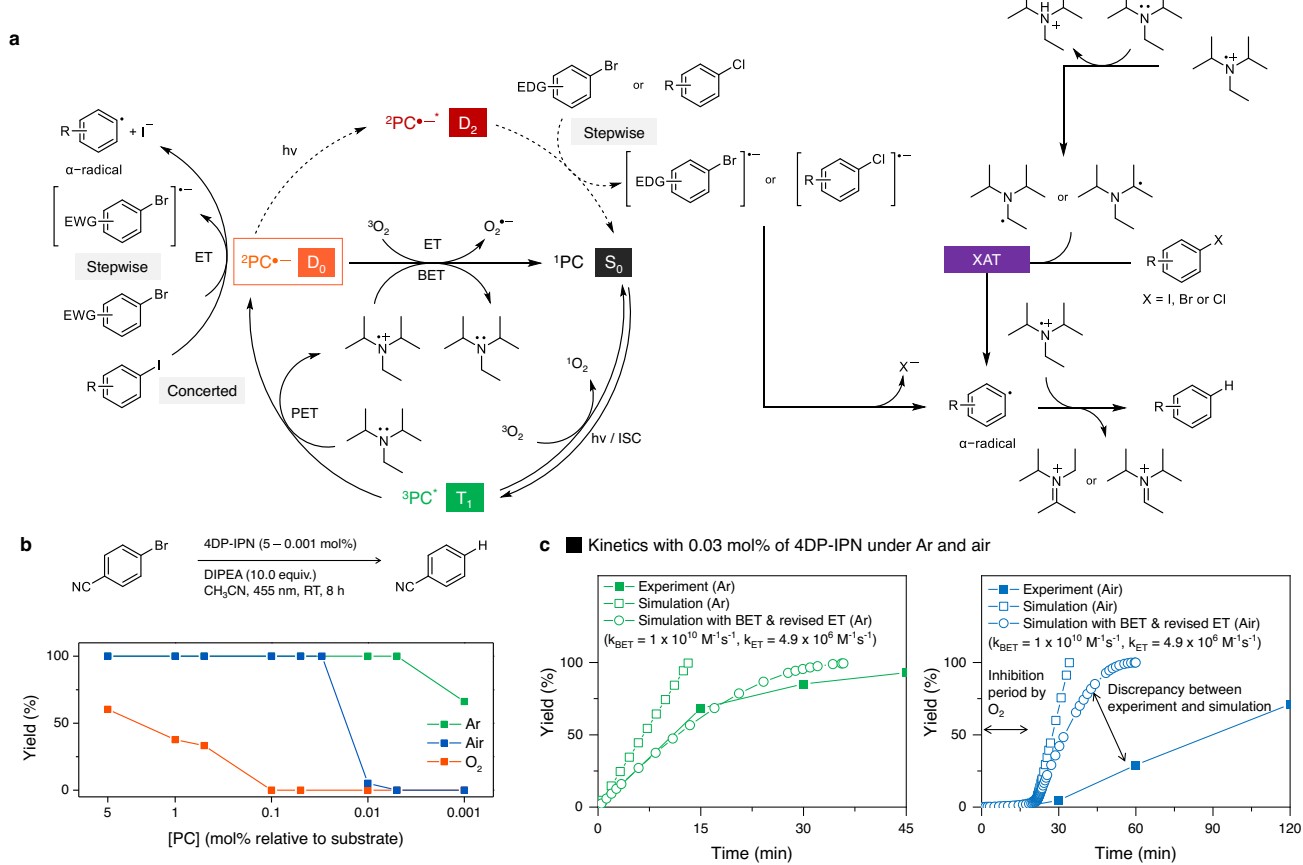

**Fig. 8 | Oxygen tolerance in the dehalogenation reactions. a** Proposed mechanism of the photocatalyzed reductive dehalogenation of aryl halides mediated by $^2PC^{•-}$. Here, (B)ET and XAT denote (back)electron transfer and halogen atom transfer, respectively. **b** Effect of oxygen for the reductive dehalogenation of 4-bromobenzonitrile with 4DP-IPN as a PC in the presence of Ar (green-filled squares), air (blue-filled squares), and $O_2$ (orange-filled squares). Reactions were performed with 4-bromobenzonitrile (0.1 M), DIPEA (10.0 equiv.), and 4DP-IPN (5−0.001 mol%) in $CH_3CN$ under the illumination of two 3 W 455 nm LEDs for 8 h. **c** Experimental (filled squares) and simulated (empty squares) reaction kinetics of

photoredox-mediated reductive dehalogenation of 4-bromobenzonitrile with 4DP-IPN (0.03 mol%) under Ar and air atmospheres in a closed glass vial; simulated (empty circles) reaction kinetics indicate that the BET process ($k_{BET} = 1 \times 10^{10}$ M$^{-1}$ s$^{-1}$) and a slower ET process ($k_{PET} = 4.9 \times 10^6$ M$^{-1}$ s$^{-1}$) are involved. Kinetics simulations were performed based on the proposed mechanism described in Supplementary Fig. 15. The rate constants for all processes were evaluated from experiments or calculations (see Supplementary Fig. 15 and Supplementary Table 7 for the full details of the kinetics simulation).

different electronic and steric properties and (ii) the photo-degradation of 4DP-IPN in actual dehalogenation reactions with different substrates, providing significant insights into the pho-toredox catalysis carried out by cyanoarene-based PCs. Based on the in-depth analysis of photodegradation behaviors, we also demonstrated highly efficient reductive dehalogenation, indicat-ing that it is possible to significantly reduce PC loading when photodegradation processes are not involved. We believe that our work will provide a better understanding of the formation, degradation, and photochemical/electrochemical features of PC$^{•-}$. Moreover, our results pave the way to in-depth knowledge of the reductive cycles and the stability of PCs in purely organic PC-driven photoredox catalysis, which can be applied to address many challenging issues in a variety of photoredox-mediated organic reactions and polymerizations.

## Methods
### General procedure for generation of PC$^{•-}$
Inside the glove box, a sealable quartz cuvette was charged with PC ($1.0 \times 10^{-4}$ M) and DIPEA (0.5 M) in 3 mL of anhydrous $CH_3CN$. After that, the quartz cuvette was capped with a screw cap and sealed with

parafilm. Subsequently, the reaction was carried out under the illu-mination of two 3 W 455 or 515 nm LEDs for 1 min at RT. After illumi-nation, UV-Vis absorption spectra of the illuminated solutions were measured immediately. In preparation for the reaction, pre-prepared stock solutions of the PCs were used for higher reproducibility of results.

### General procedure for PC photodegradation
Outside the glove box, a 20 mL glass vial equipped with a stirring bar was charged with PC ($1.0 \times 10^{-4}$ M) and DIPEA (0.5 M) in 1 mL of anhydrous $CH_3CN$ as a solvent. Then, the vial was capped with a rubber septum or screw cap and sealed with parafilm. The reaction batches were purged with $N_2$ (99.999%) for 30 min. Subsequently, the reaction was carried out for 2 h under the illumination of two 3 W 455 nm LEDs at RT. Without the work-up process, the aliquots of the crude reaction mixture were diluted by $CH_2Cl_2$ and monitored by TLC. In preparation for the reaction, pre-prepared stock solutions of the PCs were used for higher reproducibility of results. For the scale-up reaction of photo-degradation, a 20 mL glass vial equipped with a stirring bar was charged with PC ($1.0 \times 10^{-2}$ M) and DIPEA (1 M) in 5−6 mL of anhydrous $CH_3CN$ as a solvent. Then, the vial was capped with a rubber septum or

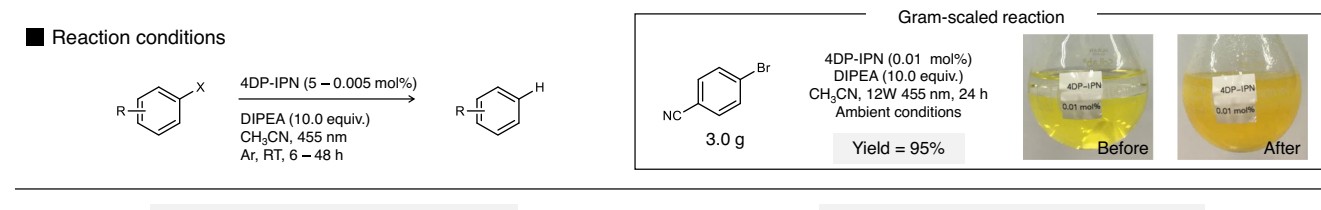

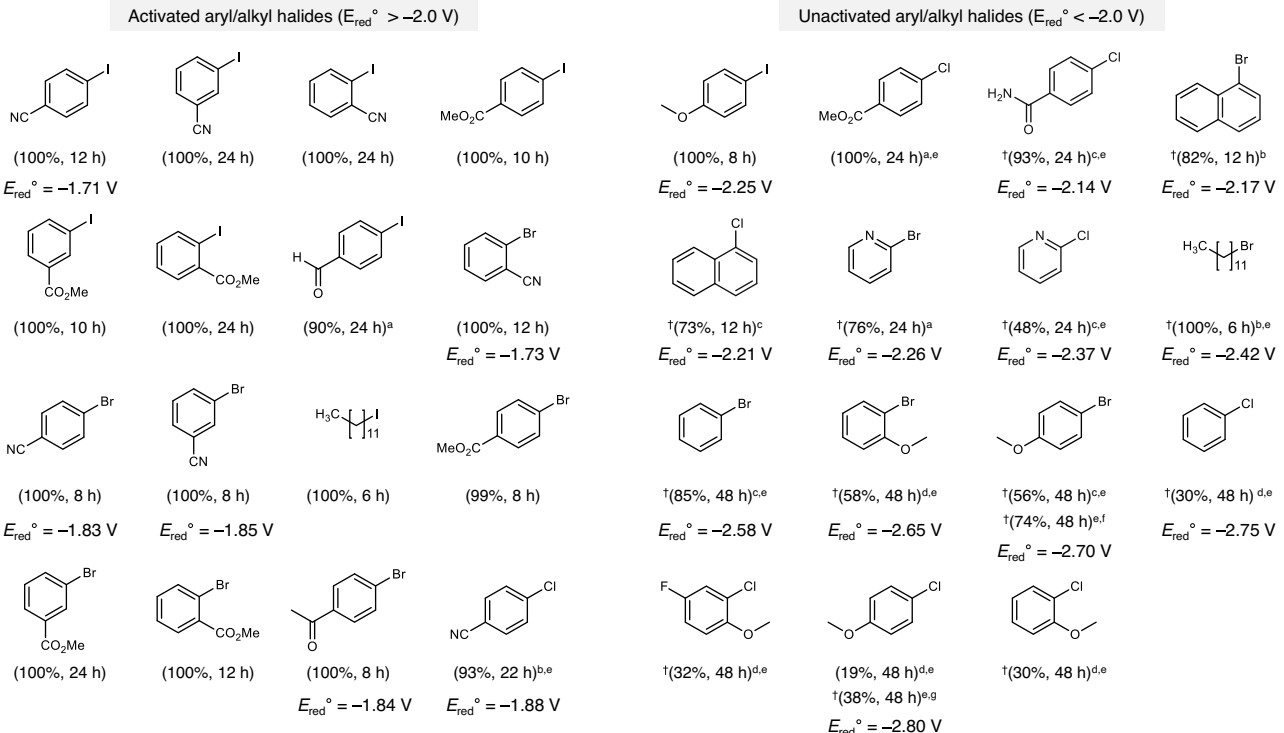

**Fig. 9 | Results of reductive dehalogenation of various aryl/alkyl halides in the presence of 4DP-IPN as a PC.** Reactions were performed with substrates (0.1 M), DIPEA (10.0 equiv.), and 4DP-IPN (0.005 mol%) in CH₃CN (1 mL) under the illumination of two 3 W 455 nm LEDs for 6 to 48 h at RT. The gram-scaled reaction was performed with 4-bromobenzonitrile (3.0 g, 16.48 mmol), DIPEA (10.0 equiv.), and 4DP-IPN (0.01 mol%) in CH₃CN (30 mL) with irradiation by four 3 W 455 nm LEDs under ambient conditions without any degassing. Reactions were performed with

ᵃ0.01 mol%, ᵇ0.05 mol%, ᶜ0.5 mol%, and ᵈ5 mol% 4DP-IPN, and ᵉillumination by four 3 W 455 nm LEDs. Injection of a total of ᶠ1.5 mol% and ᵍ3 mol% 4DP-IPN divided over three additions every 16 h during the course of the reaction. Yields were determined by GC-FID or ᵗᴴH NMR using 1,3,5-trimethoxybenzene as an internal standard. All redox potential values were obtained from the literature, where the potential values were measured against the SCE[72–77].

screw cap and sealed with parafilm. Subsequently, the reaction was carried out for 2 h under the illumination of four 3 W 455 nm LEDs at RT without degassing process[53]. Afterward, the reaction mixture was evaporated under low pressure and the concentrated crude products were further purified by column chromatography on silica gel.

**General procedure for photoredox reductive dehalogenation**
Inside the glove box, a 20 mL glass vial equipped with a stirring bar was charged with aryl halides (0.1 mmol), DIPEA (174 µL, 1 mmol), PC (5–0.001 mol% to relative aryl halides), 1,3,5-trimethoxybenzene (33.6 mg, 0.2 mmol) as an internal standard for GC-FID and ¹H NMR, and anhydrous CH₃CN (1 mL, 0.1 M of aryl halides) as solvent. After, the vial was capped with a rubber septum or screw cap and sealed with parafilm. The reaction batches were purged with Ar (99.9999%) (or with air or with O₂ (99.995%)) for 30 min outside the glove box. Subsequently, the reaction was carried out for hours under the illumination of two 3 W 455 nm LEDs at RT. Without a work-up process, the aliquots of the crude reaction mixture were analyzed by GC-FID or ¹H NMR to obtain yields of dehalogenated products. In preparation for the reaction, pre-prepared stock solutions of the PCs were used for the higher reproducibility of results.

## Data availability
The authors declare that the data supporting the findings of this study are available within the paper and its Supplementary Information.

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

## Acknowledgements

We thank Prof. Soo Young Park, Prof. Ki Tae Nam, and Prof. Chulbom Lee for sharing instruments. This work was supported by the TJ Park Science Fellowship of POSCO TJ Park Foundation and by the Basic Science Research Program through the National Research Foundation of Korea (NRF), which was funded by the Ministry of Science and ICT (MSIT) (2021R1A5A1030054 and 2022R1A2C2011627). The work at IMDEA was supported by the Spanish Ministry for Science (MINECO/MICINN–FEDER projects CTQ2017–87054 and RTI2018-097508-B-I00) by the "Severo Ochoa" program for Centers of Excellence in Research and Development (MINECO grant SEV–2016–0686 and MICINN grant CEX2020-001039-S), European Union structural funds and Comunidad de Madrid NMAT2D-CM Program (P2018/NMT-4511), and Campus of International Excellence UAM + CSIC.

## Author contributions

Y.K. and M.S.K. were responsible for the initial conception of the project. Y.K., J.G., R.W., and M.S.K were involved in the discussion of the photophysics and photochemical reactions and were responsible for writing and editing the final manuscript. Y.K. performed most of the experiments and wrote the initial draft of the manuscript. J. L., Y.L., and C.Y. assisted in the synthesis of cyanoarene compounds. Y.N. and D.K. assisted in the dehalogenation reaction. S.F. and R.W. performed CW and gated photoluminescence measurements at variable temperatures and analyzed the data. Y.K. performed DFT calculations supported by J.C.R. Y.K. performed the kinetic simulation under supervision from R.W. and M.S.K. All the authors discussed the results and commented on the manuscript. J.G., R.W., and M.S.K. supervised the project.

## Competing interests

The authors declare no competing interests.

## Additional information

[1]Department of Materials Science and Engineering, Seoul National University, Seoul 08826, Republic of Korea. [2]Department of Materials Science and Engineering, Ulsan National Institute of Science and Technology (UNIST), Ulsan 44919, Republic of Korea. [3]Madrid Institute for Advanced Studies, IMDEA Nanoscience, Calle Faraday 9, Campus Cantoblanco, Madrid 28049, Spain. [4]Donostia International Physics Center (DIPC), Manuel Lardizabal Ibilbidea 4, San Sebastián 20018, Spain. ✉e-mail: johannes.gierschner@imdea.org; reinhold.wannemacher@imdea.org; minsang@snu.ac.kr

