## [Peer Review File · Nature Communications]

Reviewers' comments:

Reviewer #1 (Remarks to the Author):

In this manuscript Kwon and coworkers report a strategy for the generation of photocatalytically active radical anions using alkyl amine reductants. They examine the effectiveness of this system in aryl halide reductions. They perform several photophysical experiments to characterize the optimal catalyst, 4DP-IPN. Furthermore, the authors offer a well-rationalized explanation for the unique effectiveness of 4DP-IPN as a photoreductant, namely its highly twisted donor-acceptor structure leads to high triplet yield and that there is a higher barrier to back electron compared to heavy metal photocatalysts due to a lack of spin-orbit coupling. However, while the results presented in this manuscript are interesting, they are not unexpected based on related work that has appeared in the past year that is uncited in the manuscript. Two recent papers from Wickens regarding very similar photoreduction behavior of 4DP-IPN. In the first paper, *Angew. Chem. Int. Ed.* 2021, 60, 21418, the radical anion of 4DP-IPN was electrochemically generated and demonstrated to catalytically photoreduce aryl electrophiles, including ones significantly more challenging to reduce than those presented in this manuscript. The second publication *J. Am. Chem. Soc.* 2021, 143, 29, 10882–10889, demonstrated conPET behavior of 4DP-IPN using formate and alkylamine reductants. Additionally, Wu, *J. Am. Chem. Soc.* 2021, 143, 33, 13266–13273, has also achieved challenging aryl electrophile reductions using a structurally similar photocatalyst to 4DP-IPN. In regard to synthetic utility, all three of these reports illustrated coupling reactions and dehalogenations (as opposed to just dehalogenation, as shown in this manuscript) and a broader substrate scope. In addition to including a discussion of these related publications in the main text, we also recommend that Adachi, *Nature* 492, 234–238 (2012), be credited for the design principle of 4DP-IPN and analogs, as well as Miyake, *J. Phys. Chem. A* 2020, 124, 5, 817–823, for using a similar design principle in a different class of photocatalysts. Finally, I have to point out that the substrates reduced in this manuscript can be reduced using conventional single photon reductants, such as N-phenyl phenothiazine (*Chem. Commun.* 2015, 51, 11705-11708).

The observation that radical anions of photocatalysts can act as strong photoreductants is an important one. However, this has precedent in the literature, including with the same catalyst but with a broader scope. Overall, even once the concerns regarding the literature cited are addressed, I assess this manuscript as too premature for publication without an extensive expansion of either its synthetic or the mechanistic aspects relative to the current state of the art in the field.

Reviewer #2 (Remarks to the Author):

The manuscript from Kwon et al. reported that a photocatalyst (PC), 4DP-IPN, can catalyze the dehalogenation of aryl halides with a loading as low as 0.001%. This extraordinary catalytic activity was proposed to come from a long-lived triplet excited state of this PC. Specifically, this allows the generation of the corresponding PC radical anion in its D2 state, which has a highly negative oxidation potential of -3.38 eV. Overall, I feel that this exciting and solid work is publishable in Nature Communications.

From the perspective of a computational chemist, I have a couple of questions about the DFT/TDDFT modelings:

- In the Jablonski diagram in Figure 2b), is there any clear evidence that S1 and T1 states from TDDFT calculations are involved in ISC and RISC? The stick spectra in Figure 2c and S9b show that S2 and S3 are only slightly higher in energy. An evaluation of spin-orbital couplings between S_m and T_n calculations might help identify the pairs of singlet and triplet states with the strongest coupling.

- As shown in Figure S9c, "D0->D2", "D0->D3" and all higher TD-B3LYP excitations of the PC radical anion have substantial charge transfer character. I would caution against using only the B3LYP functional in these calculations. I would also try M06-2X or wB97X-D functionals at the same time, which will increase the energy of these charge transfer excitations and reduce the predicted wavelengths. Also, the x-axis range of the computed absorption spectra in Fig. S9c for the PC radical anion can potentially be extended in order to show the two excitations (with energies of 0.85 and 1.91 eV) that were suggested to play a key role in the catalytic mechanism.

Reviewer #3 (Remarks to the Author):

These comments concern the communication "Maximizing the formation of strongly reducing radical anions for highly efficient purely organic photoredox catalysis and multiphoton excitation catalysis" from the groups of Gierschner, Wannemacher, and Kwon. I have found this work to be both exciting as well as lacking in some ways. What is almost certainly (see specific comment about controls) strong is the discovery by these researchers of a potent catalytic system for a challenging organic transformation. As to the meaning of potent, I would point to the very high yields at very low catalyst concentration in reactions that are also completed in reasonable times (of order several hours). As well, it is argued that the catalyst can adapt to its needs in aryl-halide reductive dehalogenations. In other words, the reactivity of different aryl halides is reported to be catalyzed in different ways. When the reduction potential of an aryl halide is lowered by electron withdrawing groups, the purported mechanism follows from thermal chemistry of the photo-produced catalyst radical anion engaging, first, in electron transfer with Ar-X. But when this is not possible

thermodynamically (in aryl-chlorides or in deactivated aryl halides), it is argued that the photocatalyst radical anion can absorb photons to become a more potent reductant. Thus, there is potentially considerable synthetic flexibility afforded by the PC system and it could be highly valued by the synthetic community and industries relying on such chemistry. The editor should certainly weigh the opinion of evaluators who have strong synthetic experience, but this is my sense for what it is worth.

What I find somewhat lacking are mechanistic details and dealing with important loose ends such that this work can stand on its own. While what is written below is critical, I do acknowledge the many valuable things these researchers report including the informative kinetic modeling that explains the O₂ behavior, the PC photophysical properties, quenching data, and the nice TD-DFT analysis of chromophore and radical anion properties.

The observation that troubles me the most is that the PC is unstable in the presence of the sacrificial reductant alone when irradiated for longer than one minute (reactions are run for hours). The authors argue that this is an issue only when there is not aryl halide present. (I don't really have a sense of what the degradation pathway would be). Even if the reactivity of the aryl radical with the amine radical cation (the capping pathway) is fast, it would be the case that as the reaction proceeds towards 100%, the amount of aryl halide present becomes limited and a diffusion limited bimolecular reaction would be slowed. Thus it is hard to envision a situation where the PC survives the reaction as has been claimed by the authors. Since it is hard to imagine the survival during long reactions, it brings up the critical question as to whether it, versus a byproduct, is the active catalyst. The authors could do more to support their mechanistic idea. I think that Figure S7 (top pathway) could be helpful in that it suggests that the catalyst is regenerated when excess aryl halide is present. If the authors could experimentally connect a sequence of such events (1. addition of aryl halide to the PC radical anion; 2. Light back on until the aryl halide is consumed and the PC radical anion is re-formed) until there is spectroscopic (NMR) or TLC evidence for product – and not just regeneration of the catalyst – it would go a long way towards convincing the reader of the purported simple mechanism being driven by the PC radical anion alone.

Another issue I have has to do with the novelty of the PC design. I am concerned that photophysical attributes being discussed as important may not be meaningful or maybe not even be present. Before proceeding I would say that it is clear that the triplet lifetimes are an excellent attribute of these PC systems and are very likely to be important. For example, they enable electron transfer (sacrificial amine to PC*) even in situations where the driving force is zero or even positive such as with the three sacrificial reductants (DIPEA, TBA, and TEA) explored herein. For example, with DIPEA concentrations of 0.1 M, electron transfer to the PC* is still taking 10 microseconds. In other words, the initial electron transfer is slow (not at all fast as the Fig. 1d suggests) and the long lifetime of the triplet saves the day. In terms of the more speculative issues that I think are unlikely to impact mechanism I would point to the discussion about spin states retarding back electron transfer rates (Figure 1d again). In the radical pair it is hard to imagine much of an interaction between the spin ½

systems on the oxidized amine donor and the reduced PC acceptor. Even if they are born coupled (on a 10 microsecond time scale as discussed above) it is unlikely that they would remain that way for long given the weak exchange coupling and the geometric fluctuations of the ion pair in solvent. It seems to me that the issues saving the day are (a) the substantial driving force for back electron transfer (-2.3 eV based on the reduction potentials of the PC and DIPEA) which would make this firmly in the inverted region and (b) the speed of H-atom abstraction to cement the DIPEA radical cation as an iminium salt. To this latter point, these researchers should be discussing the importance of this aspect of the reaction (its barely mentioned in the modeling in the SI). To me it seems that this chemistry of the sacrificial reductant (and the driving forces of the overall reaction) is essential for high yield given that the initial photoinduced electron transfer from DIPEA to PC* is thermodynamically uphill and given the reduction of the aryl halides are also thermodynamically uphill (as mentioned in the SI).

A third issue of concern to me involves the discussions of ConPET. In the text (line 269) it is stated that addition of bromo anisole to a solution of the PC radical anion results in no color change. This seems to contradict what is seen in Fig S7. Is the reader to draw a distinction from the shades of yellow produced when the bromoanisole is added? To me it looks from the photos more like PC than PC- and if it is the former then the ConPET is not needed. I am not intending to nit-pick but the issues of mechanism are made to sound simpler than they appear and that detracts from the impact of the work.

Specific issues:

On line 161 it is stated that the PC radical anion once formed is stable for a week in the absence of O₂. This seems impossibly long for a system with -2.3 eV of driving force for back electron transfer to the oxidized DIPEA. Did the authors generate the PC radical anion chemically in this case? If not and DIPEA was used, then there must be something going on besides capping (there is no aryl halide present in this example) and that could be very important in the mechanism.

Were any controls explored for yields? For example measuring yields in the absence of PC or in the absence of sac reductant? This seems standard in the PC field.

Is it possible to use EPR to confirm generation of the PC radical anions? Or does the degradation issues preclude generation of the needed concentrations? Can the PC radical anions be produced chemically allowing correlations to be made between EPR and UV-Vis?

Reviewers' comments to author:

We are grateful to the reviewers for their constructive comments and suggestions, which greatly helped to clarify a number of important points, to avoid misunderstandings, and to improve clarity and referencing of the paper. The reports of the three reviewers were overall very positive on the significance and novelty, stressing the suitability of our work for publication in *Nature Communications*. We have carefully revised our manuscript in line with all the reviewers' comments. The point-by-point response is given below.

Reviewer #1 (Remarks to the Author):

In this manuscript Kwon and coworkers report a strategy for the generation of photocatalytically active radical anions using alkyl amine reductants. They examine the effectiveness of this system in aryl halide reductions. They perform several photophysical experiments to characterize the optimal catalyst, 4DP-IPN. Furthermore, the authors offer a well-rationalized explanation for the unique effectiveness of 4DP-IPN as a photoreductant, namely its highly twisted donor-acceptor structure leads to high triplet yield and that there is a higher barrier to back electron compared to heavy metal photocatalysts due to a lack of spin-orbit coupling.

Response: We thank the reviewer for his encouraging comments and constructive suggestions. Our responses to reviewer #1's comments are highlighted in *blue color* in the revised manuscript and SI.

However, while the results presented in this manuscript are interesting, they are not unexpected based on related work that has appeared in the past year that is uncited in the manuscript. Two recent papers from Wickens regarding very similar photoreduction behavior of 4DP-IPN. In the first paper, *Angew. Chem. Int. Ed.* 2021, 60, 21418, the radical anion of 4DP-IPN was electrochemically generated and demonstrated to catalytically photoreduce aryl electrophiles, including ones significantly more challenging to reduce than those presented in this manuscript. The second publication *J. Am. Chem. Soc.* 2021, 143, 29, 10882–10889, demonstrated conPET behavior of 4DP-IPN using formate and alkylamine reductants. Additionally, Wu, *J. Am. Chem. Soc.* 2021, 143, 33, 13266–13273, has also achieved challenging aryl electrophile reductions using a structurally similar photocatalyst to 4DP-IPN. In regard to synthetic utility, all three of these reports illustrated coupling reactions and dehalogenations (as opposed to just dehalogenation, as shown in this manuscript) and a broader substrate scope. In addition to including a discussion of these related publications in the main text, we also recommend that Adachi, *Nature* 492, 234–238 (2012), be credited for the design principle of 4DP-IPN and analogs, as well as Miyake, *J. Phys. Chem. A* 2020, 124, 5, 817–823, for using a similar design principle in a different class of photocatalysts. Finally, I have to point out that the substrates reduced in this manuscript can be reduced using conventional single photon reductants, such as N-phenyl phenothiazine (*Chem. Commun.* 2015, 51, 11705-11708).

Response: Thanks for the reviewer's comments, which definitely help us to prevent missing the important previous works conducted by other groups within the community of photocatalysis research. Following the reviewer's suggestion, all aforementioned references have properly been included in the revised manuscript. In fact, the papers by Wickens' group and Wu's group have recently been published very close to the time when we first submitted this paper to another journal (2nd August, 2021), which ultimately caused our failure to include those references. We apologize for our mistake and now properly cite those papers in the revised manuscript. For why we consider our findings still novel, despite the

work of Wickens et al. and Wu et al., see below our following responses.

As the referee might recognize, the absorption of N-phenyl phenothiazine is limited to wavelengths below 400 nm. Therefore, in the paper of de Alanez's group (*Chem. Commun.* 11705 (2015)), authors used 380 nm LEDs as an excitation light source to proceed the dehalogenation reactions. Since, in our current work, we focus on the visible-light-driven photocatalysis, we do not discuss this reference in detail. In the revised manuscript, a brief comment on N-phenyl phenothiazine was included to prevent the confusion of the readers.

The observation that radical anions of photocatalysts can act as strong photoreductants is an important one. However, this has precedent in the literature, including with the same catalyst but with a broader scope. Overall, even once the concerns regarding the literature cited are addressed, I assess this manuscript as too premature for publication without an extensive expansion of either its synthetic or the mechanistic aspects relative to the current state of the art in the field.

Response: Taking into account the comments of the referee, we carefully reviewed the manuscript to clarify the novelty and importance of our work, which prompted us to prepare a major revision to the text as well as to conduct detailed additional experiments. In the current photocatalysis research, a profound understanding of the features of a PC radical anion and its excited states is greatly in demand to expand the scope of conventional photocatalysis. This includes understanding the entire process by which radical anions are generated, involved in (photo)chemical reactions, and degraded. However, despite the recent efforts by the groups of König, Nicewicz, Wickens, and Miyake, detailed understanding is still largely lacking, which results in many misinterpretations of the photocatalytic phenomena.

Figure XI. (a) Chemical structures of 3CzEPAIPN and the lifetime of 3CzEPAIPN⁻ reported by Wu's group (*J. Am. Chem. Soc.* 13266 (2021)) The structure shown on the right shows the degradation product 3CzEPAIPN inferred from our experiments. (b) Absorbance profiles for 3CzEPAIPN and 3CzEPAIPN⁻ (3.3×10^{-5} M). (c) Emission profiles (excitation at 390 nm) for 3CzEPAIPN and 3CzEPAIPN⁻ (3.3×10^{-5} M, 1.0 cm path length). (d) A color change from yellow to light yellow was observed before and after light irradiation with the solution of 3CzEPAIPN (3.2×10^{-5} M) and DIPEA after irradiation with 456 nm kessil LED (2×40 W). From Figure S10 of the quoted publication. (e) A color change of the solution of 3CzEPAIPN (3.2×10^{-5} M) and DIPEA (3.1×10^{-1} M) after irradiation with 456 nm kessil LED (2×40 W). All content of **Figure XI** is reproduced from the cited publication, except the structure of the methylated adduct shown on the right-hand side of **Figure XIa**.

For example, very recently, Wu's group reported a new dicyanoarene-based PC (*i.e.* 3CzEPAIPN, **Figure X1a**) with a long-lived excited state of its radical anion that facilitates the challenging C-X bond activation through a highly efficient ConPET process (*J. Am. Chem. Soc.* 13266 (2021)). However, considering our experience concerning radical anion intermediates of a variety of cyanoarene-based PCs, the interpretation of Wu's group might be wrong. This is clearly supported by the fact that 456 nm LED irradiation of a mixture of acetonitrile solutions of "3CzEPAIPN + DIPEA" differing only in the concentration of 3CzEPAIPN (3.2×10^{-5} M and 1.42×10^{-2} M, respectively) gave substantially different color response (**Figures X1d, e**). Mixture solutions with lower 3CzEPAIPN concentration (3.2×10^{-5} M) turn its color from yellow to pale yellow (blue-shift in UV/Vis spectrum) upon 456 nm LED irradiations whereas solutions with higher 3CzEPAIPN concentration (1.42×10^{-2} M) change its color from yellow to brownish-yellow (red-shift in UV/Vis spectrum).

Figure X2. (a) Reaction scheme of photobleaching reaction of 4DP-IPN. Reactions were performed with 4DP-IPN (1.0×10^{-4} M) and DIPEA (0.5 M) in CH₃CN under two 3W 515 nm LEDs or two 3W 455 nm LEDs at RT. In-situ TLC was monitored using CH₂Cl₂:hexane (7:3 v/v) mixture. ¹H NMR data of 4DP-IPN and 4DP-Me-BN was measured in CDCl₃ at RT. (b) Time-resolved photoluminescence (PL) spectra of 4DP-IPN (1.0×10^{-5} M; orange line) and 4DP-Me-BN (1.0×10^{-5} M; green line) in a CH₃CN solution at RT. (c) UV/Vis absorption spectra of 4DP-IPN (1.0×10^{-5} M; orange line) and 4DP-Me-BN (1.0×10^{-5} M; green line) in a CH₃CN solution at RT. (d) Steady-state PL spectra of 4DP-IPN (1.0×10^{-5} M; orange line) and 4DP-Me-BN (1.0×10^{-5} M; green line) in a CH₃CN solution at RT. (e) Stern-Volmer plots for the PL quenching of 4DP-Me-BN (1.0×10^{-5} M) in CH₃CN by DIPEA at RT. Stern-Volmer plots were obtained from PL decays of 4DP-Me-BN, excitation at $\lambda_{ex} = 377$ nm, detection at $\lambda_{det} = 470$ nm.

As we described in the manuscript (in the revised manuscript more in detail), all the cyanoarene-based PCs showed distinct red-shift in their UV/Vis spectra upon light irradiation in the presence of DIPEA, which is in accordance with our QC calculations. We thus speculate that the blue shift in UV/Vis absorption and PL emission observed by Wu's group (**Figures X1b, c**) arises not from its radical anion (3CzEPAIPN⁻), but from its degradation adduct (shown on the right of **Figure X1a**). This hypothesis was confirmed by our in-detailed studies about the degradation of 4DP-IPN.

As can be seen in **Figures X2**, in our additional experiments, the degradation pathway of 4DP-IPN was clearly revealed. Isolation of the degradation product by column chromatography (**Figure X2a**, center figure) followed by intensive NMR analyses (¹H NMR, COSY, and NOESY, **Figure X2a**, right-hand figure) unequivocally confirmed that a methyl substitution reaction occurred at the CN position of 4DP-IPN to generate 4DP-Me-BN (**Figure X2a**). The degradation

adduct, 4DP-Me-BN, showed clear blue-shift in UV/Vis absorption and PL emission (**Figures X2b-e**), which is very similar to the observation made by Wu's group. Given that 3CzEPAIPN has similar structure with 4DP-IPN, Wu's group might observe 3CzEPAMeBN, not 3CzEPAIPN⁻. We believe that this example shows the deficient understanding of photophysical processes involved in the features of a radical anion intermediate including its generation, degradation, photophysical/electrochemical features. This insufficient understanding about a PC radical anion results in misinterpretation of chemical reactions, which might delay the development of this exciting field. Moreover, any identification of a Con-PET process should demonstrate a superlinear dependence on the excitation intensity at equal dose, accompanied by proper modeling of the populations of ³PC and PC⁻.

Moreover, through the additional detailed experiments and analyses, we noticed that the chemistry of the sacrificial reductant as well as the degradation of PC intermediates is essential for the generation of long-lived radical anion intermediates (PC⁻) that are a basis for successful dehalogenation reactions of various aryl/alkyl halides. In this context, we realized that the difficulty in the generation of a highly reducing radical anion arises from the difficulty to find out a PC with both i) a decent initial PET with a sacrificial reductant, ii) a highly negative reduction potential, at the same time, and iii) a visible-light absorption. Here, a PC with a highly efficient generation of long-lived triplet excitons provides a solution. Since abundant generation of long-lived triplet states maximizes the concentration of excited state PC and hence, facilitates thermodynamically unfavorable initial PET processes, the PC can now have all those three properties (see **Figure X3**). This argument is in complete disagreement with the publication of Wu's group (for a very similar PC), where they state that in their case TADF, and, hence, the triplet is quenched. The authors of this publication do not offer a convincing explanation on how the radical anion is formed in that case.

Moreover, since electron transfer between the triplet excited state of PC (³PC*) and a sacrificial electron donor produces a triplet contact radical ion pair (TCRIP), the BET process should further be retarded, which helps the generation of radical anions. Taking all this into account, we significantly revised the manuscript with three additional Figures. With the help of the referees, we now provide very nice mechanistic insight of a radical anion-mediated photoredox catalysis, which will be very helpful for the whole field of organic photocatalysis.

Figure X3. Schematic description of photoinduced electron transfer (PET) between a purely organic PC and a (sacrificial) reducing agent (Red) for a different spin state of the PC in the excited state, i.e. singlet (left-hand side) and triplet (right-hand side). Spin-multiplicities as well as electron configurations are indicated in each state of the PC. Enhanced PET in the case of ³PC* is due to the fact that [³PC*] >> [¹PC*] for the case of continuous illumination.

Furthermore, as the referee #3 suggested, we thoroughly investigated the photodegradation behavior of 4DP-IPN in an actual dehalogenation reaction of various aryl halides. As shown in (**Figure X4**), in the presence of aryl halides substrates, PC photodegradation was monitored by in-situ TLC with aliquots of reactions were taken at a given time associated

with the yields trace by $^1\text{H-NMR}$. The decomposition of the PC in the actual dehalogenation reaction was significantly retarded compared to that in the presence of DIPEA alone (see **Figure X4a: TLC at 455 nm**). This is likely because that the PC degradation competes with the electron transfer from $4\text{DP-IPN}^{\bullet-}$ to aryl halide substrates, as depicted in **Figure X4b**. In other words, in the presence of the aryl halides substrate, the electron transfer from $4\text{DP-IPN}^{\bullet-}$ to the substrate seems to be significantly faster than i) the formation of $4\text{DP-IPN}^{\bullet-}$ and ii) the photosubstitution reaction to form 4DP-Me-BN , which results in a substantial delay in the PC degradation.

Figure X4. (a) Fate of catalyst during dehalogenation reactions. Reactions were performed with substrates (0.1 M), DIPEA (10.0 equiv.) and 4DP-IPN (1 mol%, 1.0×10^{-3} M) in CH_3CN (1 ml) under illumination by two 3W 455 nm LED at RT for 1.5 or 12 hours. Yields were determined by $^1\text{H NMR}$ using 1,3,5-trimethoxybenzene as an internal standard. In-situ TLC was monitored using EA:hexane (1:4 v/v) mixture. (b) Proposed mechanistic pathway for photobleaching reaction in the presence of aryl halides.

To prove our hypothesis, the substrates with various reduction potentials were screened in which the electron transfer rate would be varied (**Figure X4: Reaction A, B and C**). As shown in **Figure X4a: Reaction A**, 100% yield was obtained for 4-bromobenzonitrile ($E_{\text{red}}^\circ = -1.83 \text{ V}$) in 1.5 h at which the trace amount of photodegradation product, *i.e.* 4DP-Me-BN, was observed, implying that 4DP-IPN is an active PC for the dehalogenation reaction of 4-bromobenzonitrile. However, in fact, more PC degradation was observed in the dehalogenation of more challenging substrates with stronger C-X bond strength and more negative reduction potential (*i.e.* 4-chlorobenzonitrile ($E_{\text{red}}^\circ = -1.88 \text{ V}$) and 4-bromoanisole ($E_{\text{red}}^\circ = -2.70 \text{ V}$), see **Figure X4a: Reactions B, C**), further supporting our hypothesis. In **Figure X4a: Reaction B**, 100% yield of dehalogenation of 4-chlorobenzonitrile was obtained in 12 h and a somewhat larger amount of 4DP-Me-BN

was observed after this time. This is probably because, due to the strong C-Cl bond, the dehalogenation of the 4-chlorobenzonitrile radical anion was retarded and the associated back electron transfer from 4-chlorobenzonitrile radical anion to the PC enhanced degradation of the PC. In the case of 4-bromoanisole, the slow rate of electron transfer from 4DP-IPN⁻ to the 4-bromoanisole compared to other substrates and the presence of C-Br bond resulted in more PC degradation. More interestingly, however, even after 10.5 h of additional continuous irradiation after the completion of the dehalogenation reaction of 4-bromobenzonitrile, only small amount of the PC degradation adduct could be observed, suggesting that the PC degradation is still retarded in the absence of 4-bromobenzonitrile (see **Figure X4a: Reaction A, TLC**). This might be due to the fact that the reaction product, benzonitrile ($E_{red}^{\circ} = -2.40$ V, *Synlett* 2016, 27, 714), can act as an electron acceptor instead of 4-bromobenzonitrile that competes with the PC degradation process.

All these points have now properly been included in the revised version of the manuscript and SI. We believe that our intensive additional investigations during the revision process have produced essential additional information about the mechanistic aspects and, in this way we have progressed significantly beyond the present state of the art in this field. Therefore, we hope that the manuscript can be reconsidered for publication in *Nature Communications*.

Reviewer #2 (Remarks to the Author):

The manuscript from Kwon et al. reported that a photocatalyst (PC), 4DP-IPN, can catalyze the dehalogenation of aryl halides with a loading as low as 0.001%. This extraordinary catalytic activity was proposed to come from a long-lived triplet excited state of this PC. Specifically, this allows the generation of the corresponding PC radical anion in its D2 state, which has a highly negative oxidation potential of -3.38 eV. Overall, I feel that this exciting and solid work is publishable in Nature Communications.

Response: Thank you for reviewer's very positive and constructive comments on the work. Our responses to reviewer #2's comments are highlighted in *blue color* in the revised manuscript and SI.

From the perspective of a computational chemist, I have a couple of questions about the DFT/TDDFT modelings:

In the Jablonski diagram in Figure 2b), is there any clear evidence that S₁ and T₁ states from TDDFT calculations are involved in ISC and RISC? The stick spectra in Figure 2c and S9b show that S₂ and S₃ are only slightly higher in energy. An evaluation of spin-orbital couplings between S_m and T_n calculations might help identify the pairs of singlet and triplet states with the strongest coupling.

Response: We thank the reviewer for the critical remarks on this matter. In fact, the Jablonski diagram is very simplified for the current paper, as it only represents the S₁ and T₁ states, which are the most relevant states with respect to the photocatalytic activity; in reality, the photophysics is more complicated. According to our TD-DFT calculations, ISC is hardly promoted by the T₁ state because the state description is very similar to S₁ and thus spin-orbit coupling is small. Instead, the main channel for ISC is T₂, and to some degree T₃. On one side this is due to the orthogonality of the electronic situation in T_{2,3} with respect to S₁; on the other hand, the energy differences between T₂ and T₃ with respect to S₁ are also small, which further promotes ISC (*Macromolecules* 2019, 52, 5538). We did calculate ISC and RISC by TD-DFT, which are of the same order as the experimental values. It is further remarked that we don't expect ISC from S₂ and S₃ as the lifetime of these states are expected to be very short because of ultrafast internal conversion and vibrational relaxation to S₁. In any case, the detailed discussion of all these issues is considered as out of scope for the current paper, and will be reported in a forthcoming paper.

Following the reviewer's thoughts, we now slightly changed the Jablonski scheme in *Figure 2b* of the manuscript, indicating the involvement of higher excited states. We further updated the given rates, which were determined from experimental data (*Macromolecules* 2019, 52, 5538). In fact, it turned out that the PL quantum yield is much higher than reported there; a correction of the aforesaid paper concerning this quantity (but not affecting any of the conclusions of that publication) is in preparation.

As shown in Figure S9c, "D0->D2", "D0->D3" and all higher TD-B3LYP excitations of the PC radical anion have substantial charge transfer character. I would caution against using only the B3LYP functional in these calculations. I would also try M06-2X or wB97X-D functionals at the same time, which will increase the energy of these charge transfer excitations and reduce the predicted wavelengths. Also, the x-axis range of the computed absorption spectra in Fig. S9c for the PC radical anion can potentially be extended in order to show the two excitations (with energies of 0.85 and 1.91 eV) that were suggested to play a key role in the catalytic mechanism.

Response: We agree with the reviewer with the general concern about selecting DFT functionals. In fact, we have done comparisons with different functionals (including CAMB3LYP, M06-2X and wB97XD), and already for the neutral species, we saw very different behaviors for absolute and relative energies of the S2 and Tn states (and in particular with respect to the singlet-triplet (ST) gap) as well as for the state descriptions. The B3LYP functional was chosen because of its good correspondence of the resulting ST gaps to experiment, as shown by us earlier (*Nat. Catal.* 2018, 794-804).

For the calculated radical anion absorption, the lowest excited state was actually calculated to be lower with CAMB3LYP and M06-2X in comparison with B3LYP, so that the latter method still appears quite reasonable to describe the electronic nature of ground and excited state properties for both the neutral and the radical anion species.

In order to assist the reader in this, a small discussion was now added in the SI. For the calculated spectrum, Fig. S10c already extends to 1500 nm (0.83 eV), and in the inset up to 1600 nm.

Reviewer #3 (Remarks to the Author):

These comments concern the communication “Maximizing the formation of strongly reducing radical anions for highly efficient purely organic photoredox catalysis and multiphoton excitation catalysis” from the groups of Gierschner, Wannemacher, and Kwon. I have found this work to be both exciting as well as lacking in some ways.

Response: Thank you for reviewer’s critical comments and very constructive suggestions. We believe that the reviewer made important comments to improve the quality of our work. Thus, based on the reviewer’s comments, we provided more detailed explanation on the presented data and conducted detailed additional experiments to address the questions raised by the reviewer. Our responses to reviewer #3’s comments are highlighted in *blue color* in the revised manuscript and SI.

What is almost certainly (see specific comment about controls) strong is the discovery by these researchers of a potent catalytic system for a challenging organic transformation. As to the meaning of potent, I would point to the very high yields at very low catalyst concentration in reactions that are also completed in reasonable times (of order several hours). As well, it is argued that the catalyst can adapt to its needs in aryl-halide reductive dehalogenations. In other words, the reactivity of different aryl halides is reported to be catalyzed in different ways. When the reduction potential of an aryl halide is lowered by electron withdrawing groups, the purported mechanism follows from thermal chemistry of the photo-produced catalyst radical anion engaging, first, in electron transfer with Ar-X. But when this is not possible thermodynamically (in aryl-chlorides or in deactivated aryl halides), it is argued that the photocatalyst radical anion can absorb photons to become a more potent reductant. Thus, there is potentially considerable synthetic flexibility afforded by the PC system and it could be highly valued by the synthetic community and industries relying on such chemistry. The editor should certainly weigh the opinion of evaluators who have strong synthetic experience, but this is my sense for what it is worth. What I find somewhat lacking are mechanistic details and dealing with important loose ends such that this work can stand on its own. While what is written below is critical, I do acknowledge the many valuable things these researchers report including the informative kinetic modeling that explains the O₂ behavior, the PC photophysical properties, quenching data, and the nice TD-DFT analysis of chromophore and radical anion properties.

Response: We are grateful for the reviewer’s positive and encouraging comments on our work.

The observation that troubles me the most is that the PC is unstable in the presence of the sacrificial reductant alone when irradiated for longer than one minute (reactions are run for hours). The authors argue that this is an issue only when there is not aryl halide present. (I don’t really have a sense of what the degradation pathway would be). Even if the reactivity of the aryl radical with the amine radical cation (the capping pathway) is fast, it would be the case that as the reaction proceeds towards 100%, the amount of aryl halide present becomes limited and a diffusion limited bimolecular reaction would be slowed. Thus it is hard to envision a situation where the PC survives the reaction as has been claimed by the authors. Since it is hard to imagine the survival during long reactions, it brings up the critical question as to whether it, versus a biproduct, is the active catalyst. The authors could do more to support their mechanistic idea. I think that Figure S7 (top pathway) could be helpful in that it suggests that the catalyst is regenerated when excess aryl halide is present. If the authors could experimentally connect a sequence of such events (1. addition

of aryl halide to the PC radical anion; 2. Light back on until the aryl halide is consumed and the PC radical anion is reformed) until there is spectroscopic (NMR) or TLC evidence for product – and not just regeneration of the catalyst – it would go a long way towards convincing the reader of the purported simple mechanism being driven by the PC radical anion alone.

Response:

Figure X5. (a) Photobleaching reaction of 4DP-IPN in the presence of DIPEA. Reactions were performed with 4DP-IPN (1.0×10^{-4} M) and DIPEA (0.5 M) in CH_3CN under the illumination of two 3W 515 nm LEDs or two 3W 455 nm LEDs at RT. PC degradations were monitored in-situ by TLC with eluent conditions, CH_2Cl_2 :hexane (7:3 v/v). The degradation product was successfully isolated by column chromatography, which results in the ^1H NMR spectrum shown in green line confirming that a methyl substitution reaction occurred at the CN position of 4DP-IPN to generate 4DP-Me-BN. (b) Proposed mechanistic pathways for photobleaching behavior of 4DP-IPN in the presence of DIPEA (upper part) and DFT calculations for each the most plausible reaction pathways (lower part).

Following the reviewer's comment, we first carefully monitored the degradation of 4DP-IPN in the sacrificial reductant (i.e. DIPEA) alone under both 455 nm and 515 nm LED irradiation conditions. Surprisingly, unlike most organic molecules that have very complex degradation patterns, 4DP-IPN was found to be degraded into one major product in

a similar way for both conditions, while the degradation rates at 455 nm and 515 nm were very different; degradation at 455 nm LED illuminations was much faster as compared to that at 515 nm (**Figure X5a, left**). The degradation product was successfully isolated by column chromatography. Detailed NMR analyses (^1H NMR, COSY, and NOESY) clearly confirmed that a methyl substitution reaction occurred at the CN position of 4DP-IPN to generate 4DP-Me-BN (**Figure X5a, right**).

Figure X6. (a) Fate of catalyst during dehalogenation reactions. Reactions were performed with substrates (0.1 M), DIPEA (10.0 equiv.) and 4DP-IPN (1 mol%, 1.0×10^{-3} M) in CH_3CN (1 ml) under illumination by two 3W 455 nm LED at RT for 1.5 or 12 hours. Yields were determined by ^1H NMR using 1,3,5-trimethoxybenzene as an internal standard. In-situ TLC was monitored using EA:hexane (1:4 v/v) mixture. (b) Catalytic performance of 4DP-Me-BN. Reactions were performed using the same conditions as for (a). Early reaction kinetics with 4DP-IPN and 4DP-Me-BN were monitored on a 2 ml scale, yields determined by ^1H NMR using 1,3,5-trimethoxybenzene as an internal standard. (c) Proposed mechanistic

A control experiment combined with quantum chemical (QC) calculations was performed to investigate the mechanistic pathway of 4DP-IPN degradation at the given conditions. Interestingly, a completely different degradation pattern was observed in the absence of DIPEA, suggesting that DIPEA plays an important role in the photodegradation process of 4DP-IPN. Therefore, we assumed that the long-lived radical anion intermediates of 4DP-IPN are first formed in the presence of DIPEA followed by the substitution reaction of the PC radical anion by CH_3^\bullet that might be generated from the solvent, CH_3CN . To validate our hypothesis, the most possible reaction pathway was screened by DFT calculations (**Figure X5b**). Addition of CH_3^\bullet to PC radical anion was found to be a rate determining step with a surmountable activation energy of 22.1 kcal/mol. In particular, this radical addition reaction provided an energetically stable intermediate with 8.3 kcal/mol lower energy than that of PC radical anion, which confirms the feasibility of this reaction pathway. Further in-depth studies are currently underway to understand the full mechanistic pathways of the 4DP-IPN photodegradation including the formation of CH_3^\bullet .

We then investigated the photodegradation behavior of 4DP-IPN in an actual dehalogenation reaction of various aryl halides (**Figure X6a**). Aliquots of reactions were taken at a given time to monitor the reactions as well as PC degradation. Since only a small amount of PC (here, 1 mol%) was used, the degradations of PC were monitored by TLC whereas the reactions were tracked by $^1\text{H-NMR}$.

As shown in **Figure X6a: Reaction A**, 100% yield was obtained for 4-bromobenzonitrile in 1.5 h at which the trace amount of the photodegradation product, 4DP-Me-BN, was observed, implying that 4DP-IPN is an active PC for the dehalogenation reaction of 4-bromobenzonitrile. Here, the decomposition of the PC was significantly retarded in the dehalogenation reaction as compared to that in the presence of DIPEA alone (see **Figure X5a: TLC at 455 nm**). This is likely due to the fact that the PC degradation competes with the dehalogenation reaction, as described in **Figure X6c**. In other words, in the presence of 4-bromobenzonitrile, the electron transfer from 4DP-IPN $^-$ to 4-bromobenzonitrile seems to be significantly faster than i) the formation of 4DP-IPN $^-$ and ii) the substitution reaction to form 4DP-Me-BN, which results in a substantial delay in the PC degradation. In fact, more PC degradation was observed in the dehalogenation reactions of more challenging substrates (*i.e.* 4-chlorobenzonitrile and 4-bromoanisole, see **Figure X6a: Reactions B, C**), further supporting our hypothesis. More interestingly, 10.5 h additional continuous irradiation after the completion of the dehalogenation reaction of 4-bromobenzonitrile provided only a small amount of the PC degradation adduct, suggesting that the PC degradation is still retarded in the absence of 4-bromobenzonitrile (see **Figure X6a: Reaction A, TLC**). This might be due to the fact that the reaction product, benzonitrile ($E_{\text{red}}^\circ = -2.4 \text{ V}$), can act as an electron acceptor, instead of 4-bromobenzonitrile that competes with the PC degradation process.

We further examined the catalytic activity of PC degradation adduct, 4DP-Me-BN (see **Figure X6b**). We used 1 mol% of 4DP-Me-BN as a PC instead of 4DP-IPN to proceed the dehalogenation reactions of 4-bromobenzonitrile and 4-bromoanisole; here, it is noted that the reactions proceeded under exactly the same conditions, only except for the PC. With 4DP-Me-BN as a PC, 71% and 3% yields were obtained for 4-bromobenzonitrile and 4-bromoanisole, respectively, whereas 4DP-IPN provided 100% and 42% yields under the same conditions, indicating the lesser catalytic performance of 4DP-Me-BN. Reaction kinetics studies of 4-bromobenzonitrile further confirmed the lesser catalytic activity of 4DP-Me-BN compared to 4DP-IPN (see **Figure X6b, right**).

Finally, the molecular origin of lower catalytic activity of 4DP-Me-BN was studied. According to the DFT calculations, CH₃ substitution did not significantly change the three-dimensional structure of the molecule, and thus the molecular structure of 4DP-Me-BN remained similar to that of 4DP-IPN, a strongly twisted D-A conformation (see **Figure X7a**). Unlike the three-dimensional structure, the electronic structure was found to be substantially changed in 4DP-Me-BN. The substitution of strongly electron-withdrawing CN with weakly electron-donating CH₃ in the acceptor moiety located at the center of the molecule resulted in an increase of HOMO and LUMO energies. The increase of LUMO energy is steeper than that of HOMO, because the acceptor is directly affected by CH₃ substitution, while the donor is indirectly affected by the inductive effect caused by the decrease in the electron-withdrawing power of the acceptor unit.

Figure X7. (a) Calculated energies and topologies of frontier molecular orbitals (MO) of 4DP-Me-IPN. (b) PL decay of 4DP-IPN (1.0×10^{-5} M; orange line) and 4DP-Me-BN (1.0×10^{-5} M; green line) in a CH₃CN solution at RT. (c) Steady-state PL spectra of 4DP-IPN (1.0×10^{-5} M; orange line) and 4DP-Me-BN (1.0×10^{-5} M; green line) in a CH₃CN solution at RT. (d) UV/Vis absorption spectra of 4DP-IPN (1.0×10^{-5} M; orange line) and 4DP-Me-BN (1.0×10^{-5} M; green line) in a CH₃CN solution at RT. (e) Stern-Volmer plots for the PL quenching of 4DP-Me-BN (1.0×10^{-5} M) in CH₃CN by DIPEA at RT. Stern-Volmer plots were obtained from PL decays of 4DP-Me-BN, excitation at $\lambda_{ex} = 377$ nm, detection at $\lambda_{det} = 470$ nm.

Photophysical and electrochemical measurements are in accordance with the QC calculations. Strong delayed fluorescence with a long decay lifetime of 150 μs was seen in 4DP-Me-BN under purged conditions (**Figure X7b**), indicating that the three-dimensional structure remains in a strongly twisted D-A conformation and hence, long-lived triplet excitons are produced well in 4DP-Me-BN. UV/vis and PL spectra of 4DP-Me-BN showed a distinct blue shift as compared to those of 4DP-IPN (**Figures X7c, d**), which confirms that the LUMO energy increases more than the HOMO energy upon CH₃ substitution. The decrease in excited state reduction potential ($E_{red}^*(4DP-Me-BN) = 0.54$ V) and ground state reduction potential ($E_{red}^0(4DP-Me-BN) = -2.05$ V) is also well-matched to the DFT results where 4DP-Me-BN shows higher HOMO and LUMO energies.

We still questioned the reduced catalytic performance of 4DP-Me-BN compared to that of 4DP-IPN despite its i) highly negative reduction potential, ii) decent long-lived triplet exciton generation, and iii) greatly improved stability. We

concluded that the lowered catalytic activity might be caused by i) a very low light absorption at 455 nm of 4DP-Me-BN (clearly confirmed by UV/Vis spectrum as shown in *Figure X7d*) and ii) a decrease in the photoinduced electron transfer rate between 4DP-Me-BN and DIPEA due to the lowered excited state reduction potential of 4DP-Me-BN. In fact, k_{PET} of 4DP-Me-BN in the presence of DIPEA was measured by a PL quenching experiment in which approximately one order of magnitude lower value ($k_{PET}(4DP-Me-BN) = 8.6 \times 10^4 \text{ M}^{-1}\cdot\text{s}^{-1}$) was obtained as compared to that of 4DP-IPN, consistent with Marcus “normal region” behavior as we described in the main text (see *Figure X7e*), which supports our hypothesis.

In summary, from the additional experiments and QC calculations, the following conclusions could be drawn: i) 4DP-IPN is degraded to 4DP-Me-BN in the presence of DIPEA under visible light irradiation conditions in which the 4DP-IPN radical anion plays an essential role as an intermediate (*Figure X5*), ii) the PC degradation competes with the dehalogenation reaction so that it is retarded in actual dehalogenation reaction conditions (*Figure X6*), iii) the PC degradation adduct, 4DP-Me-BN showed much lower catalytic performance and therefore, 4DP-IPN is mainly responsible for the dehalogenation reaction.

All these contents are properly reflected in the revised manuscript.

Another issue I have has to do with the novelty of the PC design. I am concerned that photophysical attributes being discussed as important may not be meaningful or maybe not even be present. Before proceeding I would say that it is clear that the triplet lifetimes are an excellent attribute of these PC systems and are very likely to be important. For example, they enable electron transfer (sacrificial amine to PC*) even in situations where the driving force is zero or even positive such as with the three sacrificial reductants (DIPEA, TBA, and TEA) explored herein. For example, with DIPEA concentrations of 0.1 M, electron transfer to the PC* is still taking 10 microseconds. In other words, the initial electron transfer is slow (not at all fast as the Fig. 1d suggests) and the long lifetime of the triplet saves the day. In terms of the more speculative issues that I think are unlikely to impact mechanism, I would point to the discussion about spin states retarding back electron transfer rates (*Figure 1d* again). In the radical pair it is hard to imagine much of an interaction between the spin $\frac{1}{2}$ systems on the oxidized amine donor and the reduced PC acceptor. Even if they are born coupled (on a 10 microsecond time scale as discussed above) it is unlikely that they would remain that way for long given the weak exchange coupling and the geometric fluctuations of the ion pair in solvent. It seems to me that the issues saving the day are (a) the substantial driving force for back electron transfer (-2.3 eV based on the reduction potentials of the PC and DIPEA) which would make this firmly in the inverted region and (b) the speed of H-atom abstraction to cement the DIPEA radical cation as an iminium salt. To this latter point, these researchers should be discussing the importance of this aspect of the reaction (its barely mentioned in the modeling in the SI). To me it seems that this chemistry of the sacrificial reductant (and the driving forces of the overall reaction) is essential for high yield given that the initial photoinduced electron transfer from DIPEA to PC* is thermodynamically uphill and given the reduction of the aryl halides are also thermodynamically uphill (as mentioned in the SI).

Response: First of all, we really appreciate the referee’s very constructive and critical comments. In principle, we agree with the referee’s argument that the chemistry of the sacrificial reductant is essential for the generation of long-lived radical anion intermediates (PC⁻) that are a basis for successful dehalogenation reactions of various aryl/alkyl halides.

On the basis of known theory with a proper approximation, we calculated the reorganization energy for the electron transfer process between the PC^{•-} and DIPEA^{•+} (**Figure X8**). The calculated value is *ca.* 0.9 eV that is substantially lower than the driving force for the BET process (*ca.* -2.3 eV), implying that the BET is likely located in “Marcus inverted region” as the referee pointed out. Therefore, in the current system, the BET process can be retarded for a radical anion with a higher reducing power.

Figure X8. Summary of the experimental and calculated values to obtain the rate constant, (a) k_{ET} , for electron transfer from 4DP-IPN^{•-} to 4-bromobenzonitrile and (b) k_{BET} , for electron transfer from 4DP-IPN^{•-} to DIPEA radical cation. Molecular radii of each species are indicated in orange color and calculated rate constants are indicated in dark turquoise color; Z is the pre-exponential factor, ΔG^\ddagger is the activation energy of the reaction, and λ is the reorganization energy.

Given this fact, the question still remains: Why is the PC radical anion with a highly negative reduction potential rarely reported so far? Highly negative reduction potential (E_{red}°) of a PC should require a high-energy LUMO. In order to ensure visible-light absorption in such a PC, the HOMO energy should also be high, which in turn, decreases the excited state reduction potential (E_{red}^*) that makes PET between a PC and sacrificial reductant difficult. Since the synthetic chemists commonly choose a visible-light absorbing dye as a PC and a tertiary amine as a sacrificial reductant for a reductive cycle, it is very difficult to find out a PC having both i) a decent initial PET with a sacrificial reductant and ii) a highly negative reduction potential.

A PC with a highly efficient generation of long-lived triplet excitons provides a solution. Since abundant generation of long-lived triplet states maximizes the concentrations of excited state PC and hence, facilitates thermodynamically unfavorable PET processes, a PC can have a good visible-light absorption as well as a highly negative reduction potential. Moreover, since electron transfer between the triplet excited state of PC ($^3\text{PC}^*$) and a sacrificial electron donor produces a triplet contact radical ion pair (TCRIP), the BET process could further be retarded, which helps the generation of radical anions.

Figure X9. Schematic description of photoinduced electron transfer (PET) between a purely organic PC and a (sacrificial) reducing agent (Red) for a different spin state of the PC in the excited state, i.e. singlet (left-hand side) and triplet (right-hand side). Spin-multiplicities as well as electron configurations are indicated in each state of the PC. Enhanced PET in the case of $^3\text{PC}^*$ is due to the fact that $[^3\text{PC}^*] \gg [^1\text{PC}^*]$ for continuous illumination.

In addition to all these, in-depth understanding of the fate of a radical anion is essential for the development of a catalyst and photocatalytic reaction. However, this is still largely lacking, which leads to many misinterpretations of the photocatalytic phenomena. For example, very recently, Wu's group reported a new dicyanoarene-based PC (i.e. 3CzEPAIPN, see **Figure X10a**) with a long-lived excited state of its radical anion that facilitates the challenging C-X bond activation through a highly efficient ConPET process. However, considering our experience concerning radical anion intermediates of a variety of cyanoarene-based PCs, the interpretation of Wu's group might be wrong. This is clearly supported by the fact that 456 nm LED irradiation on a mixture of acetonitrile solutions of "3CzEPAIPN + DIPEA" differing only in the concentration of 3CzEPAIPN (3.2×10^{-5} M and 1.42×10^{-2} M, respectively) gave substantially different color (see **Figures X10d, e**). Mixture solutions of lower 3CzEPAIPN concentration (3.2×10^{-5} M) turns its color yellow to pale yellow (blue shift in UV/Vis absorption) upon 456 nm LED irradiations whereas solutions of higher CzEPAIPN concentration (1.42×10^{-2} M) change its color yellow to brownish-yellow (red shift in UV/Vis absorption).

As we reported in our manuscript, all the cyanoarene-based PCs showed distinct red-shift in their UV/Vis spectra upon light irradiation in the presence of DIPEA, which is also well in-accordance with QC calculations. We thus speculate that the blue shift in UV/Vis absorption and PL emission observed by Wu's group (see **Figures X10b, c**) arises not from its radical anion ($3\text{CzEPAIPN}^{\cdot-}$), but from its degradation adduct. As can be seen in **Figures X6**, in the additional experiments, the degradation pathway of 4DP-IPN was clearly revealed. Isolation of the degradation product by column chromatography followed by intensive NMR analyses (^1H NMR, COSY, and NOESY) clearly confirmed that a methyl substitution reaction occurred at the CN position of 4DP-IPN to generate 4DP-Me-BN (**Figure X5a**). The degradation adduct, 4DP-Me-BN, showed clear blue shift in UV/Vis absorption and PL emission (**Figures X7b-e**), which is very similar to the observation made by the Wu's group. Given that 3CzEPAIPN has similar structure with 4DP-IPN, Wu's

group might observe 3CzEPAMeBN, not 3CzEPAIPN^{•-}.

All these contents are properly reflected in the revised manuscript. With the help of the referees, we now provide very detailed mechanistic insight of a radical anion-mediated photoredox catalysis, which will be valuable for the whole field of organic photocatalysis.

Figure X10. (a) Chemical structures of 3CzEPAIPN and the lifetime of 3CzEPAIPN^{•-} reported by Wu's group (*J. Am. Chem. Soc.* 13266 (2021)) The structure shown on the right shows the degradation product 3CzEPAIPN inferred from our experiments. (b) Absorbance profiles for 3CzEPAIPN and 3CzEPAIPN^{•-} (3.3×10^{-5} M). (c) Emission profiles (excitation at 390 nm) for 3CzEPAIPN and 3CzEPAIPN^{•-} (3.3×10^{-5} M, 1.0 cm path length). (d) A color change from yellow to light yellow was observed before and after light irradiation with the solution of 3CzEPAIPN (3.2×10^{-5} M) and DIPEA after irradiation with 456 nm Kessil LED (2×40 W). (e) A color change of the solution of 3CzEPAIPN (3.2×10^{-5} M) and DIPEA (3.1×10^{-1} M) after irradiation with 456 nm Kessil LED (2×40 W). All content of **Figure X10** is reproduced from the cited publication, except the structure of the methylated adduct shown on the right-hand side of **Figure X10a**.

Additionally, in terms of H-atom abstraction, the rate constant of H-atom abstraction to cement the trialkyl amine radical cation as an iminium salt have been studied widely and already reported (*Chem. Rev.* 2016, 116, 10075). The rate constants of deprotonation for H-atom at α -carbon of tertiary alkylamine have been estimated ranging from 10^5 to 10^8 $M^{-1}\cdot s^{-1}$ according to the measurement methods. Considering that the rate of deprotonation of trimethylamine radical cation is so fast ($pK_a(NMe_3^{\bullet+}) = 8$), it is reasonable to assume that the rate of deprotonation of the DIPEA radical cation is also very fast approaching 10^8 $M^{-1}\cdot s^{-1}$. Therefore, in kinetic modelling of dehalogenation, the rate constant of deprotonation of the DIPEA radical cation was set to 10^8 $M^{-1}\cdot s^{-1}$. Furthermore, in the kinetic simulation, as the rate of deprotonation was changed ranging from 10^5 to 10^8 $M^{-1}\cdot s^{-1}$, the simulated yields and kinetics were not changed significantly, which means the H-atom abstraction is not rate-determining.

A third issue of concern to me involves the discussions of ConPET. In the text (line 269) it is stated that addition of bromo anisole to a solution of the PC radical anion results in no color change. This seems to contradict what is seen in Fig S7. Is the reader to draw a distinction from the shades of yellow produced when the bromoanisole is added? To me it looks from the photos more like PC than PC⁻ and if it is the former then the ConPET is not needed. I am not intending

to nit-pick but the issues of mechanism are made to sound simpler than they appear and that detracts from the impact of the work.

Response: Sorry for the misunderstanding of the referee caused by experiments performed by us with insufficient control of the reaction conditions. Since we did not purge with inert gas in the case of the bromoanisole (and also, 4-bromobenzonitrile), the PC radical anion disappeared substantially. In our additional experiments, we obtained high quality images by purging the bromoanisole and 4-bromobenzonitrile with inert gas (see **Figure XI1**).

Figure XI1. Evaluation of PET process in photoredox reductive dehalogenation. Inside glovebox using flame-dried glass vials, the radical anion of 4DP-IPN (1.0×10^{-3} M) was generated with DIPEA (0.5 M) in CH₃CN (2 ml) under 3W 515 nm LED for 3 min, aryl bromides solution (0.1 M) as quencher in CH₃CN (0.2 ml) were added. Subsequently, the added solutions were re-irradiated under 3W 515 nm LED for 3 min. All pictures were taken immediately without any additional delay. Based on these observations, it was suggested that unactivated aryl halides can be reduced from 4DP-IPN^{•-}.

Specific issues:

On line 161 it is stated that the PC radical anion once formed is stable for a week in the absence of O₂. This seems impossibly long for a system with -2.3 eV of driving force for back electron transfer to the oxidized DIPEA. Did the authors generate the PC radical anion chemically in this case? If not and DIPEA was used, then there must be something going on besides capping (there is no aryl halide present in this example) and that could be very important in the mechanism.

Response: This is a misunderstanding of the referee caused by our unclear statement in the text. In our previous experiments, the radical anion stability was qualitatively examined in daylight conditions; the nitrogen-purged sample was stored on the bench top in the laboratory where light from sun and fluorescent lamps comes in. To prevent the confusion of readers, a more quantitative experiment was carried out as shown in **Figure XI2**. Changes in the UV/Vis spectrum of freshly generated 4DP-IPN radical anion were recorded every 2 min under perfectly dark conditions. Gradual decay was observed for more than 20 min, which is approximately 2 times longer than that of Ir(dtbbpy)(ppy)₂PF₆ in similar conditions (CH₃CN, DIPEA) observed by König's group (**Figure 2c, Nat. Catal.** 2020, 40).

Figure X12. (a) Schematic description of back-electron transfer (BET) between $PC^{\bullet-}$ and a radical cation of (sacrificial) reducing agent (Red) in the encounter complex. (b) Time-resolved UV/Vis absorption spectra of $4DP-IPN^{\bullet-}$ (1.0×10^{-4} M) in CH_3CN at RT. UV/Vis absorption spectra were taken from the degassed mixture solutions of $4DP-IPN$ (1.0×10^{-4} M) and $DIPEA$ (0.5 M) in CH_3CN right after illumination by two 3W 515 nm LEDs for 3 min at RT. The absorbance decay at 525 nm was monitored at every 2 min.

Were any controls explored for yields? For example measuring yields in the absence of PC or in the absence of sac reductant? This seems standard in the PC field.

Response: We have properly added the negative control experiments in **Table 1** in the revised manuscript.

Table X1.

Entry	PC	Loading (mol%)	Yield (%)
1	–	–	0
2	$Ru(bpy)_3Cl_2$	5	13
3	$Ru(phen)_3(PF_6)_2$	5	0
4	Rh6G	5	2
5 ^a	PDI	5	0
6 ^{a,b}		5	4
7	4Cz-IPN	5	75
8 ^a		5	100
9	fac -Ir(ppy) ₃	0.5	100
10		0.05	26
11		0.005	0
12		5	100
13		0.5	100
14		0.05	100
15	4DP-IPN	0.005	100
†16 ^c		0.005	0
†17 ^d		0.005	0

Reactions were performed with 4-bromobenzonitrile (0.1 M), PC (5 to 0.001 mol%), DIPEA (10.0 equiv.) in CH_3CN (1 ml) under two 3W 455 nm LEDs for 8 hours. All solutions were prepared inside glovebox and were degassed by bubbling with Ar. ^aReaction was performed in DMF due to solubility. Control experiments were performed ^bwith TEA instead of DIPEA, ^cin the dark room or ^din the absence of DIPEA. Yields were determined by GC-FID or [†]¹H NMR using 1,3,5-trimethoxybenzene as an internal standard.

Is it possible to use EPR to confirm generation of the PC radical anions? Or does the degradation issues preclude generation of the needed concentrations? Can the PC radical anions be produced chemically allowing correlations to be made between EPR and UV-Vis?

Response: All the PC degradation issues are properly addressed in our responses above, which confirms that all measured photophysical properties of the PC radical anion are now reliable.

REVIEWER COMMENTS

Reviewer #1 (Remarks to the Author):

Overall, the authors have substantially updated this manuscript and offered significant new insights beyond the works recently published by Wu and Wickens. Additionally, the authors make a good case in their revision letter for the merits of their work. Although I still have concerns regarding the synthetic utility of the reactions developed, I would now support the publication of this manuscript in Nature Communications. However, I do have a lingering concern with the scholarship of this revised manuscript.

A central concern of my initial review centered around how this project interfaces with the recent work from Wickens (now cited, refs 23 and 26) as well as Wu (now cited, ref 14). These projects, however, are not discussed in the main text at all and remain extremely related to the work. While extensive discussion of these projects is not necessary as this manuscript reports fairly contemporaneous findings, in my opinion, two things struck me as particularly odd:

1) In section 2.1, the authors lay out their design strategy for why they selected 4DP-IPN as a prospective photocatalyst for their study. While this rationale is clear and well-reasoned, the work from Wickens on the same catalyst, revealing very similar features should probably be explicitly mentioned. Ideally, the related work from Wu should also be cited but since this is technically with a different catalyst, this is slightly less fundamental to the project. It would be totally reasonable for the authors to make an explicit mention that their work was conducted in parallel but ignoring the work in the main text feels inappropriate.

2) As a more minor point: the following line: "Since König and coworkers reported the ConPET of a perylene diimide (PDI) analogue for the reduction of aryl chlorides, 7 Nicewicz and coworkers, 8 Miyake and coworkers, 22 Wenger and coworkers, 18 and others^{20,23} further advanced this strategy by developing a new catalyst system, 24 by merging electrocatalysis,^{20,25,26} and by optimizing reaction conditions, 27 which enables unactivated alkyl/aryl chloride activations, 8,²¹ photoredox-mediated Birch reductions, 22 and sulfonamide cleavages 8 that are not possible by conventional photoredox catalysis." – 'and others' refers exclusively to Wickens' papers on this topic (Refs 20 and 23). It seems more appropriate to refer to the group by name or change the overall construction of this sentence.

Reviewer #2 (Remarks to the Author):

The authors have satisfactorily addressed all my previous comments. I would recommend the publication of the manuscript.

Reviewer #4 (Remarks to the Author):

Overall, the authors have conducted a significant amount of additional experimental and computational work to examine the degradation pathway of the photocatalyst, 4DP-IPN, in the course of hydrodehalogenation reactions. A degraded species was isolated and characterized via TLC, NMR, UV-visible absorption, cyclic voltammetry, and time-resolved photoluminescence studies. This work likely has broad applicability in improving the mechanistic understanding of reactions catalyzed by 4DP-IPN in general which employ DIPEA as a sacrificial donor. Broadly speaking, rigorous studies of the degradation pathways of organic photocatalysts as used in photoredox catalysis are greatly needed and will be beneficial to the field. As such, this new work presented by the authors has the potential to be both commendable and timely. Reviewer #3 provided a thoughtful analysis of some unaddressed mechanistic questions which the authors have endeavored to answer. While the experimental work done since the first submission is a significant step in the right direction, several important questions raised by Reviewer #3 have not been adequately addressed. As such, I cannot recommend publication of the updated manuscript in Nature Communications in its current form.

For clarity, I will consider each comment separately and in the order presented in the rebuttal. Firstly, Reviewer #3 stated:

“The observation that troubles me the most is that the PC is unstable in the presence of the sacrificial reductant alone when irradiated for longer than one minute (reactions are run for hours). The authors argue that this is an issue only when there is not aryl halide present. (I don't really have a sense of what the degradation pathway would be). Even if the reactivity of the aryl radical with the amine radical cation (the capping pathway) is fast, it would be the case that as the reaction proceeds towards 100%, the amount of aryl halide present becomes limited and a diffusion limited bimolecular reaction would be slowed. Thus it is hard to envision a situation where the PC survives the reaction as has been claimed by the authors. Since it is hard to imagine the survival during long reactions, it brings up the critical question as to whether it, versus a byproduct, is the active catalyst. The authors could do more to support their mechanistic idea. I think that Figure S7 (top pathway) could be helpful in that it suggests that the catalyst is regenerated when excess aryl halide is present. If the authors could experimentally connect a sequence of such events (1. Addition of aryl

halide to the PC radical anion; 2. Light back on until the aryl halide is consumed and the PC radical anion is re-formed) until there is spectroscopic (NMR) or TLC evidence for product – and not just regeneration of the catalyst – it would go a long way towards convincing the reader of the purported simple mechanism being driven by the PC radical anion alone.”

The authors observe the presence of a new blue-shifted emissive spot in TLC after irradiation of a mixture of 4DP-IPN and DIPEA which is assigned to a methyl-substituted derivative of 4DP-IPN. This species was isolated and characterized, and a possible pathway for its formation was studied via density functional theory (DFT) calculations. However, the pathway proposed is missing a necessary scheme. Specifically, there is no discussion of how a methyl radical might be formed in this system other than proposing acetonitrile solvent as the originating component. Acetonitrile is a stable solvent with a broad potential window as exemplified by its common use in electrochemistry and inert behavior under reducing potentials in the authors' electrochemical experiments. The authors cite published work on gamma-irradiated solid-state acetonitrile at low temperatures of 77-87 K in support of their idea that the methyl radical might originate from the solvent. I would raise the question as to whether these referenced conditions are relevant to the reaction conditions herein, namely room temperature solution with low energy visible light irradiation.

Furthermore, I am not convinced that the isolated derivative of 4DP-IPN has the structure the authors assign. While the NMR data is presented in the SI, I am concerned that there is no discussion of the 2D spectra, and the alkyl region of the 1D ¹H NMR spectrum is quite messy. The protons assigned as methyl integrate to only 2.44. Further, if this catalyst derivative is a new compound, the standard in the field is that its purity should be supported by at least one other technique – typically high resolution mass spectrometry (HRMS) is used. If it is not a new compound, its purity should be compared to reported data in the literature. The UV-vis data and time-resolved/steady state emission data is helpful here in the sense that it confirms the isolated species is distinct from 4DP-IPN and that it reacts with DIPEA at a slower comparative rate. It does not, however, support the asymmetric methyl-substituted structure proposed. Comparison of the UV-vis spectrum of the isolated byproduct with that predicted by TD-DFT could potentially be helpful here if there is good agreement, but I would emphasize that another experimental technique to establish the identity of the isolated species is necessary (HRMS or single crystal X-ray diffraction would be ideal). Finally, the degradation pathway is a highly interesting and important discovery, in my opinion vital this work, and its importance might be better communicated to the reader if it were added to the proposed mechanistic scheme in the main text.

Next, Reviewer #3 states:

“Another issue I have has to do with the novelty of the PC design. I am concerned that photophysical attributes being discussed as important may not be meaningful or maybe not even be present. Before proceeding I would say that it is clear that the triplet lifetimes are an excellent attribute of these PC systems and are very likely to be important. For example, they enable electron transfer

(sacrificial amine to PC*) even in situations where the driving force is zero or even positive such as with the three sacrificial reductants (DIPEA, TBA, and TEA) explored herein. For example, with DIPEA concentrations of 0.1 M, electron transfer to the PC* is still taking 10 microseconds. In other words, the initial electron transfer is slow (not at all fast as the Fig. 1d suggests) and the long lifetime of the triplet saves the day. In terms of the more speculative issues that I think are unlikely to impact mechanism, I would point to the discussion about spin states retarding back electron transfer rates (Figure 1d again). In the radical pair it is hard to imagine much of an interaction between the spin $\frac{1}{2}$ systems on the oxidized amine donor and the reduced PC acceptor. Even if they are born coupled (on a 10 microsecond time scale as discussed above) it is unlikely that they would remain that way for long given the weak exchange coupling and the geometric fluctuations of the ion pair in solvent. It seems to me that the issues saving the day are (a) the substantial driving force for back electron transfer (-2.3 eV based on the reduction potentials of the PC and DIPEA) which would make this firmly in the inverted region and (b) the speed of H-atom abstraction to cement the DIPEA radical cation as an iminium salt. To this latter point, these researchers should be discussing the importance of this aspect of the reaction (its barely mentioned in the modeling in the SI). To me it seems that this chemistry of the sacrificial reductant (and the driving forces of the overall reaction) is essential for high yield given that the initial photoinduced electron transfer from DIPEA to PC* is thermodynamically uphill and given the reduction of the aryl halides are also thermodynamically uphill (as mentioned in the SI)."

The authors effectively address Reviewer #3's points here regarding driving forces, and they mention a relevant literature example where their mechanistic findings are likely applicable. As Reviewer #3 mentioned concerns regarding PC novelty, I noticed that the PCs in this work were synthesized by the authors, but it was not emphasized whether any of them are new compounds. In fact, there is only a short discussion of these PCs in the main text. Some discussion on the structure property relationships such as the effect of substituents on absorption spectra would be quite interesting and would add to the impact of this work. In addition, comparison of how PC properties might affect the catalytic reaction performance across the PCs would be great to see.

Reviewer #3:

"A third issue of concern to me involves the discussions of ConPET. In the text (line 269) it is stated that addition of bromo anisole to a solution of the PC radical anion results in no color change. This seems to contradict what is seen in Fig S7. Is the reader to draw a distinction from the shades of yellow produced when the bromoanisole is added? To me it looks from the photos more like PC than PC- and if it is the former then the ConPET is not needed. I am not intending to nit-pick but the issues of mechanism are made to sound simpler than they appear and that detracts from the impact of the work."

The authors reply here with a series of photos under various conditions. However, it would be beneficial to acquire UV-vis spectra under these conditions instead. As Reviewer #3 mentions, it is

difficult to conclude which pictures are different by eye. The authors suggest that this data supports the conPET mechanism. While these observations may be consistent with a conPET pathway in the case of more challenging aryl halides as substrates, the authors' conclusion would be greatly strengthened by the use of time-resolved measurements to follow the excited state of the radical anion to probe directly whether it reacts with the substrate. In the event access to such techniques is limited, the authors could probe whether slow reactivity with the substrate is observed with the photogenerated radical anion in the absence of light that may also be consistent with the above observations. Potentially, generating the radical anion in larger concentration via bulk electrolysis, turning off the applied potential, and then dosing in substrate in the dark would be a useful experiment. Without some further experimentation, I would be wary of proposing the conPET mechanism in light of the alternate hypothesis that the observed reaction scope may be obtainable via ground state thermal reactivity of the radical anion.

A series of specific comments follow. I will copy here only one which was insufficiently addressed.

“Is it possible to use EPR to confirm generation of the PC radical anions? Or does the degradation issues preclude generation of the needed concentrations? Can the PC radical anions be produced chemically allowing correlations to be made between EPR and UV-Vis?”

The authors did not attempt to generate the radical anions via another method or perform EPR measurements. I think that EPR measurement is important to confirm that the red-shifted species are, in fact, radical anions. The UV-vis kinetic measurement suggests that the red-shifted species is sufficiently long-lived for steady state EPR measurements.

Lastly, I had a few additional specific comments that came up in reading the manuscript, SI, and responses that I think would be beneficial to this work if they could be addressed.

Figure 1: No reference electrode is given for the potentials reported in this figure and it is unclear how they were calculated/measured or what the sources were for reported values. Reference electrode information is further missing in the text and other figures.

Figure 2: There is a -0.87 V difference between the computed doublet reduction potential and the experimental value which is given for a different excited state. It is also unclear how the experimental value was measured – was this a computational value that is mislabeled? There is also the troubling lack of observation of the computed absorption at ~1500 nm. Perhaps further benchmarking of the TD-DFT here with experimental values would be helpful to support the accuracy of the excited state calculations.

The Stern-Volmer plot for Ru(bpy)₃²⁺ shows a smaller slope and yet is assigned a higher rate constant – there must be some error here.

The Jablonski diagram contains detailed rate constants from the kinetic simulation. However, it is unclear how the triplet quantum yield was obtained. As far as I can see, no experiments were performed to determine this quantum yield via traditional experimental methods, such as by using nanosecond transient absorption – is it known in the literature?

Figure 5: reactions A and B, but not C, lead to formation of a blue emitting species shown in the TLC. This species is not discussed and may be another byproduct derived from the catalyst. Can the authors isolate this spot? Does this species play a role in the reaction?

Figure 7: The kinetic simulations seem to give very different results from the experimental kinetics here. Can the authors please comment on why this is the case and how the model might be improved? My worry here is that the disagreement may reflect reactivity that is occurring but is not included in the current model.

The photoreactor setup does not control the position of the vial, and so the distance from the LED to the reaction solution will vary with each setup. This lack of control may lead to issues with reproducibility and with modeling of the light flux. Setups which control vial position are the standard in the field.

Throughout the text, the authors use strong language which implies conclusions which are not consistent with the data. In some cases, simply modifying the word choice will help, while in others the conclusions may need to be reevaluated. Here are two examples:

In the last paragraph before the conclusion, the authors state “the solution of 4DP-IPN^{•-} showed no evident color change in the presence of excess 4-bromoanisole ($E_{red}^0 = -2.70$ V). This suggests that no direct electron transfer between 4DP-IPN^{•-} and 4-bromoanisole takes place and a multiphoton excitation process might be involved.”

This statement does not follow from the observation as the alternate hypothesis of a slow, thermal electron transfer from the ground state radical anion could also account for the lack of color change.

A few sentences down, the authors state “given the low to moderate yields of inactivated aryl chlorides, it seems that the ConPET process is not very efficient here (Table 2). This is presumably due to the short excited-state lifetime of 4DP-IPN•–, which is supported by the fact that steady-state PL emission was not observed for the solution of 4DP-IPN•–”

The lack of observable steady state emission may mean it is difficult to follow the excited state by techniques that depend on emission but cannot inform about the excited state lifetime. For example, the triplet excited states of PCs in this study are long lived yet do not display detectable steady state emission.

Reviewers' comments to author:

We are grateful to the reviewers for their constructive comments and suggestions, which greatly helped to clarify a number of important points, avoid misunderstandings, and improve the clarity and referencing of the paper. The reports of the three reviewers were overall very positive on the significance and novelty, stressing the suitability of our work for publication in *Nature Communications*. We have carefully revised our manuscript in line with all the reviewers' comments. The point-by-point response is given below. Our responses to reviewers' comments are highlighted in **blue color** with a highlight in the revised manuscript and SI.

Reviewer #1 (Remarks to the Author):

Overall, the authors have substantially updated this manuscript and offered significant new insights beyond the works recently published by Wu and Wickens. Additionally, the authors make a good case in their revision letter for the merits of their work. Although I still have concerns regarding the synthetic utility of the reactions developed, I would now support the publication of this manuscript in *Nature Communications*. However, I do have a lingering concern with the scholarship of this revised manuscript.

A central concern of my initial review centered around how this project interfaces with the recent work from Wickens (now cited, refs 23 and 26) as well as Wu (now cited, ref 14). These projects, however, are not discussed in the main text at all and remain extremely related to the work. While extensive discussion of these projects is not necessary as this manuscript reports fairly contemporaneous findings, in my opinion, two things struck me as particularly odd:

1) In section 2.1, the authors lay out their design strategy for why they selected 4DP-IPN as a prospective photocatalyst for their study. While this rationale is clear and well-reasoned, the work from Wickens on the same catalyst, revealing very similar features should probably be explicitly mentioned. Ideally, the related work from Wu should also be cited but since this is technically with a different catalyst, this is slightly less fundamental to the project. It would be totally reasonable for the authors to make an explicit mention that their work was conducted in parallel but ignoring the work in the main text feels inappropriate.

Response: We thank the reviewer for his encouraging comments and constructive suggestions, which definitely help us to prevent missing the important previous works conducted by other groups within the community of photocatalysis research. As the reviewer suggested, we properly mentioned the works from Wickens' and Wu's group in the "2.1. Design strategy" of the revised manuscript as follows: *"In fact, our hypothesis is strongly supported by very recent works in the groups of Wickens^{24,25} and Wu²⁶, who employed 4DP-IPN and analogues as efficient PCs for radical anion-mediated photoredox catalysis."*

2) As a more minor point: the following line: "Since König and coworkers reported the ConPET of a perylene diimide (PDI) analogue for the reduction of aryl chlorides, 7 Nicewicz and coworkers, 8 Miyake and coworkers, 22 Wenger and coworkers, 18 and others^{20,23} further advanced this strategy by developing a new catalyst system, 24 by merging electrocatalysis,^{20,25,26} and by optimizing reaction conditions, 27 which enables unactivated alkyl/aryl chloride activations, 8,21 photoredox-mediated Birch reductions, 22 and sulfonamide cleavages 8 that are not possible by

conventional photoredox catalysis." – 'and others' refers exclusively to Wickens' papers on this topic (Refs 20 and 23). It seems more appropriate to refer to the group by name or change the overall construction of this sentence.

Response: Following the reviewer's suggestions, all issues regarding the citation have properly been addressed in the revised manuscript as follows: "Since König and coworkers reported the ConPET of a perylene diimide (PDI) analogue for the reduction of aryl chlorides,⁷ the research teams of Nicewicz,^{8,22} Wenger,²¹ Wickens,^{23–25} and Wu²⁶ further expanded this strategy by reporting new catalyst systems based on highly reducing neutral radicals,⁸ carbon dioxide radical anions,²⁵ radical anions with long-lived excited states,²⁶ electrocatalysis,^{23,27} and optimizing reaction conditions,²⁸ thereby enabling unactivated alkyl/aryl chloride activations,^{8,9} photoredox-mediated Birch reductions,²² and sulfonamide cleavages⁸ that are not possible by conventional photoredox catalysis. The mechanistic pathway of this process, however, is still under debate.²⁹"

Reviewer #2 (Remarks to the Author):

The authors have satisfactorily addressed all my previous comments. I would recommend the publication of the manuscript.

Response: We thank the reviewer for the positive evaluation. We believe that the reviewer's comments helped us to significantly improve our manuscript

Reviewer #4 (Remarks to the Author):

Overall, the authors have conducted a significant amount of additional experimental and computational work to examine the degradation pathway of the photocatalyst, 4DP-IPN, in the course of hydrodehalogenation reactions. A degraded species was isolated and characterized via TLC, NMR, UV-visible absorption, cyclic voltammetry, and time-resolved photoluminescence studies. This work likely has broad applicability in improving the mechanistic understanding of reactions catalyzed by 4DP-IPN in general which employ DIPEA as a sacrificial donor. Broadly speaking, rigorous studies of the degradation pathways of organic photocatalysts as used in photoredox catalysis are greatly needed and will be beneficial to the field. As such, this new work presented by the authors has the potential to be both commendable and timely. Reviewer #3 provided a thoughtful analysis of some unaddressed mechanistic questions which the authors have endeavored to answer. While the experimental work done since the first submission is a significant step in the right direction, several important questions raised by Reviewer #3 have not been adequately addressed. As such, I cannot recommend publication of the updated manuscript in Nature Communications in its current form.

For clarity, I will consider each comment separately and in the order presented in the rebuttal. Firstly, Reviewer #3 stated:

“The observation that troubles me the most is that the PC is unstable in the presence of the sacrificial reductant alone when irradiated for longer than one minute (reactions are run for hours). The authors argue that this is an issue only when there is not aryl halide present. (I don’t really have a sense of what the degradation pathway would be). Even if the reactivity of the aryl radical with the amine radical cation (the capping pathway) is fast, it would be the case that as the reaction proceeds towards 100%, the amount of aryl halide present becomes limited and a diffusion limited bimolecular reaction would be slowed. Thus it is hard to envision a situation where the PC survives the reaction as has been claimed by the authors. Since it is hard to imagine the survival during long reactions, it brings up the critical question as to whether it, versus a biproduct, is the active catalyst. The authors could do more to support their mechanistic idea. I think that Figure S7 (top pathway) could be helpful in that it suggests that the catalyst is regenerated when excess aryl halide is present. If the authors could experimentally connect a sequence of such events (1. Addition of aryl halide to the PC radical anion; 2. Light back on until the aryl halide is consumed and the PC radical anion is re-formed) until there is spectroscopic (NMR) or TLC evidence for product – and not just regeneration of the catalyst – it would go a long way towards convincing the reader of the purported simple mechanism being driven by the PC radical anion alone.”

The authors observe the presence of a new blue-shifted emissive spot in TLC after irradiation of a mixture of 4DP-IPN and DIPEA which is assigned to a methyl-substituted derivative of 4DP-IPN. This species was isolated and characterized, and a possible pathway for its formation was studied via density functional theory (DFT) calculations. However, the pathway proposed is missing a necessary scheme. Specifically, there is no discussion of how a methyl radical might be formed in this system other than proposing acetonitrile solvent as the originating component. Acetonitrile is a stable solvent with a broad potential window as exemplified by its common use in electrochemistry and inert behavior under reducing potentials in the authors’ electrochemical experiments. The authors cite published work on gamma-irradiated solid-state acetonitrile at low temperatures of 77-87 K in support of their idea that the methyl radical might originate from the solvent. I would raise the question as to whether these referenced conditions are relevant to the reaction conditions herein, namely room temperature solution with low energy visible light irradiation.

Response: We thank the reviewer for his/her critical/important comments and very constructive suggestions, which in fact greatly helped us to improve the quality of our work. In order to clarify the origin of CH_3^\bullet generation, we first carried out the synthesis of 4DP-Me-BN in CD_3CN , instead of CH_3CN (**Figure XI**). In contrast to our expectation that the CD_3 -substituted derivative of 4DP-IPN could be formed, the CH_3 -substituted product was obtained, clearly indicating that CH_3^\bullet was not generated from CH_3CN , as indeed pointed out by the reviewer.

Figure XI. (a) Reaction scheme of photobleaching reaction of 4DP-IPN. Reactions were performed with 4DP-IPN (1.0×10^{-2} M) and DIPEA (1 M) in CH_3CN (or CD_3CN) solution under two 3W 455 nm LEDs for 1 hour at RT without degassing process. The reaction was performed for 1 hour in open-to-air vial. In-situ TLC was monitored using EA:hexane (1:4 v/v) mixture. (b) Stacked ^1H NMR data of isolated 4DP-IPN and 4DP-Me-BN synthesized in CH_3CN (or CD_3CN). ^1H NMR data was measured in CDCl_3 at RT.

Figure X2. (a) Reaction schemes reported for organic synthesis using β -scission of radical cation of tertiary amine. (b) Proposed mechanistic pathway for photobleaching behavior of 4DP-IPN in the presence of DIPEA and DFT calculations for BDE of β -scission of DIPEA and DIPEA $^{+\bullet}$.

All content of Figure X2a is reproduced from the cited publication.

This unexpected result allowed us to rethink the origin of the CH_3^\cdot generation. Through an intensive literature survey (see **Figure X2a**), we realized that one-electron-oxidized DIPEA ($\text{DIPEA}^{+\cdot}$) is in fact the most probable source of methyl radical species. Interestingly, C–C (and C–H) bonds located in the β -position of $\text{DIPEA}^{+\cdot}$ are substantially weakened as compared to those of the neutral one (which also becomes evident from our DFT calculations, see **Figure X2b**) and hence, β -scission normally happens to generate the radical species (see examples in **Figure X2a**). As such, in our system, the methyl radical might be generated from the $\text{DIPEA}^{+\cdot}$ via the β -scission mechanism. We note, however, that we are unable to provide direct evidence of this pathway at this point, due to the practical difficulty in purchasing deuterated tertiary amines from commercial vendors. In any case, we note that DIPEA remains as the only CH_3 -source, after the preclusion of CH_3CN . Further in-depth studies will be conducted to achieve a full mechanistic understanding of the formation of 4DP-Me-BN. These are properly added in the revised manuscript as follows: “The latter might be generated via the β -scission of the one-electron-oxidized adduct of DIPEA (i.e. $\text{DIPEA}^{+\cdot}$). In fact, it is well known that C–C (and C–H) bonds located in the β -position of $\text{DIPEA}^{+\cdot}$ are substantially weakened as compared to those of neutral DIPEA; hence, β -scission normally takes place to generate the radical species,^{46–50} which is also well reproduced by our DFT calculations (**Figure 4b**). On the contrary, the possibility that CH_3^\cdot is generated from the solvent (i.e. CH_3CN) could be ruled out; in fact, in deuterated acetonitrile (CD_3CN), the CH_3 substitution reaction instead of the CD_3 substitution was still observed (**Figure 4a**).”

Furthermore, I am not convinced that the isolated derivative of 4DP-IPN has the structure the authors assign. While the NMR data is presented in the SI, I am concerned that there is no discussion of the 2D spectra, and the alkyl region of the 1D ^1H NMR spectrum is quite messy. The protons assigned as methyl integrate to only 2.44. Further, if this catalyst derivative is a new compound, the standard in the field is that its purity should be supported by at least one other technique – typically high resolution mass spectrometry (HRMS) is used. If it is not a new compound, its purity should be compared to reported data in the literature. The UV-vis data and time-resolved/steady state emission data is helpful here in the sense that it confirms the isolated species is distinct from 4DP-IPN and that it reacts with DIPEA at a slower comparative rate. It does not, however, support the asymmetric methyl-substituted structure proposed. Comparison of the UV-vis spectrum of the isolated byproduct with that predicted by TD-DFT could potentially be helpful here if there is good agreement, but I would emphasize that another experimental technique to establish the identity of the isolated species is necessary (HRMS or single crystal X-ray diffraction would be ideal). Finally, the degradation pathway is a highly interesting and important discovery, in my opinion vital this work, and its importance might be better communicated to the reader if it were added to the proposed mechanistic scheme in the main text.

Response: As the reviewer suggested, MALDI-TOF and DART HRMS experiments were performed, which further clarify the structural assignment of 4DP-Me-BN (see **Figure X3a**). In order to prevent confusion of the readers, the detailed structural assignment by 1D and 2D ^1H NMR (COSY, NOESY) of the bleached product was provided in the revised SI as shown in **Figure X3b** (**Figure S15** in the revised SI). Also, following the reviewer’s suggestion, we have added several sentences about the proposed mechanistic scheme in the revised manuscript as follows: “To validate our hypothesis, the most possible reaction pathway was screened by DFT calculations (**Figure 4b**). The addition of CH_3^\cdot to

PC radical anion was found to be a rate-determining step with a surmountable activation energy of 22.1 kcal/mol. In particular, this radical addition reaction provided an energetically stable intermediate with 8.3 kcal/mol lower energy than that of the PC radical anion, which confirms the feasibility of this reaction pathway. Further in-depth studies are currently underway to understand the full mechanistic pathways of the 4DP-IPN photodegradation including the formation of CH_3^- .

Figure X3. (a) MALDI-TOF (middle) and DART (right) HRMS of 4DP-Me-BN. (b) 1D (upper part), 2D COSY (left-lower part), 2D NOESY (right-lower part) ^1H NMR of 4DP-Me-BN. ^1H NMR data of 4DP-Me-BN was measured in CDCl_3 at RT.

Next, Reviewer #3 states:

“Another issue I have has to do with the novelty of the PC design. I am concerned that photophysical attributes being discussed as important may not be meaningful or maybe not even be present. Before proceeding I would say that it is

clear that the triplet lifetimes are an excellent attribute of these PC systems and are very likely to be important. For example, they enable electron transfer (sacrificial amine to PC*) even in situations where the driving force is zero or even positive such as with the three sacrificial reductants (DIPEA, TBA, and TEA) explored herein. For example, with DIPEA concentrations of 0.1 M, electron transfer to the PC* is still taking 10 microseconds. In other words, the initial electron transfer is slow (not at all fast as the Fig. 1d suggests) and the long lifetime of the triplet saves the day. In terms of the more speculative issues that I think are unlikely to impact mechanism, I would point to the discussion about spin states retarding back electron transfer rates (Figure 1d again). In the radical pair it is hard to imagine much of an interaction between the spin $1/2$ systems on the oxidized amine donor and the reduced PC acceptor. Even if they are born coupled (on a 10 microsecond time scale as discussed above) it is unlikely that they would remain that way for long given the weak exchange coupling and the geometric fluctuations of the ion pair in solvent. It seems to me that the issues saving the day are (a) the substantial driving force for back electron transfer (-2.3 eV based on the reduction potentials of the PC and DIPEA) which would make this firmly in the inverted region and (b) the speed of H-atom abstraction to cement the DIPEA radical cation as an iminium salt. To this latter point, these researchers should be discussing the importance of this aspect of the reaction (its barely mentioned in the modeling in the SI). To me it seems that this chemistry of the sacrificial reductant (and the driving forces of the overall reaction) is essential for high yield given that the initial photoinduced electron transfer from DIPEA to PC* is thermodynamically uphill and given the reduction of the aryl halides are also thermodynamically uphill (as mentioned in the SI).”

The authors effectively address Reviewer #3’s points here regarding driving forces, and they mention a relevant literature example where their mechanistic findings are likely applicable. As Reviewer #3 mentioned concerns regarding PC novelty, I noticed that the PCs in this work were synthesized by the authors, but it was not emphasized whether any of them are new compounds. In fact, there is only a short discussion of these PCs in the main text. Some discussion on the structure property relationships such as the effect of substituents on absorption spectra would be quite interesting and would add to the impact of this work. In addition, comparison of how PC properties might affect the catalytic reaction performance across the PCs would be great to see.

Response: As the reviewer suggested, newly synthesized PCs in the current work have been emphasized in the revised manuscript as follows: “For this reason, in addition to 4DP-IPN, other PCs including five completely new compounds with strongly twisted donor-acceptor structures (mostly, cyanoarene-based PCs) were also investigated to verify the general applicability of our strategy.” and “It should be noted that the PCs prepared here contain five completely new compounds (4DCDP-IPN, 3DP-Cz-IPN, 3DP-DCDP-IPN, 4-p-MCDP-IPN, and 4-o,p-DCDP-IPN).(see Figure 3, legend)”

The structure-property relationships of the substituted compounds, in particular regarding the impact on the absorption spectra and PC activity, are indeed very relevant questions. This seems however to be out of the scope of the current work; in fact, in-depth catalytic, spectroscopic and quantum chemical investigations are currently performed on this issue and will be published in a forthcoming paper.

Nevertheless, we have added the results of newly prepared two cyanoarene-based PCs (i.e. 4-p-MCDP-IPN and 4-o,p-DCDP-IPN) to further support our main argument that triplet excited state generation is an essential factor to form a long-lived radical anion. These two PCs have i) sufficient excited-state oxidizing power to be easily reduced by DIPEA, but ii) no triplet generation. Very interestingly, these PCs did not generate the radical anion in the presence of DIPEA

under blue LED illumination conditions, which clearly indicates the importance of the triplet generation capability for the generation of the radical anion. These results have been properly added in the revised manuscript with Figures 3h and 3i as follows: *“Notably, radical anions were also successfully and efficiently generated for a variety of other organic PCs with donor-acceptor structure as shown in **Figure 3a–g** and **Figure S7**. In all cases, the low-energy absorption bands appear red-shifted compared to those of the PC. These results indicate the broad applicability of our strategy. Furthermore, to verify the PC’s ability to effectively generate triplet excited states upon radical anion formation, two 4DP-IPN analogues with better E_{red}^* (lower HOMO), but negligible T_1 generation, were prepared (**Figure 3h, i** and **Figure S3**). Very interestingly, noticeable radical anion generation was not observed for such PCs in our experimental conditions, clearly confirming that long-lived T_1 generation of PC is crucial for the generation of persistent $PC^{\cdot-}$. Nevertheless, in most strongly-twisted donor-acceptor structures, triplet excited states are efficiently generated; furthermore, by changing the donor and/or acceptor moieties, the redox potentials are delicately controlled in a broad range, allowing to use radical ions with tailored redox potentials for a variety of highly efficient conventional photoredox catalysis, multiphoton excitation catalysis and photoelectrocatalysis.^{8,11}”*

Reviewer #3:

“A third issue of concern to me involves the discussions of ConPET. In the text (line 269) it is stated that addition of bromo anisole to a solution of the PC radical anion results in no color change. This seems to contradict what is seen in Fig S7. Is the reader to draw a distinction from the shades of yellow produced when the bromoanisole is added? To me it looks from the photos more like PC than $PC^{\cdot-}$ and if it is the former then the ConPET is not needed. I am not intending to nit-pick but the issues of mechanism are made to sound simpler than they appear and that detracts from the impact of the work.”

The authors reply here with a series of photos under various conditions. However, it would be beneficial to acquire UV-vis spectra under these conditions instead. As Reviewer #3 mentions, it is difficult to conclude which pictures are different by eye. The authors suggest that this data supports the conPET mechanism. While these observations may be consistent with a conPET pathway in the case of more challenging aryl halides as substrates, the authors’ conclusion would be greatly strengthened by the use of time-resolved measurements to follow the excited state of the radical anion to probe directly whether it reacts with the substrate. In the event access to such techniques is limited, the authors could probe whether slow reactivity with the substrate is observed with the photogenerated radical anion in the absence of light that may also be consistent with the above observations. Potentially, generating the radical anion in larger concentration via bulk electrolysis, turning off the applied potential, and then dosing in substrate in the dark would be a useful experiment. Without some further experimentation, I would be wary of proposing the conPET mechanism in light of the alternate hypothesis that the observed reaction scope may be obtainable via ground state thermal reactivity of the radical anion.

Response: We are sorry for the apparent misunderstanding, which evidently resulted from our insufficient explanation of the experiment. In fact, the experiment we had performed here is conceptually the same experiment that the reviewer suggested. We first generated the radical anion as much as possible with a gentle green LED irradiation in the presence of excess amount of DIPEA. As soon as the radical anion was generated, we turned off the LEDs, followed by the addition of the substrate of interest (i.e. 4-bromobenzonitrile and 4-bromoanisole). As shown in **Figure X3**, drastic

changes were observed for 4-bromobenzonitrile, while no evident color change was observed for 4-bromoanisole. In other words, direct electron transfer between the generated radical anion and 4-bromoanisole is not feasible, which indirectly indicates that the ConPET mechanism might be involved in the dehalogenation of 4-bromoanisole.

As we mentioned in the manuscript, the radical anion is very sensitive to small amounts of oxygen and moisture and thus, all the experiments should be performed in the glovebox. Therefore, it is very difficult to track these color changes by UV-vis in our laboratory because our UV-vis spectrometer is not installed in the glove box. Moreover, the large potential difference between the radical anion of 4DP-IPN and 4-bromoanisole is found, which is hardly overcome at room temperature; this further supports the ConPET process. Also, as far as we understand, the ConPET process was not directly observed yet, which in fact results in many debates on the relevance of the ConPET process in the photoredox catalysis community.

In any case, as the reviewer suggested, the advanced spectroscopic experiments are scheduled to directly observe i) the excited state of radical anion for various cyanoarene-based PCs and ii) their involvement in a ConPET process.

In the revised manuscript, we have properly added several sentences to prevent the confusion of readers as follows: *“In fact, the addition of an excess amount of 4-bromobenzonitrile ($E_{red}^0 = -1.83\text{ V}$) to the preformed solution of 4DP-IPN⁻ gave a clear color change from dark green to yellow in a few seconds, which is indicative of the regeneration of 4DP-IPN, whereas the solution of 4DP-IPN⁻ showed no evident color change in the presence of excess 4-bromoanisole ($E_{red}^0 = -2.70\text{ V}$). This suggests that fast electron transfer between 4DP-IPN⁻ and 4-bromoanisole can be ruled out and a multiphoton excitation process, i.e. ConPET, might be involved.”*

A series of specific comments follow. I will copy here only one which was insufficiently addressed. “Is it possible to use EPR to confirm generation of the PC radical anions? Or does the degradation issues preclude generation of the needed concentrations? Can the PC radical anions be produced chemically allowing correlations to be made between EPR and UV-Vis?”

The authors did not attempt to generate the radical anions via another method or perform EPR measurements. I think that EPR measurement is important to confirm that the red-shifted species are, in fact, radical anions. The UV-vis kinetic measurement suggests that the red-shifted species is sufficiently long-lived for steady state EPR measurements.

Response: Unfortunately, the use of EPR is currently very limited in our institution. In any case, we note that the EPR spectrum of 4DP-IPN⁻ has already been reported by Wickens' group (*Angew. Chem. Int. Ed.* 21418 (2021)). In this work, the radical anion of 4DP-IPN was generated by the electrochemical reduction and the UV-vis and EPR spectra of the resultant radical anion were recorded. As shown in *Figure X4*, the UV-vis spectrum of 4DP-IPN⁻ obtained by Wickens' group was found to be the same as the one obtained by us, which clearly supports the formation of the radical anion of 4DP-IPN.

Figure X4. (a) UV/Vis absorption spectra of 4DP-IPN / 4DP-IPN^{•-}. The generation of 4DP-IPN^{•-} was performed with 4DP-IPN (1.0×10^{-4} M), DIPEA (0.5 M) in CH₃CN under 6W 515 nm LED for 1 min at RT. (b) UV/Vis absorption spectra for 4DP-IPN and 4DP-IPN^{•-} reported by Wickens' group (*Angew. Chem. Int. Ed.* 21418 (2021)). The generation of 4DP-IPN^{•-} was performed with 4DP-IPN (2.0×10^{-2} M), ⁿBu₄PF₆ (0.05 M) in DMF (8 ml) electrolyzed at -1.6 V (vs SCE) at RT. EPR spectra of (c) 4DP-IPN and (d) 4DP-IPN^{•-} confirm that the red-shifted species electrolyzed -1.6 V (vs SCE) have a single unpaired electron. All content of Figure X4 is reproduced from the cited publication, except for Figure X4a.

Lastly, I had a few additional specific comments that came up in reading the manuscript, SI, and responses that I think would be beneficial to this work if they could be addressed.

Figure 1: No reference electrode is given for the potentials reported in this figure and it is unclear how they were calculated/measured or what the sources were for reported values. Reference electrode information is further missing in the text and other figures.

Response: We thank the reviewer for the careful comments. All these contents are now properly reflected in the revised manuscript.

Figure 2: There is a -0.87 V difference between the computed doublet reduction potential and the experimental value which is given for a different excited state. It is also unclear how the experimental value was measured – was this a computational value that is mislabeled? There is also the troubling lack of observation of the computed absorption at ~ 1500 nm. Perhaps further benchmarking of the TD-DFT here with experimental values would be helpful to support the accuracy of the excited state calculations.

Response: We further agree with the reviewer regarding the concerns about the first excited singlet state of the radical anion. We did in fact a benchmarking using functionals that usually operate very differently (as mentioned in fact at the end of the SI); this included the variation of Hartree-Fock exchange (i.e. B3LYP vs. M06HF), as well as long-range corrected functionals (CAM-B3LYP, M06-2X). For all methods, however, the low lying S₁ state was found, varying between 1300-1800 nm. Therefore, we assume that this is not an artifact of the calculation. We note anyway that the

oscillator strength is very small (around 0.005) so that it might be difficult to be detected in the absorption experiment.

The Stern-Volmer plot for Ru(bpy)₃²⁺ shows a smaller slope and yet is assigned a higher rate constant – there must be some error here.

Response: In the Stern-Volmer plot, the rate constant can be evaluated from the slope divided by excited-state lifetime, $\tau_0/\tau = 1 + k_q \tau_0 [Q]$, where τ_0 (s) (or τ) is the excited state lifetime without quencher (or with quencher), k_q ($M^{-1}s^{-1}$) is electron transfer rate constant and $[Q]$ is the concentration of a quencher. Because although the slope of the Stern-Volmer plot for Ru(bpy)₃²⁺ is smaller, the excited state of Ru(bpy)₃²⁺ has a shorter lifetime (0.92 μ s), therefore the rate constant is higher than 4DP-IPN.

The Jablonski diagram contains detailed rate constants from the kinetic simulation. However, it is unclear how the triplet quantum yield was obtained. As far as I can see, no experiments were performed to determine this quantum yield via traditional experimental methods, such as by using nanosecond transient absorption – is it known in the literature?

Response: The triplet (or rather total ISC) quantum yield in a TADF cycle is defined as $\Phi_{ISC} = \eta_{ISC} / (1 - \eta_{ISC} \eta_{RISC})$, where $\eta_{ISC} = k_{ISC} / (k_F + k_{nr,S} + k_{ISC})$ and $\eta_{RISC} = k_{RISC} / (k_{Ph} + k_{nr,T} + k_{RISC})$, see for instance C. Baleizão, M. N. Berberan-Santos, *J. Chem. Phys.* 2007, 126, 204510. All rate constants, given in the Jablonski scheme, were obtained from a kinetic analysis, based on photoluminescence (PL) quantum yields and prompt and delayed PL lifetimes under purged and unpurged conditions at room and low temperatures. This will be reported in detail in a forthcoming paper; a similar procedure on a related compound was reported in S. Tsuchiya et al, *J. Phys. Chem. A* 8074 (2021).

Figure 5: reactions A and B, but not C, lead to formation of a blue emitting species shown in the TLC. This species is not discussed and may be another byproduct derived from the catalyst. Can the authors isolate this spot? Does this species play a role in the reaction?

Response: We are grateful for the reviewer's careful comments on our work. In fact, we were trying to isolate the blue-emitting species observed in the TLC, however, this was not successful due to its extremely small amount. Interestingly, whereas this spot was observed in the reaction of 4-bromobenzonitrile and 4-iodobenzonitrile, the blue-emitting species was not observed in the reaction of 4-bromoanisole and 4-iodoanisole. This implies that the blue species has no catalytic activity and might be a byproduct derived from the benzonitrile radical, but the amount of the blue species is too small to detect them in ¹H NMR.

■ Fate of catalyst during dehalogenation reactions

Figure X5. Fate of catalyst during dehalogenation reactions. Reactions were performed with substrates (0.1 M), DIPEA (10.0 equiv.) and 4DP-IPN (1 mol%, 1.0×10^{-3} M) in CH_3CN (1 ml) under illumination by two 3W 455 nm LED at RT for several hours. Yields were determined by ^1H NMR using 1,3,5-trimethoxybenzene as an internal standard. PC degradation was monitored by in-situ TLC using EA:hexane (1:4 v/v) as an eluent.

Figure 6: The kinetic simulations seem to give very different results from the experimental kinetics here. Can the authors please comment on why this is the case and how the model might be improved? My worry here is that the disagreement may reflect reactivity that is occurring but is not included in the current model.

Response: We agree with the reviewer regarding the concerns about our simulation model. In our simulation model, several assumptions are involved to simplify the model, and the electron transfer from 4DP-IPN⁻ to 4-bromobenzonitrile, k_{ET} , which are hard to be measured experimentally, are calculated from *Marcus–Savéant theory* (as mentioned in the SI). This might explain the disagreement with the experimental kinetics; therefore, the aforementioned information can make the simulation model more accurate.

The other factors are inaccurately evaluated rate constants, especially k_{abs} . k_{abs} represent the rate constant for the transition from $\text{S}_0 \rightarrow \text{S}_1$ when the PC molecule in the ground state absorbs the photon, therefore, the evaluation of photon flux is a key point for the simulation of the excited state population. However, because it is difficult to precisely evaluate all factors affecting photon flux such as a vial's curvature/refractive index, they were not involved in the simulation. To check the effect of k_{abs} in the kinetics, we simulated the kinetics of the reaction by varying k_{abs} (**Figure X6b**). As k_{abs} is decreased by a quarter, the simulated kinetics is close to the experimental kinetics. This implies that to improve our simulation model, it is required to evaluate accurately k_{abs} . All these contents are now properly reflected in the revised SI.

Figure X6. Kinetic simulation of photoredox reductive dehalogenation. (a) Scheme of reaction batch for the reaction and the rate constants for the simulation. (b) Comparison between experimental kinetics and simulated kinetics with varying the rate constants, k_{abs} . All rate constants were obtained from experimental/computational methods in this work or from the literature (please compare the details in the revised SI).

The photoreactor setup does not control the position of the vial, and so the distance from the LED to the reaction solution will vary with each setup. This lack of control may lead to issues with reproducibility and with modeling of the light flux. Setups which control vial position are the standard in the field.

Response: We have now properly added the reproducibility test conducted by others in the revised SI. The yields of dehalogenation were highly reproducible even in the different LED setups implying that our reaction conditions are very mild and user-friendly. Furthermore, considering the manufacturing error of commercial LEDs, the small intensity deviation of LEDs (in this intensity range) could be a minor issue for the dehalogenation.

Table XI. Reproducibility test of dehalogenation

Entry	Setup	Yield (%)
1	Setup 1	100
2		97.8
3	Setup 2	96.1
4		100
5	Setup 3	100
6		100
7	Setup 4	98.2
8		99.4

Reactions were performed with 4-bromobenzonitrile (0.1 M), 4DP-IPN (0.005 mol%), DIPEA (10.0 equiv.) in CH_3CN (1 ml) under two 3W 455 nm LEDs for 8 hours. All solutions were prepared inside glovebox and were degassed by bubbling with Ar. Yields were determined by GC-FID using 1,3,5-trimethoxybenzene as an internal standard.

Throughout the text, the authors use strong language which implies conclusions which are not consistent with the data. In some cases, simply modifying the word choice will help, while in others the conclusions may need to be reevaluated. Here are two examples:

In the last paragraph before the conclusion, the authors state “the solution of 4DP-IPN^{•-} showed no evident color change in the presence of excess 4-bromoanisole ($E_{red}^0 = -2.70$ V). This suggests that no direct electron transfer between 4DP-IPN^{•-} and 4-bromoanisole takes place and a multiphoton excitation process might be involved.”

This statement does not follow from the observation as the alternate hypothesis of a slow, thermal electron transfer from the ground state radical anion could also account for the lack of color change.

Response: We are sorry for the referee’s misunderstanding, caused by experiments performed by us with insufficient control of the reaction conditions. After we increased the concentration of 4DP-IPN^{•-} via longer irradiation, we actually obtained high-quality images (**Figure X3, lower part**). Also, following the reviewer’s suggestion, we have carefully formulated several sentences about semi-quantitative quenching experiments in the revised manuscript as follows: “the solution of 4DP-IPN^{•-} showed no evident color change in the presence of excess 4-bromoanisole ($E_{red}^0 = -2.70$ V). This suggests that fast electron transfer between 4DP-IPN^{•-} and 4-bromoanisole might be ruled out and a multiphoton excitation process might be involved.”

A few sentences down, the authors state “given the low to moderate yields of inactivated aryl chlorides, it seems that the ConPET process is not very efficient here (Table 2). This is presumably due to the short excited-state lifetime of 4DP-IPN^{•-}, which is supported by the fact that steady-state PL emission was not observed for the solution of 4DP-IPN^{•-}”

The lack of observable steady state emission may mean it is difficult to follow the excited state by techniques that depend on emission but cannot inform about the excited state lifetime. For example, the triplet excited states of PCs in this study are long lived yet do not display detectable steady state emission.

Response: We are grateful for the reviewer’s careful comments on our work. We do not observe steady-state emission from the triplet excited state of 4DP-IPN at room temperature, however, the phosphorescence of 4DP-IPN at T = 65 K with a very long lifetime of about 300 ~ 400 ms was observed, and we have published the corresponding spectrum (*Macromolecules* 2019, 52, 5538). On the other hand, in the case of the radical anion of PCs, their excited-state likely has a very short lifetime because of the many possible decay channels such that few photons will be emitted. Currently, experiments are scheduled to try to obtain the emission spectrum of the excited radical anion. This has so far not been reported (as for radical anions in general) and, in case of success, will be reported in a forthcoming publication.

Analysis of the rebuttal submission by Kwon, Wannemacher, and Gierschner and coworkers.

In a short time, the authors have significantly improved their work by disproving their hypothesis that acetonitrile was the methyl radical source experimentally, and they have now proposed a more reasonable scheme involving the radical cation of the base that is supported by new DFT calculations. However, the authors have not obtained experimental data yet to support this new hypothesis. New experiments were performed towards characterization of the methylated PC by mass spectrometry, but I am concerned that the reported error is unusually large. They have also added experiments where they show that a PC that does not efficiently form a triplet excited state will not lead to buildup of the radical anion species in their system. While they have completed commendable work to improve upon the previous submission and made significant progress towards mechanistic understanding, the work remains incomplete in a broader sense. As it stands, the significance (elaborated below) of the work remains unclear in its current form. If this major concern requiring additional experimentation were addressed, I would be supportive of publication, but as it stands I cannot support publication of this work in Nature Communications.

Significance issue: broadly speaking, the major advance in this work in my judgment is the discovery of the degradation pathway involving PC methylation. This judgment is based on two observations concerning the work – first, although dehalogenation of aryl bromides and chlorides is challenging, it has been done in many other systems and with better yield in the case of challenging aryl chlorides (e.g. *J. Am. Chem. Soc.* **2021**, 143, 10882–10889). The ability to use low PC loadings here is interesting but is limited to aryl chlorides that are easier to reduce and aryl bromides. Since many literature reports did not explore the lower limit of PC loading, it is unclear how the results with aryl bromides compare with other methods. Thus, I echo Reviewer #1's sentiment that there are some doubts regarding the synthetic utility of the method. Secondly, without a more complete mechanistic understanding, it is hard to say how the radical anion generation step impacts the catalytic cycle at large. The title of the paper and main focus of the introduction may be misleading in this regard. For example, the title is "Maximizing the formation of strongly reducing radical anions for highly efficient purely organic photoredox catalysis of unactivated substrates". It is unclear how formation of radical anions can be "maximized" as there is no quantitative analysis of their formation or comparison to other systems. The mechanism in the case of unactivated aryl chlorides remains unknown, although conPET is presented as a possibility.

However, the significance of the PC degradation pathway to the field at large remains unclear. This reactivity is currently limited to a single PC which shows decreased catalytic performance. The authors might explore either A) the generality of this new reactivity across other PCs in the same class or B) how the mechanistic knowledge gained by discovering this pathway can improve catalytic performance in hydrodehalogenation. Reorganization of the paper to focus on these new findings and additional experimentation in either of these areas would greatly improve the impact of the current work. In addition, addressing the remaining concerns below would be beneficial.

Previous comments that were insufficiently addressed in the revised version:

Furthermore, I am not convinced that the isolated derivative of 4DP-IPN has the structure the authors assign. While the NMR data is presented in the SI, I am concerned that there is no discussion of the 2D spectra, and the alkyl region of the 1D ¹H NMR spectrum is quite messy. The protons assigned as methyl integrate to only 2.44. Further, if this catalyst derivative is a new

compound, the standard in the field is that its purity should be supported by at least one other technique – typically high resolution mass spectrometry (HRMS) is used. If it is not a new compound, its purity should be compared to reported data in the literature. The UV-vis data and time-resolved/steady state emission data is helpful here in the sense that it confirms the isolated species is distinct from 4DP-IPN and that it reacts with DIPEA at a slower comparative rate. It does not, however, support the asymmetric methyl-substituted structure proposed. Comparison of the UV-vis spectrum of the isolated byproduct with that predicted by TD-DFT could potentially be helpful here if there is good agreement, but I would emphasize that another experimental technique to establish the identity of the isolated species is necessary (HRMS or single crystal X-ray diffraction would be ideal). Finally, the degradation pathway is a highly interesting and important discovery, in my opinion vital this work, and its importance might be better communicated to the reader if it were added to the proposed mechanistic scheme in the main text.

Response: As the reviewer suggested, MALDI-TOF and DART HRMS experiments were performed, which further clarify the structural assignment of 4DP-Me-BN (see Figure X3a). In order to prevent confusion of the readers, the detailed structural assignment by 1D and 2D ¹H NMR (COSY, NOESY) of the bleached product was provided in the revised SI as shown in Figure X3b (Figure S15 in the revised SI). Also, following the reviewer's suggestion, we have added several sentences about the proposed mechanistic scheme in the revised manuscript as follows: "To validate our hypothesis, the most possible reaction pathway was screened by DFT calculations (Figure 4b). The addition of CH₃• to PC radical anion was found to be a rate-determining step with a surmountable activation energy of 22.1 kcal/mol. In particular, this radical addition reaction provided an energetically stable intermediate with 8.3 kcal/mol lower energy than that of the PC radical anion, which confirms the feasibility of this reaction pathway. Further in-depth studies are currently underway to understand the full mechanistic pathways of the 4DP-IPN photodegradation including the formation of CH₃•."

The assignment of NMR spectra alleviated most of my previous concerns, however the new HRMS data raises similar questions. Performing HRMS is a step in the right direction, but the difference between the calculated and found masses of ionized species is 3-4 orders of magnitude larger than typically reported in the literature. This is true for all the new HRMS data added to the SI.

The authors have done an excellent job in conceiving a more reasonable hypothesis concerning DIPEA as the methyl radical source. However, experimental support is needed to test the predictions afforded by DFT. Have the authors tried switching the base to an alternate base that would yield a different radical (e.g. ethyl radical) that would result in a different PC derivative?

Previous comment:

The authors reply here with a series of photos under various conditions. However, it would be beneficial to acquire UV-vis spectra under these conditions instead. As Reviewer #3 mentions, it is difficult to conclude which pictures are different by eye. The authors suggest that this data supports the conPET mechanism. While these observations may be consistent with a conPET pathway in the case of more challenging aryl halides as substrates, the authors' conclusion would be greatly strengthened by the use of time-resolved measurements to follow the excited state of the radical anion to probe directly whether it reacts with the substrate. In the event access to such techniques is limited, the authors could probe whether slow reactivity with the substrate is

observed with the photogenerated radical anion in the absence of light that may also be consistent with the above observations. Potentially, generating the radical anion in larger concentration via bulk electrolysis, turning off the applied potential, and then dosing in substrate in the dark would be a useful experiment. Without some further experimentation, I would be wary of proposing the conPET mechanism in light of the alternate hypothesis that the observed reaction scope may be obtainable via ground state thermal reactivity of the radical anion.

Paragraph 2 of Author's response:

As we mentioned in the manuscript, the radical anion is very sensitive to small amounts of oxygen and moisture and thus, all the experiments should be performed in the glovebox. Therefore, it is very difficult to track these color changes by UV-vis in our laboratory because our UV-vis spectrometer is not installed in the glove box. Moreover, the large potential difference between the radical anion of 4DP-IPN and 4-bromoanisole is found, which is hardly overcome at room temperature; this further supports the ConPET process. Also, as far as we understand, the ConPET process was not directly observed yet, which in fact results in many debates on the relevance of the ConPET process in the photoredox catalysis community.

Few research groups have an air-free UV-vis setup, yet UV-vis of oxygen and/or moisture sensitive compounds is commonly done. Solutions are prepared in the glovebox in a cuvette that can be sealed and are then removed and placed into the spectrometer. The authors appear to have already done this in Fig. 2d, so it would be straightforward to do the same experiment but with added 4-bromoanisole. Since the difference (if there is one) between the two cases (with/without substrate) might be quite subtle and therefore difficult to detect by eye, monitoring with a more precise method such as UV-vis is likely to be important.

Previous comment:

Figure 6: The kinetic simulations seem to give very different results from the experimental kinetics here. Can the authors please comment on why this is the case and how the model might be improved? My worry here is that the disagreement may reflect reactivity that is occurring but is not included in the current model.

Response paragraph 2:

The other factors are inaccurately evaluated rate constants, especially k_{abs} . k_{abs} represent the rate constant for the transition from $S_0 \rightarrow S_1$ when the PC molecule in the ground state absorbs the photon, therefore, the evaluation of photon flux is a key point for the simulation of the excited state population. However, because it is difficult to precisely evaluate all factors affecting photon flux such as a vial's curvature/refractive index, they were not involved in the simulation. To check the effect of k_{abs} in the kinetics, we simulated the kinetics of the reaction by varying k_{abs} (Figure X6b). As k_{abs} is decreased by a quarter, the simulated kinetics is close to the experimental kinetics. This implies that to improve our simulation model, it is required to evaluate accurately k_{abs} . All these contents are now properly reflected in the revised SI.

Measurement of light flux in a particular reaction setup is commonly done by chemical actinometry. For an example, see (*J. Am. Chem. Soc.* 2020, 142, 37, 15830–15841). Perhaps measurement of this quantity could improve the model.

Reviewer #4 (Remarks to the Author):

Importantly, the model predicts linear behavior in the conversion vs. time plot, while the actual behavior is nonlinear (Fig. X6b). This discrepancy may suggest a difference in the turnover-limiting step between the model and the experimental mechanism.

Reviewers' comments to author:

We are grateful to the reviewers for their constructive comments and suggestions, which greatly helped to clarify a number of important points, avoid misunderstandings, and improve the clarity and referencing of the paper. We have carefully revised our manuscript in line with all the reviewer's comments. The point-by-point response is given below. Our responses to reviewer's comments are highlighted in *blue color* with a highlight in the revised manuscript and SI.

Reviewer #4 (Remarks to the Author):

Analysis of the rebuttal submission by Kwon, Wannemacher, and Gierschner and coworkers.

In a short time, the authors have significantly improved their work by disproving their hypothesis that acetonitrile was the methyl radical source experimentally, and they have now proposed a more reasonable scheme involving the radical cation of the base that is supported by new DFT calculations. However, the authors have not obtained experimental data yet to support this new hypothesis. New experiments were performed towards characterization of the methylated PC by mass spectrometry, but I am concerned that the reported error is unusually large. They have also added experiments where they show that a PC that does not efficiently form a triplet excited state will not lead to buildup of the radical anion species in their system. While they have completed commendable work to improve upon the previous submission and made significant progress towards mechanistic understanding, the work remains incomplete in a broader sense. As it stands, the significance (elaborated below) of the work remains unclear in its current form. If this major concern requiring additional experimentation were addressed, I would be supportive of publication, but as it stands I cannot support publication of this work in Nature Communications.

Response: We thank the reviewer for careful comments and constructive suggestions, which definitely help us to present our work more clearly. The point-by-point response are below and our responses are highlighted in *blue color* in the revised manuscript and SI.

Significance issue: broadly speaking, the major advance in this work in my judgment is the discovery of the degradation pathway involving PC methylation. This judgment is based on two observations concerning the work – first, although dehalogenation of aryl bromides and chlorides is challenging, it has been done in many other systems and with better yield in the case of challenging aryl chlorides (e.g. J. Am. Chem. Soc. 2021, 143, 10882–10889). The ability to use low PC loadings here is interesting but is limited to aryl chlorides that are easier to reduce and aryl bromides. Since many literature reports did not explore the lower limit of PC loading, it is unclear how the results with aryl bromides compare with other methods. Thus, I echo Reviewer #1's sentiment that there are some doubts regarding the synthetic utility of the method.

Response: Since we agree with the reviewer's comments, we have substantially revised the manuscript following the reviewer's suggestion. In particular, the sentences related to the catalyst performance have been reduced in the introduction and conclusion of the manuscript. Instead, we emphasized i) the role of triplet excited states of cyanoarene-based PCs on electron transfer and radical anion ($PC^{\cdot-}$) formation and ii) the degradation behaviors of $PC^{\cdot-}$, in the revised manuscript, based on the additional intensive experiments about the photodegradation of the PCs (*see below*).

Secondly, without a more complete mechanistic understanding, it is hard to say how the radical anion generation step impacts the catalytic cycle at large. The title of the paper and main focus of the introduction may be misleading in this regard. For example, the title is “Maximizing the formation of strongly reducing radical anions for highly efficient purely organic photoredox catalysis of unactivated substrates”. It is unclear how formation of radical anions can be “maximized” as there is no quantitative analysis of their formation or comparison to other systems. The mechanism in the case of unactivated aryl chlorides remains unknown, although conPET is presented as a possibility.

Response: Following the reviewer’s comments, we removed the word “Maximizing” from the title. In the revised manuscript, the title has been changed to “Formation and degradation of strongly reducing radical anions for highly efficient purely organic photoredox catalysis”. We also removed a substantial amount of sentences related to “ConPET” in the introduction and conclusion of the revised manuscript because the photocatalytic mechanism of dehalogenation of unactivated aryl chlorides still remains largely unclear and is still being debated in the community, as the reviewer pointed out.

However, the significance of the PC degradation pathway to the field at large remains unclear. This reactivity is currently limited to a single PC which shows decreased catalytic performance. The authors might explore either A) the generality of this new reactivity across other PCs in the same class or B) how the mechanistic knowledge gained by discovering this pathway can improve catalytic performance in hydrodehalogenation. The reorganization of the paper to focus on these new findings and additional experimentation in either of these areas would greatly improve the impact of the current work. In addition, addressing the remaining concerns below would be beneficial.

Response: Following the reviewer’s suggestion, we intensively investigated the degradation pathway of a total of 9 different cyanoarene-based PCs. As described in **Figure XI**, very interesting degradation patterns were observed, which relies on the electronic properties (*i.e.* capability for the radical anion generation) and structural features (*i.e.* steric environments nearby the CN group) of PCs; detailed information on the degradation behavior of PCs are presented at the end of the response letters and the revised supplementary information.

As in 4DP-IPN, methyl substitution at one of the two CN groups was observed for 3DP-Cz-IPN and 3DP-DCDP-IPN which have similar electronic and structural properties to 4DP-IPN (**Figure XIa**). However, in 4Cz-IPN and 4tCz-IPN, C₂H₅ substitution occurred with a small amount of CH₃ substitution, suggesting that the steric environment near the CN group is a crucial factor in the substitution reaction (**Figure XIb**). The ethyl group here is likely provided by the C–N bond cleavage of DIPEA⁺ assisted by a 1,2-methyl shift (see the proposed mechanism illustrated in **Figure XIb**). Indeed, the use of diisopropylmethylamine (DIPMA) instead of DIPEA resulted in the generation of a CH₃-substituted adduct as a major product, which clearly supports our hypothesis.

Figure XIc shows the results of photodegradation experiments for the PCs in which PC^{•-} was not properly formed. No photodegradation was observed for 3DP-DMDP-IPN and 4-o,p-DCDP-IPN whereas complex degradation mixtures were formed in 4-p-MCDP-IPN. These results imply that, in such PCs, there is no well-defined degradation pathway through the PC^{•-} intermediate, and thus the photodegradation behavior is determined by the intrinsic photostability of the PCs. Finally, we examined the photodegradation behavior of 3DP-F-IPN and 4-p,p-DCDP-IPN that nicely generate PC^{•-} and also, contain additional labile groups such as C–F bond or other types of C–CN bonds. Complex reaction

mixtures were formed for both PCs, which is likely due to the degradation of the aforementioned labile groups (*Figure XI*).

Figure XI. Photodegradation behaviors of various cyanoarene-based PCs. For the characterizations of isolated products, see the supplementary information.

(a) Photodegradation behavior of 3DP-Cz-IPN and 3DP-DCDP-IPN. (b) Photodegradation behaviors of 4Cz-IPN and 4tCz-IPN and the proposed mechanism of photodegradation in the presence of DIPEA and DIPMA as a reducing agent. (c) Photodegradation behaviors of PCs non-generating PC⁻. (d) Photodegradation behaviors of PCs with labile groups.

Based on these results, we have drawn the following conclusions: i) CN substitution reaction takes place via a PC⁻ as an intermediate, ii) Substituent is determined by the steric environment near the CN group of PC⁻ and the type of Hünig's base, iii) for PCs in which PC⁻ was not properly formed, the photodegradation behavior is determined by the intrinsic photostability of the PCs. These results clearly support that the CN substitution reaction generally occurs for

PCs satisfying specific criteria. Our new results about the photodegradation of cyanoarene-based PCs in the presence of Hünig's base have been properly added to the introduction, results, and conclusion of the revised manuscript.

Previous comments that were insufficiently addressed in the revised version:

The assignment of NMR spectra alleviated most of my previous concerns, however, the new HRMS data raises similar questions. Performing HRMS is a step in the right direction, but the difference between the calculated and found masses of ionized species is 3-4 orders of magnitude larger than typically reported in the literature. This is true for all the new HRMS data added to the SI.

Response: We are sorry for our incorrect data, which is partly due to miscommunication with an operator. Original values were obtained from direct analysis in real time (DART) without having a high-resolution detector. We thus reperformed the HRMS analyses (GC-FAB-HRMS) for all prepared PCs and their photodegradation adducts. New values are nicely matched with theoretical values, as described in *Figure X2*.

Among the cyanoarene-based molecules, HRMS data were not obtained for 4-p-MCDP-IPN and 4-o,p-DCDP-IPN even changing the ionization methods. These results are likely due to the instability of the PC cation. In any case, other structural characterization methods including COSY, NOESY, and ^{19}F -NMR, were carefully conducted, which clearly confirms the structure of 4-p-MCDP-IPN and 4-o,p-DCDP-IPN.

Figure X2. High-resolution mass spectrometry (HRMS) of the cyanoarene-based PCs studied in the current work. The HRMS of PCs were obtained gas chromatography equipped with fast atom bombardment (FAB) ionization method. See the supplementary information in the end of response letter for more HRMS data of newly synthesized and photodegraded products.

The authors have done an excellent job in conceiving a more reasonable hypothesis concerning DIPEA as the methyl radical source. However, experimental support is needed to test the predictions afforded by DFT. Have the authors tried switching the base to an alternate base that would yield a different radical (e.g. ethyl radical) that would result in a different PC derivative?

Response: We thank the reviewer's suggestion. To experimentally support our hypothesis, several additional experiments were performed in the revision as follows: i) analyses of photodegradation of 4DP-IPN and ii) extensive investigation of photodegradation behavior of various cyanoarene-based PCs and photodegradation studies about 4Cz-IPN.

i) Analyses of photodegradation of 4DP-IPN

Figure X3. (a) Photodegradation behavior of 4DP-IPN in the presence of DIPEA. Reactions were performed with 4DP-IPN (1.0×10^{-4} M) and DIPEA (0.5 M) in CH_3CN under the illumination of two 3W 515 nm LEDs or two 3W 455 nm LEDs at RT. PC degradations were monitored in-situ by TLC with eluent conditions, CH_2Cl_2 :hexane (7:3 v/v). The photodegraded products were successfully isolated by column chromatography, which gives ^1H NMR spectra confirming that a methyl (or hydrogen) substitution reaction occurred at the CN position of 4DP-IPN to generate 4DP-Me-BN (or 4DP-H-BN). (b) Proposed mechanistic pathway for photodegradation behavior of 4DP-IPN in the presence of DIPEA and DFT calculations for the bond dissociation energies (ΔH) in $\text{DIPEA}^{+\bullet}$; in parentheses, the bond dissociation energies in DIPEA were calculated.

Based on the reviewer's comment, we have been rethinking the mechanism by which DIPEA participates in the reaction. Since DIPEA⁺ has methyl as well as hydrogen in its β-position (see **Figure X3**), the H-substituted adduct of 4DP-IPN should also be formed along with 4DP-Me-BN. We carefully rechecked the TLC obtained from the photodegradation reaction performed under 455 nm LED illuminations and found a pale green spot just below the spot assigned with 4DP-Me-BN (see **Figure X3**). We first scaled up the reaction to isolate the pale green spot and, fortunately, it was successfully isolated by column chromatography. Our intensive NMR analyses combined with HRMS clearly confirmed that a hydrogen substitution reaction also happened at the CN position of 4DP-IPN to produce 4DP-H-BN. As can be seen from TLC, a very small amount of 4DP-H-BN was produced compared to 4DP-Me-BN. These results are the strong pieces of evidence that DIPEA participates in the degradation of the PC. We have properly included our new results in the revised manuscript.

ii) Extensive investigation of photodegradation behavior of various cyanoarene-based PCs and photodegradation studies about 4Cz-IPN.

We extended our investigations to other cyanoarene-based PCs. As we stated above, very interesting patterns were seen for the degradation, which greatly depends on the electronic properties (capability for the radical anion generation) and structural features (steric environments nearby the CN group) of PCs. Of those, we were concerned about the photodegradation behavior of 4Cz-IPN. Different from 4DP-IPN, C₂H₅ substitution mostly happened with a small amount of CH₃ substitution (see **Figure X4** and **Figure SX8**). As described in **Figure X4**, the ethyl group is likely introduced by the C–N bond cleavage of DIPEA⁺ assisted by a 1,2-methyl shift; it should be noted that the ethyl group cannot be provided from the solvent (CH₃CN). To support our argument, we changed DIPEA with diisopropylmethylamine (DIPMA) where the ethyl group in DIPEA is replaced with a methyl group (see **Figure SX9**). Indeed, the use of DIPMA resulted in the formation of a CH₃-substituted adduct as a major product. These results strongly support our hypothesis that DIPEA⁺ could be a source of methyl substitution of 4DP-IPN. In any case, we are currently investigating more in-depth studies to fully understand the photodegradation of cyanoarene-based PCs, which is beyond the scope of this work.

Figure X4. Photodegradation behaviors of 4Cz-IPN and the proposed mechanism of photodegradation in the presence of DIPEA as a reducing agent.

Few research groups have an air-free UV-vis setup, yet UV-vis of oxygen and/or moisture sensitive compounds is commonly done. Solutions are prepared in the glovebox in a cuvette that can be sealed and are then removed and placed into the spectrometer. The authors appear to have already done this in Fig. 2d, so it would be straightforward to do the same experiment but with added 4-bromoanisole. Since the difference (if there is one) between the two cases (with/without substrate) might be quite subtle and therefore difficult to detect by eye, monitoring with a more precise method such as UV-vis is likely to be important.

Response: Following the reviewer's comments, we carefully performed the suggested experiments. As illustrated in **Figure X5b**, UV/vis spectra were successfully recorded for both samples, which well corresponds to the results obtained by the naked eye. These results have properly been added to the revised supplementary information.

Figure X5. Evaluation of PET process in photoredox reductive dehalogenation. (a) Inside a glovebox using flame-dried glass vials, the radical anion of 4DP-IPN (1.0×10^{-3} M) was generated with DIPEA (0.5 M) in CH_3CN (2 ml) under illumination of a 3W 515 nm LED for 3 min, aryl bromides solution (0.1 M) as quencher in CH_3CN (0.2 ml) were added. Subsequently, the added solutions were re-illuminated by a 3W 515 nm LED for 3 min. All pictures were taken immediately without any additional delay. (b) UV/Vis absorption spectra of 4DP-IPN $^{\bullet-}$ (1.0×10^{-4} M; orange line) in the presence of aryl halide (1.0×10^{-2} M) in CH_3CN generated with DIPEA (0.5 M) in CH_3CN under illumination of two 3W 515 nm LEDs for 3 min.

Measurement of light flux in a particular reaction setup is commonly done by chemical actinometry. For an example, see (J. Am. Chem. Soc. 2020, 142, 37, 15830–15841). Perhaps measurement of this quantity could improve the model. Importantly, the model predicts linear behavior in the conversion vs. time plot, while the actual behavior is nonlinear (Fig. X6b). This discrepancy may suggest a difference in the turnover-limiting step between the model and the experimental mechanism.

Response: We thank the reviewer's critical comments. After we carefully reexamined the equations and hypotheses used in the model, we realized that discrepancies found in the kinetics simulations are likely due to the fact that electron transfer (ET) between $\text{PC}^{\bullet-}$ and the substrate (4-bromobenzonitrile) was overestimated and back electron transfer (BET) between $\text{PC}^{\bullet-}$ and $\text{DIPEA}^{\bullet+}$ was not considered in the model. In fact, as shown in **Figure X6**, the nonlinearity seen in the experimental kinetics was nicely reproduced in the simulations by the consideration of the BET at slower ET; here, a photodiode from Thorlab was used to measure the light flux to evaluate k_{abs} .

b) ■ Simulated kinetics varying k_{ET} with BET ($k_{BET} = 1.0 \times 10^{10} M^{-1}s^{-1}$)

c) ■ Proposed mechanism of consumption of tertiary amines by singlet oxygen (1O_2)

Entry	Reaction conditions	Yield (%)
1	DIPEA (2.0 equiv.) under air	0
2	DIPEA (5.0 equiv.) under air	12
3	DIPEA (10.0 equiv.) under air	84

Figure X6. Kinetic simulation of photoredox reductive dehalogenation. (a) Scheme of reaction batch for the reaction and the rate constants for the simulation. (b) Comparison between experimental kinetics and simulated kinetics with k_{BET} ($1.0 \times 10^{10} M^{-1}s^{-1}$) varying the k_{ET} . (c) Proposed mechanism of consumption of tertiary amines by singlet oxygen (1O_2). Oxygen tolerance in the dehalogenation reaction with various amount of DIPEA is also indicated (right); Reactions were performed with 4-bromobenzonitrile (0.1 M), 4DP-IPN (0.03 mol%), DIPEA (2.0-10.0 equiv.) in CH_3CN (1 ml) under two 3W 455 nm LEDs for 3 hours under air atmosphere in the closed glass vial. All solutions were prepared outside under ambient conditions. Yields were determined by 1H NMR using 1,3,5-trimethoxybenzene as an internal standard.

Figure X7. Summary of the experimental and calculated values to obtain the rate constant, k_{ET} , for electron transfer from 4DP-IPN $^-$ to 4-bromobenzonitrile; molecular radii of each species are indicated under the chemical structures, Z is the pre-exponential factor, ΔG^\ddagger is the activation energy of the reaction, and λ_0 is the reorganization energy.

i) Issues for the overestimation of ET between PC $^-$ and bromobenzonitrile

Since PC^- is non-emitting, it is difficult to experimentally measure the rate of ET (k_{ET}) between PC^- and the substrate through the Stern-Volmer equation. Therefore, in the original manuscript, k_{ET} was obtained by the speculation from the electron transfer of the $^3PC^*$ and substrate. As illustrated in **Figure X7**, collision frequency (Z), reorganization energy (λ_o), and thermodynamic driving force (ΔG°) were required for this calculation. Since the structure of the Arrhenius equation, small errors in λ_o and ΔG° might generate a large difference in the value of k_{ET} . However, it is very difficult to obtain these values very accurately by experiments, which is one of the unsolved important problems in the field. In any case, we will try to obtain the k_{ET} value from an advanced experimental technique (*i.e.* transient absorption electronic/vibrational spectroscopy) as a follow-up work, which we believe the beyond the scope of this work.

Figure X8. Scheme of photoinduced electron transfer (PET) and back electron transfer (BET) process via singlet/triplet contact radical ion pair.

ii) Issues for BET between PC^- and $DIPEA^+$

In fact, in the original manuscript, we completely ignored the BET between PC^- and $DIPEA^+$. However, during the revision, we found that our hypothesis did not match what we observed (see **Figure 2d** in the original manuscript). In the dark conditions, the produced radical anions disappeared over time, clearly indicating the existence of BET between PC^- and $DIPEA^+$. In fact, it has been reported that several organic dyes in the triplet excited state have the k_{BET} value close to the diffusion limit (approximately $10^9 \sim 10^{11} \text{ M}^{-1}\text{s}^{-1}$; *J. Photochem. Photobiol. C*, **5**, 79-104 (2004), *J. Chem. Soc., Faraday Trans.*, **93**, 4275-4279 (1997), *J. Am. Chem. Soc.*, **96**, 4710-4712 (1974), and *Journal of Porphyrins and Phthalocyanines*, **15**, 111-117 (2011)), which is different from that many singlet PCs have a BET faster than the diffusion limit. Therefore, k_{BET} value was assumed to be $10^{10} \text{ M}^{-1}\text{s}^{-1}$ for the model, resulting in good agreement between simulation and experimental results. Based on this, the PET and BET involving the triplet excited states of the PC can be visualized in **Figure X8** for a better understanding of the reviewer (and the potential readers).

However, significant discrepancies were observed for dehalogenation reactions in the presence of oxygen (see **Figure X6b**). This is probably due to the degradation of the tertiary amines (sacrificial reducing agents) by reactive oxygen species as illustrated in **Figure X6b**, which is not considered in the model; indeed, under air atmosphere, the dehalogenation reaction was found to be very sensitive to the amount of DIPEA (**Figure X6c**). As the reviewer pointed out, our model should be improved to perfectly describe the experiments. For this, we are currently trying to i) accurately measure the rate constants including k_{ET} , k_{BET} , etc, and ii) achieve a complete understanding of the mechanism of the reductive dehalogenation reactions (e.g. the degradation of catalysts and sacrificial reagents, the contribution of XAT process, etc.).

Supplementary information for response letter:

■ Synthetic scheme for cyanoarene-based PC

■ $1D$ ^{19}F NMR spectrum of 3DP-F-IPN (377 MHz, DMSO- d_6)

■ $1D$ ^{19}F NMR spectrum of 4-p-MCDP-IPN (565 MHz, DMSO- d_6)

■ $1D$ ^{19}F NMR spectrum of 4-o,p-DCDP-IPN (565 MHz, acetone- d_6)

Figure SX1. Synthetic scheme of cyanoarene-based on PCs and $1D$ ^{19}F NMR spectra of 3DP-F-IPN, 4-p-MCDP-IPN and 4-o,p-DCDP-IPN. Because of the absence of peak, it was confirmed that 4-p-MCDP-IPN and 4-o,p-DCDP-IPN were fully substituted by amine groups.

a) ■ Cyanoarene-based PCs with more steric-hindrance

e.g.

b) ■ Photodegradation behavior of 3DP-Cz-IPN

c) ■ Photodegradation behavior of 3DP-DCDP-IPN

Figure SX2. (a) Photodegradation behaviors of cyanoarene-based PCs with more steric-hindrance. The geometry of radical anions of 4DP-IPN was obtained by DFT calculation, B3LYP/6-311++G*, in CH_3CN with PCM solvation model. In the presence of DIPEA, photodegradation behavior of (b) 3DP-Cz-IPN and (c) 3DP-DCDP-IPN. PC degradations were monitored in-situ by TLC (EA:hexane = 1:2 v/v for 3DP-Cz-IPN and EA:hexane = 1:1 v/v for 3DP-DCDP-IPN); left/right spots were collected at before/after irradiation and middle spot was co-spot.

a) ■ Cyanoarene-based PCs with less steric-hindrance

b) ■ Photodegradation behavior of 4tCz-IPN

c) ■ Photodegradation behavior of 4Cz-IPN

d) ■ Photodegradation behavior of 4Cz-IPN with DIPMA

Figure SX3. (a) Photodegradation behaviors of cyanoarene-based PCs with less steric-hindrance. The geometry of radical anions of 4Cz-IPN was obtained by DFT calculation, B3LYP/6-311++G*, in CH₃CN with PCM solvation model. In the presence of DIPEA, Photodegradation behavior of (b) 4tCz-IPN and (c) 4Cz-IPN. (d) Photodegradation behavior of 4Cz-IPN using DIPMA instead of DIPEA. PC degradations were monitored in-situ by TLC (EA:hexane = 5:95 v/v for 4tCz-IPN and EA:hexane = 1:1 v/v for 4Cz-IPN); left/right spots were collected at before/after irradiation and middle spot was co-spot. The characterization of isolated products is in supplementary information (see **Figure SX3-SX4**).

a) ■ Photodegradation behaviors for PC with labile group

b) ■ Photodegradation behaviors for PC non-generating PC⁻

Figure SX4. (a) Photodegradation behavior of PCs with labile group, 3DP-F-IPN and 4-p,p-DCDP-IPN. Reactions were performed with PC (1.0×10^{-4} M) and DIPEA (0.5 M) in CH_3CN under the illumination of two 3W 455 nm LEDs for 2 hours at RT. PC degradations were monitored in-situ by TLC (EA:hexane=1:6 v/v for 3DP-F-IPN and acetone:hexane=2:3 v/v for 4-p,p-DCDP-IPN); left/right spots were collected at before/after irradiation and middle spot was co-spot. (d) Photodegradation behavior of PCs non-generating PC⁻, 3DP-DMDP-IPN, 4-p-MCDP-IPN and 4-o,p-DCDP-IPN. Reactions were performed with PC (1.0×10^{-4} M) and DIPEA (0.5 M) in CH_3CN under the illumination of two 3W 455 nm LEDs for 2 hours at RT. PC degradations were monitored in-situ by TLC (EA:hexane=1:1 v/v for 3DP-DMDP-IPN, EA:hexane=7:3 v/v for 4-p-MCDP-IPN and MeOH: CHCl_3 =1:4 v/v for 4-o,p-DCDP-IPN); left/right spots were collected at before/after irradiation and middle spot was co-spot.

a) ■ Photodegradation behavior of 3DP-Cz-IPN

b) ■ HRMS spectrum of 3DP-Cz-Me-BN

■ 1D ¹H NMR spectrum of 3DP-Cz-Me-BN

■ 2D COSY ¹H NMR spectrum

■ 2D NOESY ¹H NMR spectrum

Figure SX5. (a) Photodegradation behavior of 3DP-Cz-IPN. Reactions were performed with PC (1.0×10^{-4} M) and DIPEA (0.5 M) in CH₃CN under the illumination of two 3W 455 nm LEDs for 2 hours at RT. PC degradations were monitored in-situ by TLC (EA:hexane=1:2 v/v); left/right spots were collected at before/after irradiation and middle spot was co-spot. The photodegraded products were successfully isolated by column chromatography, which gives ¹H NMR spectra confirming that a methyl substitution reaction occurred at the CN position. (b) Structural analysis of 3DP-Cz-Me-BN with 2D ¹H NMR analyses combined with HRMS, MS (GC-FAB-HRMS): calc'd for C₅₆H₄₂N₅ [M+H]⁺: 784.3440; found as 784.3440. ¹H NMR (400 MHz, CDCl₃) δ 7.60–7.51 (m, 2H), 7.35 (t, 4H), 7.16 (d, 4H), 7.04 (t, 2H), 6.94–6.87 (m, 4H), 6.81–6.68 (m, 10H), 6.53 (td, 4H), 6.43 (d, 4H), 6.37 (d, 4H), 1.65 (s, 3H).

a) Photodegradation behavior of 3DP-DCDP-IPN

b) HRMS spectrum of 3DP-DCDP-Me-BN

1D ¹H NMR spectrum of 3DP-DCDP-Me-BN

2D COSY ¹H NMR spectrum

2D NOESY ¹H NMR spectrum

Figure SX6. (a) Photodegradation behavior of 3DP-DCDP-IPN. Reactions were performed with PC (1.0×10^{-4} M) and DIPEA (0.5 M) in CH₃CN under the illumination of two 3W 455 nm LEDs for 2 hours at RT. PC degradations were monitored in-situ by TLC (EA:hexane=1:1 v/v); left/right spots were collected at before/after irradiation and middle spot was co-spot. The photodegraded products were successfully isolated by column chromatography, which gives ¹H NMR spectra confirming that a methyl substitution reaction occurred at the CN position. (b) Structural analysis of 3DP-DCDP-Me-BN with 2D ¹H NMR analyses combined with HRMS, MS (GC-FAB-HRMS): calc'd for C₅₈H₄₂N₇ [M+H]⁺: 836.3502; found as 836.3497. ¹H NMR (400 MHz, CDCl₃) δ 7.29 (t, 4H), 7.17 (d, 4H), 7.13–6.98 (m, 14H), 6.87 (td, 4H), 6.67–6.56 (m, 12H), 1.58 (s, 3H).

a) Photodegradation behavior of 4tCz-IPN

b) HRMS spectrum of 4tCz-Me-BN and 4tCz-Et-BN mixture

c) 1D ¹H NMR spectrum of mixture of 4tCz-Me-BN and 4tCz-Et-BN

2D COSY ¹H NMR spectrum

2D NOESY ¹H NMR spectrum

Figure SX7. Photodegradation behavior of 4tCz-IPN. (a) Reactions were performed with PC (1.0×10^{-4} M) and DIPEA (0.5 M) in CH_3CN under the illumination of two 3W 455 nm LEDs for 2 hours at RT. PC degradations were monitored in-situ by TLC (EA:hexane=5:95 v/v); left/right spots were collected at before/after irradiation and middle spot was co-spot. (b, c) The mixture of photodegraded products were successfully isolated by column chromatography, which gives ^1H NMR spectra confirming that a methyl (or ethyl) substitution reaction occurred at the CN position combined with HRMS, MS (GC-FAB-HRMS): calc'd for 4tCz-Me-BN ($\text{C}_{88}\text{H}_{99}\text{N}_5$) $[M]^+ : 1225.7900$; found as 1225.7892, calc'd for 4tCz-Et-BN ($\text{C}_{89}\text{H}_{101}\text{N}_5$) $[M]^+ : 1239.8057$; found as 1239.8086.

Figure SX8. Photodegradation behavior of 4Cz-IPN. (a) Reactions were performed with PC (1.0×10^{-4} M) and DIPEA (0.5 M) in CH_3CN under the illumination of two 3W 455 nm LEDs for 2 hours at RT. PC degradations were monitored in-situ by TLC (EA:hexane=1:4 v/v); left/right spots were collected at before/after irradiation and middle spot was co-spot. The photodegraded products were successfully isolated by column chromatography, which gives 1H NMR spectra confirming that an ethyl substitution reaction occurred at the CN position. (b) Structural analysis of 4Cz-Et-BN with 2D 1H NMR analyses combined with HRMS, MS (GC-FAB-HRMS): calc'd for $C_{57}H_{38}N_5$ $[M+H]^+$: 792.3127; found as 792.3129.

Figure SX9. Photodegradation behavior of 4Cz-IPN. (a) Reactions were performed with PC (1.0×10^{-4} M) and DIPMA (0.5 M) in CH_3CN under the illumination of two 3W 455 nm LEDs for 2 hours at RT. PC degradations were monitored in-situ by TLC (EA:hexane=1:4 v/v); left/right spots were collected at before/after irradiation and middle spot was co-spot. The photodegraded products were successfully isolated by column chromatography, which gives ^1H NMR spectra confirming that a methyl substitution reaction occurred at the CN position. (b) Structural analysis of 4Cz-Me-BN with 2D ^1H NMR analyses combined with HRMS, MS (GC-FAB-HRMS): calc'd for $\text{C}_{56}\text{H}_{36}\text{N}_5$ $[\text{M}+\text{H}]^+$: 778.2971; found as 778.2980.

Figure SX10. Structural characterization of 4DP-Me-BN. Structural analysis of 4DP-Me-BN with (a) 1D ¹H NMR analyses combined with HRMS, MS (GC-FAB-HRMS): calc'd for C₅₆H₄₄N₅ [M+H]⁺: 786.3597; found as 786.3600, and (b) intense 2D NMR analyses including COSY, NOESY, HSQC and HMBC NMR. ¹H NMR (600 MHz, CDCl₃) δ 7.30–7.25 (t, 4H), 7.11–7.02 (m, 12H), 6.96 (t, 2H), 6.86–6.78 (m, 8H), 6.73–6.69 (d, 4H), 6.69–6.65 (d, 4H), 6.60 (t, 2H), 6.55–6.49 (d, 4H), 1.58 (s, 3H). ¹³C NMR (151 MHz, CDCl₃) δ 150.15, 148.35, 146.81, 145.76, 145.15, 144.58, 144.56, 142.97, 138.44, 129.31, 128.53, 128.44, 127.50, 122.83, 122.65, 122.46, 122.07, 121.90, 121.55, 121.14, 120.70, 116.15, 114.68, 16.54.

Figure SXII. Structural characterization of 4DP-H-BN. Structural analysis of 4DP-H-BN with (a) 1D ¹H NMR analyses combined with HRMS, MS (GC-FAB-HRMS): calc'd for C₅₅H₄N₅ [M+H]⁺: 772.3440; found as 772.3444, and (b) intense 2D NMR analyses including COSY, NOESY, HSQC and HMBC NMR. ¹H NMR (600 MHz, CDCl₃) δ 7.22–7.17 (t, 4H), 7.09–7.02 (m, 12H), 6.96 (t, 2H), 6.92–6.87 (t, 4H), 6.87–6.83 (t, 2H), 6.83–6.80 (t, 2H), 6.76 (s, 1H), 6.75–6.71 (d, 4H), 6.69–6.65 (t, 2H), 6.65–6.61 (m, 4H), 6.58–6.53 (t, 4H). ¹³C NMR (151 MHz, CDCl₃) δ 151.76, 150.19, 148.79, 146.38, 146.37, 145.05, 144.21, 136.47, 129.23, 128.56, 128.30, 127.41, 125.51, 123.66, 123.53, 123.51, 122.70, 121.92, 121.15, 114.66, 108.55.

Analysis of the 2nd rebuttal submission by Kwon, Wannemacher, and Gierschner and coworkers.

Overall, the authors have significantly improved the manuscript by shifting the experimental focus to the degradation reaction and exploring this reaction with a diverse selection of photocatalysts. They have obtained experimental support to complement theoretical studies for the involvement of the base in the degradation. They have also improved their kinetic model to obtain better agreement with experiment and addressed concerns about characterization. As such, I view the significance issue as mainly addressed, given the wide use of cyanoarene type photocatalysts in photocatalysis at large.

However, the organization and story of the manuscript has not been updated accordingly. The current emphasis in the introduction is on generation of high concentrations of radical anions, which, in my opinion, does not concisely introduce the major finding of the work herein of a general and previously unknown degradation pathway of cyanoarene type photocatalysts. I would suggest moving some of the material in the “Design” section to the introduction as it discusses the wide use of 4DP-IPN, the main PC studied, in photocatalysis at large. Further, expanding on this section to convey the broad use of 4Cz-IPN and any of the other PCs studied which appear in the photocatalysis methodology literature would add to the impact of the findings.

Overall, with those changes to improve readability of the manuscript, I will be supportive of its publication in *Nature Communications* provided that the minor concerns below are also addressed.

Minor concerns:

1. “We carefully monitored the degradation of 4DP-IPN in the presence of the sacrificial reductant (i.e., DIPEA) alone under both 455 nm and 515 nm LED irradiation, and the degradation under 455 nm illumination was far faster than that under 515 nm illumination.”
 - a. This wavelength dependence is unexpected as photoinduced oxidation of DIPEA should occur in this case from the lowest triplet excited state, T1, regardless of whether higher/lower energy excitation light is used, in accordance with Kasha’s rule. The wavelength dependence here may suggest a more complicated mechanism, for example one including an intramolecular step that is fast enough to compete with relaxation from S2+/T2+ to S1 or T1. Can the authors comment on the origin of the wavelength dependence?
 - b. Have the authors considered the alternate hypothesis of excited state C-CN bond cleavage in the PC, followed by radical addition to generate R-substituted PCs?
2. “However, it is generally very difficult to achieve a high concentration of PC•– with highly negative Ered 0(PC) (see Section 2.1. ‘Design strategy’), and thus, PC•– is normally observed in PCs with a less negative reduction potential (e.g., perylene diimide, Acr-Mes+BF4 –, Rh6G, and Ru(bpy)3 2+)7,8,23–25 or under special reaction conditions (Figure S1 and Table S1).21,26–28”
 - a. This statement is unclear as there is no quantitative evaluation of the concentration of PC•– achieved in these works, or herein. With unknown molar absorptivity, the concentration of PC•– cannot be calculated from UV-vis spectra unless complete conversion can be confirmed.

3. "The formation of the PC \bullet^- was further confirmed by (time-dependent) density functional theory, (TD-)DFT, calculations (using the B3LYP functional, 6-311++G * basis set, and the polarizable continuum model, PCM, with acetonitrile as the solvent)"
 - a. While calculations provide some support the dark colored species is PC \bullet^- , the persistent lifetime of this species should enable facile characterization by EPR to confirm that it is in fact a radical and not some other diamagnetic species, such as the dianion PC $^{2-}$ as has been observed in other systems (e.g. reduction of naphthalene diimides, see Acc. Chem. Res. 2018, 51, 2225–2236)
4. "Notably, 455 nm LED irradiation gave better results (i.e., faster reaction kinetics) than 515 nm LED irradiation, which seems to contradict the results of the UV/Vis experiments concerning the degradation of the PC \bullet^- . This inconsistency was most likely due to the fact that the ET between the PC \bullet^- and the substrate occurred much faster than the photodegradation of the PC \bullet^- ."
 - a. How does the light source affect kinetics here? (i.e. compare light flux, possible unproductive absorption of other species). It would be interesting if you find that with equal photon flux the green irradiation is actually superior, consistent with your other findings.
5. "The fact that the catalytic performance of 4DP-Me-BN was worse than that of 4DP-IPN might be surprising, considering that 4DP-Me-BN has a i) highly negative reduction potential, ii) decent long-lived T1 generation, and iii) greatly improved stability."
 - a. Although the T1 state reacts with DIPEA at a lower rate compared with the undegraded PC, did the authors check for S1 state quenching? At the high concentrations of DIPEA used in reactions, the S1 state reactivity could be relevant despite its short lifetime. To evaluate this possibility, it is crucial to use concentrations in the quenching experiment that reflect the reaction conditions.
6. "In other words, in the presence of 4-bromobenzonitrile, the ET from 4DP-IPN \bullet^- to 4-bromobenzonitrile seemed to be significantly faster than i) the formation of 4DP-IPN \bullet^- and ii) the substitution reaction to form 4DP-Me-BN, which results in a substantial delay in the PC degradation."
 - a. I don't understand claim i) – why would presence of substrate affect the PET event between the PC * and DIPEA? Does the substrate react with PC * ?
7. "Perfect isosbestic points (at 428 and 493 nm) also appeared for the reaction, indicating the absence of intermediates other than 4DP-IPN and its PC radical anion."
 - a. This statement involves a common misconception. If the additional species absorb at the right wavelengths, clear isosbestic points can still be observed. The observation of isosbestic points cannot rule out the presence of additional species.
8. In figure 2F, the phenothiazine PC is not identified. What is the structure of this PC? Phenothiazines typically have long-lived triplet excited states – only plotting the singlet state is hardly a fair comparison.
9. The title of the manuscript ("Formation and degradation of strongly reducing radical anions for highly efficient purely organic photoredox catalysis") is unspecific, specifically in terms of what is meant by the second half "for highly efficient purely organic photoredox catalysis". It is unclear what aspect of the system under study is "highly efficient". The use of "purely organic" also seems odd since no discussion in the manuscript is devoted to the "organic" nature of the PCs (presumably as opposed to those containing precious metals?)

10. Authors state in the introduction “This formerly inaccessible redox-neutral reaction strategy has enabled significant developments in radical chemistry for organic and polymer synthesis.^{6–19}”
 - a. Photoredox catalysis isn't always redox-neutral, and it is unclear how this discussion is relevant to the work herein. The reaction studied in this work is net reductive.

Reviewers' comments to author:

We are grateful to the reviewers for their constructive comments and suggestions, which greatly helped to clarify important points, avoid misunderstandings, and improve the clarity and referencing of the manuscript for publication in *Nature Communications*. We have carefully revised our manuscript in line with all the reviewers' comments. Our point-by-point responses are given below. Our modifications in response to the reviewers' comments are highlighted in *blue color* below and in the revised manuscript and SI.

Analysis of the 2nd rebuttal submission by Kwon, Wannemacher, and Gierschner and coworkers. Overall, the authors have significantly improved the manuscript by shifting the experimental focus to the degradation reaction and exploring this reaction with a diverse selection of photocatalysts. They have obtained experimental support to complement theoretical studies for the involvement of the base in the degradation. They have also improved their kinetic model to obtain better agreement with experiment and addressed concerns about characterization. As such, I view the significance issue as mainly addressed, given the wide use of cyanoarene type photocatalysts in photocatalysis at large.

However, the organization and story of the manuscript has not been updated accordingly. The current emphasis in the introduction is on generation of high concentrations of radical anions, which, in my opinion, does not concisely introduce the major finding of the work herein of a general and previously unknown degradation pathway of cyanoarene type photocatalysts. I would suggest moving some of the material in the "Design" section to the introduction as it discusses the wide use of 4DP-IPN, the main PC studied, in photocatalysis at large. Further, expanding on this section to convey the broad use of 4Cz-IPN and any of the other PCs studied which appear in the photocatalysis methodology literature would add to the impact of the findings.

Overall, with those changes to improve readability of the manuscript, I will be supportive of its publication in *Nature Communications* provided that the minor concerns below are also addressed.

Response: We thank the reviewer for his encouraging comments and constructive suggestions. Following the reviewer's suggestion, the introduction part of the manuscript has been rewritten to better show the importance of this work as follows: *To further enhance the efficiency and expand the reaction scope of visible-light-driven photoredox catalysis, it is essential to maximize the reducing power and concentration of active PC species that activate the substrate of interest through an electron transfer (ET) process. This can mostly be facilitated in a reductive quenching cycle wherein a one-electron-reduced PC (i.e., $PC^{\bullet-}$) commonly acts as an active PC species because $PC^{\bullet-}$ usually exhibits a far longer lifetime than the optically excited PC species (i.e., $^1,^3PC^*$) acting as an active PC intermediate in an oxidative quenching cycle.¹⁹ Moreover, $PC^{\bullet-}$ is regarded as a core intermediate for the recently*

proposed multiphoton excitation catalysis mechanism based on consecutive photoinduced electron transfer (ConPET)^{20,21} and electrophotocatalysis,^{19–23} therefore, merits special attention.

The concentration of $PC^{\bullet-}$ in the photostationary state and the ground state reduction potential of the PC ($E_{red}^0(PC)$) both play critical roles in photoredox-mediated catalytic reactions employing $PC^{\bullet-}$ as an active species.¹⁹ A high concentration of $PC^{\bullet-}$ implies a high collision frequency with the substrate under illumination by visible light, which facilitates ET events. In addition, a more negative E_{red}^0 indicates an increase in the driving force for ET, thus accelerating ET processes. To achieve a highly negative E_{red}^0 of a PC, a high energy lowest unoccupied molecular orbital (LUMO) is required.¹² Furthermore, to ensure visible light absorption by such a PC, the energy of the highest occupied molecular orbital (HOMO) should scale with that of the LUMO; however, the accompanying decrease in E_{red}^* is detrimental to the photoinduced electron transfer (PET) between a PC and a reductant. In other words, it is very difficult to target PCs combining the following properties: i) good visible light absorption, ii) adequate initial PET with a sacrificial reductant, and iii) a highly negative reduction potential. Thus, the generation of $PC^{\bullet-}$ is normally targeted in PCs with a less negative reduction potential (e.g., perylene diimide,^{21,24} $Acr-Mes^+BF_4^-$,^{7,25} Rh6G,²⁶ and $Ru(bpy)_3Cl_2$ ^{27,28}) or under special reaction conditions (**Figure S1** and **Table S1** in the Supplementary Information (SI)).^{20,23,29,30}

As exemplified by the recent reports by the groups of Zhang,³¹ Zeitler,³² Kwon,¹² and others,^{11,33–38} cyanoarenes have emerged as attractive organic PCs. Such PCs exhibit excellent catalytic performances for a variety of visible light-driven organic reactions^{33–35} and polymerizations.^{39–42} Among them, 4DP-IPN and its analogues have attracted considerable interest owing to their superior catalytic performance for radical anion-mediated photoredox catalysis. For example, Wickens et al. reported the successful photocatalyzed reductive cleavage of strong $C(sp^2)-N$ and $C(sp^2)-O$ bonds by the electrochemically generated 4DP-IPN radical anion.³⁷ More recently, the groups of Wickens et al.³⁸ and Wu et al.³⁶ used 4DP-IPN analogues as PCs to perform the phosphorylation, borylation, and hydroarylation of highly inactivated aryl chlorides. Through careful characterization of the radical anion of 4DP-IPN, they proposed that its high reducing power ($E_{red}^0 = -1.66$ V) is the crucial factor. However, despite these research efforts, it is still unclear which factors affect the formation and degradation of the radical anion of cyanoarene-based PCs. This lack of understanding can lead to inefficiencies in radical anion-mediated photoredox reactions such as inappropriate choice of PC, excessive PC loading, and inadequate selection of the excitation source. However, no studies have focused on the in-depth investigation of the cyanoarene-based PC radical anion.

Herein, we investigate the formation and degradation of the radical anion of cyanoarene-based PCs under widely-used photoredox-mediated reaction conditions. Through the investigation of various cyanoarene-based PCs with different redox potentials and abilities to generate triplet excited states (T_1),

we found that organic PCs exhibiting both the ultra-efficient generation of long-lived T_1 and adequately positive excited state reduction potentials ($E_{red}^(PC)$) enable these PCs to efficiently form strongly reducing $PC^{\cdot-}$ under mild visible light illumination. During the screening of these cyanoarenes, we also found that the different photodegradation behaviors of PCs depend on their electronic and steric properties. We also identified a strong correlation between the photodegradation reaction of cyanoarenes and the abilities of PCs to be one-electron reduced. To further investigate the photodegradation behavior of $PC^{\cdot-}$ in actual photoredox catalysis, we carried out the reductive dehalogenation of aryl halides as a model reaction. From in-situ monitoring of the reaction, we revealed that the dehalogenation and photodegradation of PCs are co-dependent on the rate of the ET process. Furthermore, we demonstrated the highly efficient dehalogenation of aryl/alkyl halides at very small loading of 4DP-IPN with a high oxygen tolerance; it thus outperformed other conventional PCs that were used as controls.*

The abstract has also been changed as follows: “Cyanoarene-based photocatalysts (PCs) have attracted significant interest owing to their superior catalytic performance for radical anion-mediated photoredox catalysis. However, the factors affecting the formation and degradation of the radical anions of cyanoarene-based PCs are still insufficiently understood. We therefore investigated the formation and degradation of the radical anion of cyanoarene-based PCs under widely-used photoredox-mediated reaction conditions. By screening various cyanoarene-based PCs, we elucidated strategies to efficiently generate PC radical anions ($PC^{\cdot-}$) with adequate excited-state reduction potentials (E_{red}^*) via supra-efficient generation of long-lived triplet excited states (T_1). To thoroughly investigate the behavior of $PC^{\cdot-}$ in actual photoredox-mediated reactions, we carried out a reductive dehalogenation as a model reaction and identified the dominant photodegradation pathways of the $PC^{\cdot-}$. Dehalogenation and photodegradation of $PC^{\cdot-}$ are coexistent depending on the rate of electron transfer (ET) to the substrate and the photodegradation strongly depends on the electronic and steric properties of the PCs. Based on the understanding of both the formation and photodegradation of $PC^{\cdot-}$, we demonstrate that the efficient generation of highly reducing $PC^{\cdot-}$ allows for the highly efficient photoredox catalyzed dehalogenation of aryl/alkyl halides at a PC loading as low as 0.001 mol% with a high oxygen tolerance. The present work provides new insights into the reactions of cyanoarene $PC^{\cdot-}$ in photoredox-mediated reactions.”

Minor concerns:

1. “We carefully monitored the degradation of 4DP-IPN in the presence of the sacrificial reductant (i.e., DIPEA) alone under both 455 nm and 515 nm LED irradiation, and the degradation under 455 nm illumination was far faster than that under 515 nm illumination.”

a. This wavelength dependence is unexpected as photoinduced oxidation of DIPEA should occur in this case from the lowest triplet excited state, T1, regardless of whether higher/lower energy excitation light is used, in accordance with Kasha's rule. The wavelength dependence here may suggest a more complicated mechanism, for example one including an intramolecular step that is fast enough to compete with relaxation from S2+/T2+ to S1 or T1. Can the authors comment on the origin of the wavelength dependence?

Response: We thank the reviewer for their insightful comments. It is very difficult to confirm the full kinetics of a photodegradation reaction; however, from in-situ monitoring by TLC (**Figure 5a** in original manuscript; **Figure 4a** in the revised manuscript), we could check that the conversion of 4DP-IPN is dependent on the wavelength. We are of the opinion that the wavelength-dependency originates from the differing absorption of 4DP-IPN at 455 and 515 nm. To clarify this, we have revised the manuscript as follows: *"We carefully monitored the degradation of 4DP-IPN in the presence of the sacrificial reductant (i.e., DIPEA) alone under both 455 and 515 nm LED irradiation, and the degradation under 455 nm illumination was much faster than that under 515 nm illumination. This can possibly be ascribed to differences in the absorption efficiencies ($\epsilon = 9.0 \times 10^3 \text{ M}^{-1} \cdot \text{cm}^{-1}$ at 455 nm and $\epsilon = 4.5 \times 10^2 \text{ M}^{-1} \cdot \text{cm}^{-1}$ at 515 nm)."*

b. Have the authors considered the alternate hypothesis of excited state C-CN bond cleavage in the PC, followed by radical addition to generate R-substituted PCs?

Response: Thank you for the reviewer's comments. As we mentioned in the original manuscript, the photodegradation mechanism of cyanoarene-based PCs is influenced by a variety of factors, such as β -scission of tertiary amines and steric effects in the structure of the PC. As part of follow-up investigations, we are preparing various cyanoarenes and evaluating them under various reaction conditions as part of an in-depth mechanism study of photodegradation.

2. "However, it is generally very difficult to achieve a high concentration of PC^{•-} with highly negative E_{red} 0(PC) (see Section 2.1. 'Design strategy'), and thus, PC^{•-} is normally observed in PCs with a less negative reduction potential (e.g., perylene diimide, Acr-Mes+BF₄⁻, Rh6G, and Ru(bpy)₃²⁺).^{7,8,23-25} or under special reaction conditions (Figure S1 and Table S1).^{21,26-28}"

a. This statement is unclear as there is no quantitative evaluation of the concentration of PC^{•-} achieved in these works, or herein. With unknown molar absorptivity, the concentration of PC^{•-} cannot be calculated from UV-vis spectra unless complete conversion can be confirmed.

Response: Thank you for the careful revision of our manuscript. We recognize that our statements can mislead the reader without full quantitative evaluation of the concentration of the PC radical anion. Therefore, to avoid misunderstanding, we reworded the statements as follows: *"Thus, the generation of PC^{•-} is normally targeted in PCs with a less negative reduction potential (e.g., perylene diimide,^{21,24}*

Acr-Mes⁺BF₄⁻,^{7,25} Rh6G,²⁶ and Ru(bpy)₃Cl₂^{27,28}) or under special reaction conditions (Figure S1 and Table S1 in the Supplementary Information (SI)).^{20,23,29,30}

3. “The formation of the PC•⁻ was further confirmed by (time-dependent) density functional theory, (TD-)DFT, calculations (using the B3LYP functional, 6-311++G* basis set, and the polarizable continuum model, PCM, with acetonitrile as the solvent)”

a. While calculations provide some support the dark colored species is PC•⁻, the persistent lifetime of this species should enable facile characterization by EPR to confirm that it is in fact a radical and not some other diamagnetic species, such as the dianion PC²⁻ as has been observed in other systems (e.g. reduction of naphthalene diimides, see Acc. Chem. Res. 2018, 51, 2225–2236)

Response: Thank you for the constructive comments. As we responded in the last revision, the PC⁻ of 4DP-IPN was already characterized by Wickens’ group (*Angew. Chemie Int. Ed.* **60**, 21418–21425 (2021)) with EPR spectroscopy to confirm that the generated species are one-electron reduced radicals. We therefore characterized the PC radical anion using UV-Vis absorption and computational methods (TD-DFT) instead of EPR spectra. To stress that the EPR characterization of 4DP-IPN⁻ has been reported, we placed more emphasis on the work of Wickens’s group in the revised manuscript as follows: “More recently, the groups of Wickens et al.³⁸ and Wu et al.³⁶ used 4DP-IPN analogues as PCs to perform the phosphorylation, borylation, and hydroarylation of highly inactivated aryl chlorides. Through careful characterization of the radical anion of 4DP-IPN, they proposed that its high reducing power ($E_{red}^0 = -1.66$ V) is the crucial factor.”

4. “Notably, 455 nm LED irradiation gave better results (i.e., faster reaction kinetics) than 515 nm LED irradiation, which seems to contradict the results of the UV-Vis experiments concerning the degradation of the PC•⁻. This inconsistency was most likely due to the fact that the ET between the PC•⁻ and the substrate occurred much faster than the photodegradation of the PC•⁻.”

a. How does the light source affect kinetics here? (i.e. compare light flux, possible unproductive absorption of other species). It would be interesting if you find that with equal photon flux the green irradiation is actually superior, consistent with your other findings.

Response: We are grateful for the reviewer’s comment. As we responded in 1-a, we suggest that the origin of the fast kinetics of the dehalogenation and photodegradation of PC is because both reactions are determined equally by the formation of the PC radical anion. We used the same power LEDs (3 W) for blue and green light irradiation, respectively, to compare them in an unbiased way, as the reviewer commented; however, considering the photon energy, the photon flux of the LEDs can be different. We are in the process of identifying and acquiring the proper irradiation source for future investigations.

5. “The fact that the catalytic performance of 4DP-Me-BN was worse than that of 4DP-IPN might be

surprising, considering that 4DP-Me-BN has a i) highly negative reduction potential, ii) decent long-lived T1 generation, and iii) greatly improved stability.”

a. Although the T1 state reacts with DIPEA at a lower rate compared with the undegraded PC, did the authors check for S1 state quenching? At the high concentrations of DIPEA used in reactions, the S1 state reactivity could be relevant despite its short lifetime. To evaluate this possibility, it is crucial to use concentrations in the quenching experiment that reflect the reaction conditions.

Response: Thank you for the excellent comments. Yes, we already checked the possibility of S₁ quenching (*Figure XI* and revised *Figure S5-2*) of 4DP-IPN and 4DP-Me-BN with prompt fluorescence quenching experiments in the presence of high concentration of DIPEA (0–0.4 M); the concentration of DIPEA in the actual dehalogenation reactions is 1 M; however, due to solubility issues of DIPEA in CH₃CN, we reduced the concentration of DIPEA in preparation of the measurement samples. According to prompt fluorescence decay, we can conclude that the PET event between S₁ and DIPEA is negligibly small. To clarify more, we revised the SI with adding prompt PL quenching experiments as well.

Figure XI. Stern-Volmer plots of prompt fluorescence of (a) 4DP-IPN and (b) 4DP-Me-BN quenched by DIPEA (0–0.4 M) in CH₃CN (1.0×10^{-5} M).

6. “In other words, in the presence of 4-bromobenzonitrile, the ET from 4DP-IPN•– to 4-bromobenzonitrile seemed to be significantly faster than i) the formation of 4DP-IPN•– and ii) the substitution reaction to form 4DP-Me-BN, which results in a substantial delay in the PC degradation.”

a. I don’t understand claim i) – why would presence of substrate affect the PET event between the PC* and DIPEA? Does the substrate react with PC*?

Response: Thank you for the comments. In the presence of 4-bromobenzonitrile or 4-iodobenzonitrile, we observed that the photodegradation reaction evidently slows down (*Figure X2* and *Figure 5a* of the revised manuscript). Our statements meant that before accumulation of the PC radical anion, it undergoes an electron transfer process to the substrate. From these observations, we concluded that this

electron transfer process is much faster than the photodegradation reaction of PC radical anion, thus, the PC radical anion have a decreased chance to produce the photodegraded adduct.

Figure X2. Fate of catalyst during dehalogenation reactions. Reactions were performed with substrates (0.1 M), DIPEA (10.0 equiv.), and 4DP-IPN (1 mol%, 1.0×10^{-3} M) in CH_3CN (1 mL) under the illumination of two 3 W 455 nm LEDs for several hours at RT. Yields were determined by ^1H NMR using 1,3,5-trimethoxybenzene as an internal standard. PC degradation was monitored by in-situ TLC with eluent conditions EA:hexanes = 1:4 v/v. All redox potential values were obtained from the literature where the potential values were measured against the standard calomel electrode (SCE).

7. “Perfect isosbestic points (at 428 and 493 nm) also appeared for the reaction, indicating the absence of intermediates other than 4DP-IPN and its PC radical anion.”

a. This statement involves a common misconception. If the additional species absorb at the right wavelengths, clear isosbestic points can still be observed. The observation of isosbestic points cannot rule out the presence of additional species.

Response: We thank the reviewer for their comment. To avoid the misunderstanding, we have revised the statements in the manuscript as follows: “A decrease in the absorption peak at 470 nm implied the depletion of 4DP-IPN, while a new broad absorption band appeared at 500–750 nm, indicating the generation of $4\text{DP-IPN}^{\cdot-}$; perfect isosbestic points (at 428 and 493 nm) also appeared for the reaction.”

8. In figure 2F, the phenothiazine PC is not identified. What is the structure of this PC? Phenothiazines typically have long-lived triplet excited states – only plotting the singlet state is hardly a fair comparison.

Response: We thank the reviewer for their insightful comments. As the reviewers points out, the class of phenothiazines normally have long-lived triplet excited states (e.g., *Polym. Chem.*, **7**, 6039 (2016), *J. Phys. Chem. A* **124**, 817–823 (2020), and *J. Am. Chem. Soc.*, **143**, 3613–3627 (2021)); however, our selected phenothiazine PC is 10-phenylphenothiazine (PTH, **Figure X3b**), it is well-known that its S_1 excited state mainly contributes to PET (e.g., *Chem. Rev.* **116**, 10075–10166 (2016), *Chem. Commun.* **51**, 11705–11708 (2015), and *J. Am. Chem. Soc.*, **138**, 2411–2425 (2016)). Thus, to demonstrate that the T_1 in excited state can make the exciton concentration much higher, we needed to select the PTH as a

PC with well-known S_1 character. To clarify our purpose, we have revised the manuscript by adding the chemical structure of PCs into *revised Figure 1f* and the full name of PTH to the caption/manuscript.

Figure X3. Reported studies of phenothiazine derivatives as a PC in the (a) singlet (S_1) and (b) triplet (T_1) excited state.

9. The title of the manuscript (“Formation and degradation of strongly reducing radical anions for highly efficient purely organic photoredox catalysis”) is unspecific, specifically in terms of what is meant by the second half “for highly efficient purely organic photoredox catalysis”. It is unclear what aspect of the system under study is “highly efficient”. The use of “purely organic” also seems odd since no discussion in the manuscript is devoted to the “organic” nature of the PCs (presumably as opposed to those containing precious metals?)

Response: Thank you for the kind comments. To avoid misunderstanding, we have revised the title of the manuscript as follows: “Formation and degradation of strongly reducing cyanoarene-based radical anions towards efficient radical anion-mediated photoredox catalysis”

10. Authors state in the introduction “This formerly inaccessible redox-neutral reaction strategy has enabled significant developments in radical chemistry for organic and polymer synthesis.6–19”

a. Photoredox catalysis isn’t always redox-neutral, and it is unclear how this discussion is relevant to the work herein. The reaction studied in this work is net reductive.

Response: We thank the reviewer for pointing this out to us. To avoid misunderstanding, we have removed the relevant sentences from the revised manuscript.

REVIEWERS' COMMENTS

Reviewer #4 (Remarks to the Author):

I would like to commend the authors for addressing my concerns and responding to my comments and am supportive of publication of the manuscript in Nature Communications.